# Lagrangian simulations of the transport of young air masses to the top of the Asian monsoon anticyclone and into the tropical pipe

Bärbel Vogel[1], Rolf Müller[1], Gebhard Günther[1], Reinhold Spang[1], Sreeharsha Hanumanthu[1], Dan Li[1,2], Martin Riese[1], and Gabriele P. Stiller[3]

[1]Forschungszentrum Jülich, Institute of Energy and Climate Research - Stratosphere (IEK-7), Jülich, Germany
[2]Key Laboratory of Middle Atmosphere and Global Environment Observation (LAGEO), Institute of Atmospheric Physics, Chinese Academy of Sciences, Beijing, China
[3]Institute for Meteorology and Climate Research, Karlsruhe Institute of Technology, Karlsruhe, Germany

**Correspondence:** Bärbel Vogel (b.vogel@fz-juelich.de)

**Abstract.**

We have performed backward trajectory calculations and simulations with the 3-dimensional Chemical Lagrangian Model of the Stratosphere (CLaMS) for two succeeding monsoon seasons using artificial tracers of air mass origin. With these tracers we trace back the origin of young air masses (age < 6 months) at the top of the Asian monsoon anticyclone and of air masses within

the tropical pipe (6 months < age <18 months) during summer 2008. The occurrence of young air masses (<6 months) at the top of the Asian monsoon anticyclone up to ≈460 K is in agreement with satellite measurements of chlorodifluoromethane (HCFC-22) by the Michelson Interferometer for Passive Atmospheric Sounding (MIPAS) instrument. HCFC-22 can be considered as a regional tracer for continental eastern Asia and the Near East as it is mainly emitted in this region.

Our findings show that the transport of air masses from boundary layer sources in the region of the Asian monsoon into

the tropical pipe occurs in three distinct steps. First, very fast uplift in 'a convective range' transports air masses up to 360 K potential temperature within a few days. Second, air masses are uplifted from about 360 K up to 460 K within 'an upward spiralling range' within a few months. The large-scale upward spiral extends from northern Africa to the western Pacific. The air masses are transported upwards by diabatic heating with a rate of up to 1–1.5 K per day, implying strong vertical transport above the Asian monsoon anticyclone. Third, transport of air masses occurs within the tropical pipe up to 550 K associated

with the large-scale Brewer-Dobson circulation within ∼ one year.

In the upward spiralling range, air masses are uplifted by diabatic heating across the (lapse rate) tropopause, which does not act as a transport barrier in contrast to the extratropical tropopause. Further, in the upward spiralling range air masses from inside the Asian monsoon anticyclone are mixed with air masses convectively uplifted outside the core of the Asian monsoon anticyclone in the tropical adjacent regions. Moreover, the vertical transport of air masses from the Asian monsoon anticyclone

into the tropical pipe is weak in terms of transported air masses compared to the transport from the monsoon anticyclone into the northern extratropical lower stratosphere. Air masses from the Asian monsoon anticyclone (India/China) contribute a minor fraction to the composition of air within the tropical pipe at 550 K (6%), the major fractions are from Southeast Asia (16%) and the tropical Pacific (15%).

# 1 Introduction

The Asian summer monsoon is associated with deep convection over the Indian subcontinent and with an anticyclonic flow that extends from the upper troposphere into the lower stratosphere (UTLS) region which is the most pronounced circulation pattern in these altitudes during boreal summer (e.g., Mason and Anderson, 1963; Li et al., 2005; Randel and Park, 2006; Park et al., 2007). The strong anticyclonic circulation in the UTLS acts as an effective transport barrier (e.g., Ploeger et al., 2015) causing a confinement of tropospheric trace gases in the anticyclone, isolating them from the surrounding air (stratospheric background) as shown by a variety of satellite measurements (e.g., Rosenlof et al., 1997; Li et al., 2005; Park et al., 2007; Fadnavis et al., 2014; Glatthor et al., 2015; Chirkov et al., 2016; Santee et al., 2017).

The transport of tropospheric trace gases by the Asian monsoon anticyclone into the lower stratosphere changes the chemical composition in this part of the Earth's atmosphere. Radiatively active species transported into the lowermost extratropical stratosphere have a significant impact on surface climate (e.g., Solomon et al., 2010; Riese et al., 2012; Hossaini et al., 2015) or can cause regional radiative forcing such as the Asian tropopause aerosol layer (ATAL) (e.g., Vernier et al., 2015).

There is large variability of the spatial extent, strength, and location of the monsoon anticyclone in the UTLS, which reaches from Northeast Africa to East Asia (e.g., Annamalai and Slingo, 2001; Randel and Park, 2006; Garny and Randel, 2013; Vogel et al., 2015; Pan et al., 2016). In particular, the location and the shape of the anticyclone change from day to day caused by internal dynamical variability, manifesting in an oscillation between a state with one anticyclone and two separated anticyclones (2 modes) often referred to as western (Iranian) and eastern (Tibetan) mode (e.g., Zhang et al., 2002; Vogel et al., 2015; Nützel et al., 2016). In addition, smaller anticyclones characterised by low PV values break off a few times each summer from the main anticyclone, a process which is referred to as "eddy shedding" (Hsu and Plumb, 2001; Popovic and Plumb, 2001; Garny and Randel, 2013; Vogel et al., 2014, 2016; Ungermann et al., 2016).

The Asian monsoon circulation provides an effective pathway for tropospheric trace gases such as pollutants, gaseous aerosol precursors, as well as aerosol particles into the lower stratosphere which could play an important role in the formation of the ATAL layer (e.g., Vernier et al., 2015, 2018; Höpfner et al., 2016; Brunamonti et al., 2018). There is also export of monsoon air quasi-isentropically out of the monsoon and a certain fraction of monsoon air may reach greater altitudes in the stratosphere. There is a longstanding debate about the transport mechanisms at the top of the Asian monsoon anticyclone and beyond into the stratosphere (e.g., Bannister et al., 2004; Park et al., 2009; Randel et al., 2010; Bergman et al., 2012, 2013; Randel and Jensen, 2013; Uma et al., 2014; Orbe et al., 2015; Garny and Randel, 2016; Tissier and Legras, 2016; Ploeger et al., 2017). In the literature different aspects of the complex interplay between convection, large-scale upward transport (driven by radiative heating), and the anticyclonic flow in the UTLS are highlighted. Randel et al. (2010) pointed out that the monsoon circulation provides an effective pathway for pollution from Asia to enter the global stratosphere. Vertical upward transport into the deep stratosphere occurs within the tropical pipe, where tropical air masses are isolated to some extent from isentropic mixing with mid-latitude air (e.g., Plumb, 1996; Volk et al., 1996). Pan et al. (2016) highlight that the Asian monsoon anticyclone is an isolated 'bubble' of tropospheric air above the global mean tropical tropopause that isentropically sheds tropospheric air into

the stratosphere. Further, they argue that the vertical transport of Asian monsoon air into the deep stratosphere is inefficient during summer.

Here, we investigate two main questions: First, what are the transport pathways at the top of the Asian monsoon anticyclone into the stratosphere? Second, how do boundary layer source regions in Asia affect the composition of the lower/middle stratosphere within the tropical pipe?

To address these questions we performed both backward trajectory calculations and three-dimensional simulations including irreversible mixing (Konopka et al., 2007) with the Lagrangian transport model CLaMS (McKenna et al., 2002b, a; Pommrich et al., 2014, and references therein). Artificial tracers of air mass origin that mark defined regions in the Earth's boundary layer (covering the entire Earth's surface) are introduced in the CLaMS model (Vogel et al., 2015, 2016) and are compared to measurements of the Michelson Interferometer for Passive Atmospheric Sounding (MIPAS) instrument onboard the European Environmental Satellite (Envisat) (Fischer et al., 2008; Chirkov et al., 2016) to study transport processes and pathways at the top of the Asian monsoon anticyclone and beyond into the tropical pipe. We conduct a case study for the monsoon season 2008. The monsoon season 2008 is chosen because MIPAS measurements have very good data coverage in summer 2008 over Asia. Further, in 2008 there was a normal monsoon season in terms of rainfall over India in summer 2008[1]. It is established that the Indian monsoon is influenced by the El Niño Southern Oscillation (ENSO) (e.g., Kumar et al., 2006). There is evidence that a strong La Niña in winter (e.g. 2007/08 (DJF) according to the Oceanic Niño Index[2]) in combination with La Niña conditions during the subsequent summer (as in 2008) is correlated with normal rainfall over India with a certain variability in precipitation between different Indian regions (e.g., see Fig. 2c in Chakraborty, 2018).

We compare the distribution of tracers of air mass origin found in the CLaMS model with global chlorodifluoromethane (HCFC-22; $CHClF_2$) measurements from the MIPAS satellite instrument (Chirkov et al., 2016). Chirkov et al. (2016) found enhanced values of HCFC-22 in the region of the Asian monsoon anticyclone at 16 km altitude in July, August, and September (JAS) averaged over the MIPAS measurement period from 2005 until 2011. In the last few decades, HCFC-22 has been used as a substitute for more potent ozone-depleting substances such as chlorofluorocarbons (CFCs) in the chemical industry, in particular as a refrigerant, in some regions of the Earth, e.g., in continental eastern Asia and in the Near East (Fortems-Cheiney et al., 2013; Simmonds et al., 2018). In contrast, the production and utilisation of HCFC-22 has been phased out in developed countries regulated by the Montreal Protocol and its amendments and adjustments. As a consequence, HCFC-22 is emitted in locally restricted regions, in particular in the region of the Asian monsoon. Simmonds et al. (2018) estimate that between 55% and 65% of the global HCFC-22 emissions within the last decade are from Chinese production.

Therefore, HCFC-22 is a good tracer for studying transport processes in the region of the Asian monsoon anticyclone and for comparing with CLaMS artificial tracers of air mass origin (e.g., Vogel et al., 2016). In this paper similar methods as in Vogel et al. (2016), namely three-dimensional CLaMS simulations with artificial tracers of air mass origin as well as MIPAS HCFC-22, are used, however the model setup and the scientific objectives are different. In Vogel et al. (2016) horizontal transport pathways out of the Asian monsoon anticyclone from 360 K up to 400 K were analysed in a simulation for the monsoon

---

[1]see e.g. http://www.tropmet.res.in/~kolli/mol/Monsoon/Historical/air.html

[2]see e.g. http://ggweather.com/enso/oni.htm

season 2012. Vogel et al. (2016) found, in agreement with MIPAS HCFC-22 measurements, two main horizontal transport pathways from the Asian monsoon anticyclone: one to the east along the subtropical jet and subsequent transport into the northern lower stratosphere and a second horizontal transport pathway to the west into the tropical tropopause layer (TTL).

Here, a more sophisticated model setup is used with the focus on vertical transport pathways out of the Asian monsoon anticyclone and subsequent upward transport into the lower stratosphere up to 550 K. Two consecutive monsoon seasons in summer 2007 and 2008 are simulated. In this two-monsoon-season simulation, the different emission tracers are released during three different time periods each with a length of 6 months (i.e., we use a three-pulse-approach with a total simulation period of 18 months). This allows us to infer the different transport times of air parcels from the Earth's surface to the top of the anticyclone and beyond in contrast to the approach of a one-monsoon-season simulation (one-pulse-approach with a simulation period of 6 months) used earlier (Vogel et al., 2015, 2016). With this approach it is possible to quantify the impact of the monsoon season of the year before (2007), the winter time 2007/2008, and the monsoon season 2008 on the lower stratosphere and in particular on the tropical pipe at the end of August 2008. Furthermore, this model setup allows us to identify the origin of air masses found at the top of the Asian monsoon and within the tropical pipe as well as the transport times from the model boundary layer into the stratosphere including irreversible mixing processes.

## 2   CLaMS model simulations and MIPAS HCFC-22 measurements

We conduct model simulations with the three-dimensional chemistry transport model CLaMS (McKenna et al., 2002a, b; Pommrich et al., 2014, and references therein) and pure backward trajectory calculations with the CLaMS trajectory model covering the Asian monsoon season 2008.

The model simulations and trajectory calculations are driven by horizontal winds from ERA-Interim reanalysis (Dee et al., 2011) provided by the European Centre for Medium-Range Weather Forecasts (ECMWF). For the vertical velocities, the diabatic approach (with contributions to vertical velocities from radiative heating including the effects of clouds, latent heat release, mixing, and diffusion) was applied using diabatic heating rate as the vertical velocity including latent heat release (for details, see Ploeger et al., 2010). Further, CLaMS employs a hybrid vertical coordinate ($\zeta$), which transforms from a strictly isentropic coordinate $\Theta$ to a pressure-based coordinate system ($\sigma$ coordinates) below a certain reference level (in this study 300 hPa) (for more details, see Konopka et al., 2012; Pommrich et al., 2014).

The upward transport and convection in CLaMS (in both three-dimensional simulations as well as in trajectory calculations) is driven by ERA-Interim reanalysis data in which changes are implemented to improve deep and mid-level convection compared to previous reanalysis data (Dee et al., 2011). However, small-scale rapid uplift in convective cores is not included. Therefore convection over Asia is most likely underestimated in ERA-Interim. However, the focus of our paper is to understand the main transport pathways at the top of the anticyclone above 380 K and up to 460 K ($\approx$100-60 hPa), which is above the main level of tropical deep convection (e.g., Devasthale and Fueglistaler, 2010; Bergman et al., 2012). Further, previous studies demonstrated that the vertical transport in CLaMS allows the spatio-temporal distribution of carbon monoxide (CO)

within the Asian monsoon anticyclone measured by the Aura Microwave Limb Sounder (MLS) to be reproduced (Vogel et al., 2015; Ploeger et al., 2017).

## 2.1 Three-dimensional CLaMS simulations

The three-dimensional CLaMS simulations include irreversible mixing accounting for wind shear (Konopka et al., 2007) and
are therefore capable of reproducing strong gradients of atmospheric trace gases found in regions with strong transport barriers, such as the edge of the Asian monsoon anticyclone (e.g., Konopka et al., 2010; Vogel et al., 2015, 2016; Ploeger et al., 2015, 2017), the extratropical tropopause in the vicinity of the subtropical jet (e.g., Pan et al., 2006; Vogel et al., 2011, 2016), and the polar vortex (e.g., Günther et al., 2008; Vogel et al., 2008).

The three-dimensional global CLaMS simulations employed here cover an altitude range from the surface up to 900 K
potential temperature ($\approx 37\,\mathrm{km}$ altitude) with a horizontal resolution of $100\,\mathrm{km}$ and a maximum vertical resolution of approximately $400\,\mathrm{m}$ near the tropopause. A two-monsoon-season simulation is performed covering the time period from 1 May 2007 to 31 October 2008, including both the 2007 and 2008 Asian monsoon seasons to study the upwelling of surface air characterised by local emissions into the lower stratosphere and into the tropical pipe during the course of two succeeding monsoon seasons.

In the two-monsoon-season simulation, artificial tracers of air mass origin, referred to as "emission tracers", that mark
defined regions in the Earth's boundary layer (covering the entire Earth's surface) are implemented ($\approx 2$–$3\,\mathrm{km}$ above the surface following orography corresponding to $\zeta < 120\,\mathrm{K}$), as shown in Fig. 1 and Table 1. Within the model boundary layer, the sum of all the different emission tracers ($\Omega_i$) including the emission tracer for the background (remaining surface) is equal to 1 ($\Omega = \sum_{i=1}^{n} \Omega_i = 1$, see Table 1) at the mixing time step. Air masses in the model boundary layer are marked by different
emission tracers every 24 hours (the time step for mixing in CLaMS) (for details see Vogel et al., 2016).

We used a three-pulse-approach in the two-monsoon-season simulation and released the different emission tracers $\Omega_i$ in 3 different time periods (pulses) from $t_j$ until $t_j + \Delta t_j$ with $\Delta t_j$ equal to 6 months. For each pulse, the different emission tracers are continuously released (every 24 hours) at the model boundary between $t_j$ and $t_j + \Delta t_j$. The three pulses start at 1 May 2007 for the summer/fall season 2007 (Summer 07 = S07; $\Omega_{\mathrm{S07}} = \sum_{i=1}^{n} \Omega_{i,\mathrm{S07}}$), 1 November 2007 for the winter/spring
season 2007/2008 (Winter 07/08 = W07; $\Omega_{\mathrm{W07}} = \sum_{i=1}^{n} \Omega_{i,\mathrm{W07}}$), and 1 May 2008 for the summer/fall season 2008 (Summer 08 = S08; $\Omega_{\mathrm{S08}} = \sum_{i=1}^{n} \Omega_{i,\mathrm{S08}}$). The summer pulses were chosen to start a few weeks prior to the onset of the Asian monsoon. With this approach it is possible to quantify the impact of the monsoon season of the year before, namely Summer 07, on the lower stratosphere and in particular on the tropical pipe in summer 2008 in addition to the impact of younger air masses of Summer 08 (one-monsoon-season simulation) with an age lower than 6 months. Furthermore, with this approach also the
impact of the strong upwelling above the Maritime Continent (the region between Indian and Pacific oceans) and the western Pacific (emission tracers for Southeast Asia and the tropical Pacific Ocean) during Winter 07/08 on the tropical pipe in summer 2008 is quantified.

In our two-monsoon-season simulation, the composition of an air mass in the free atmosphere (outside of the model boundary layer) will be a combination of air masses younger than 1 May 2007 ($\Omega_{S08} + \Omega_{W07} + \Omega_{S07}$) and aged air masses ($A_{aged}$) older than 1 May 2007 originating in the free troposphere or stratosphere ($\Omega_{S08} + \Omega_{W07} + \Omega_{S07} + A_{aged} = 1$).

In a one-monsoon-season simulation for the year 2012, Vogel et al. (2015) showed that the emission tracers for northern India plus southern India plus eastern China (in the following referred to as 'India/China' tracer) are a good proxy for the location and shape of the Asian monsoon anticyclone using pattern correlations with potential vorticity (PV), and MLS $O_3$ and CO satellite measurements. Young air masses that are convectively uplifted outside the core of the Asian monsoon anticyclone and subsequently transported clockwise around the outer edge of the Asian monsoon mainly originate in Southeast Asia, the tropical Pacific, northwestern Pacific, and in northern Africa. Therefore, in this study the sum of these emission tracers is summarised in one emission tracer referred to as the tropical adjacent regions 'TAR'. Note that in this paper a new emission tracer for the northwestern Pacific (NWP) is introduced in the CLaMS simulation compared to previous studies (Vogel et al., 2015, 2016) because it was demonstrated that tropical cyclones in the Pacific and their interaction with the Asian monsoon anticyclone play an important role in the chemical composition of air masses found at the edge of the anticyclone (Vogel et al., 2014; Li et al., 2017).

We note that also minor fractions of the emission tracers from the tropical adjacent regions (in particular from Southeast Asia) are found inside the Asian monsoon. This is due to the south-north shift and east-west oscillations of the monsoon anticyclone itself (Vogel et al., 2015).

## 2.2 CLaMS backward trajectory calculations

The three-dimensional CLaMS simulations including mixing simulate the contribution of different source regions within the model boundary layer to an air parcel in the free atmosphere. Pure trajectory calculations consider only the advective transport neglecting mixing processes entirely. However, backward trajectories are very well suited to analyse the detailed transport pathway of an air parcel and therefore provide added value compared to three-dimensional CLaMS simulations (e.g., Vogel et al., 2014; Li et al., 2017).

Within this study, 20-day and 40-day backward trajectories are calculated driven by wind data (with a horizontal resolution of $1° \times 1°$) from the ERA-Interim reanalysis (Dee et al., 2011) and using the diabatic approach to analyse the transport pathways of air parcels at the top of the Asian monsoon anticyclone and beyond into the tropical pipe.

## 2.3 Calculation of thermal tropopause

An accurate tropopause height determination is crucial to analyse to what extent the thermal tropopause acts as a vertical transport barrier at the top of the Asian monsoon anticyclone. In the extra-tropics, the tropopause acts as a chemical transport boundary in contrast to the tropics.

Here, we use an improved determination of the lapse rate tropopause for ERA-Interim data developed by Spang et al. (2015). The vertical resolution of the retrieved tropopause height cannot be better than the vertical grid resolution of the temperature data and hence can produce a significant positive bias for analyses with tropopause related altitude coordinates (Pan and Munchak, 2011). To partly compensate for this effect a vertical spline interpolation with 30 m vertical resolution is applied to the temperature profile around the actual tropopause height computed with the original vertical resolution. The tropopause height computation is repeated with the artificially higher vertical resolution and a weighted mean with distance of the four surrounding grid points of the observation point represents now the so-called high resolution tropopause height. This approach delivers a more realistic lapse rate tropopause, because the single tropopause height values are no longer associated with the altitude grid points of the analysis data. Moreover, Spang et al. (2015) found with this approach smaller bias and standard deviation between ERA-Interim and radiosonde-based tropopause heights.

## 2.4 MIPAS HCFC-22 measurements

To compare the spatial distribution of CLaMS emission tracers with observations in the region of the Asian monsoon anticyclone, we compare results of the CLaMS simulation with global HCFC-22 measurements of the MIPAS satellite instrument (Data Version V5R) (Chirkov et al., 2016).

For MIPAS-CLaMS intercomparisons, the data density of HCFC-22 measurements is improved by synoptic interpolation of multiple days of MIPAS measurements using CLaMS 3-dimensional trajectory calculations, making use of the relatively long lifetime of HCFC-22 near the tropopause. For a specific day at the end of August 2008, trajectories were computed from the time of measurements in a time window of 4 days (i.e., $\pm 2$ days) to 12:00 UTC (Universal Time Coordinated) of the selected day. Over a period of a few days, there is practically no chemical destruction of HCFC-22 because of its global total atmospheric lifetime of about 12 years (SPARC Report 2013 'Lifetimes of Stratospheric Ozone-Depleting Substances', Ko et al., 2013).

A trajectory length of 2 days for the synoptic interpolation gives sufficient coverage of the MIPAS data in the region of the Asian monsoon. An even longer interpolation would give a higher data density, however by calculating solely trajectories mixing processes are neglected. Thus a time window of 4 days for the synoptic interpolation is a good compromise between sufficient data coverage and neglecting mixing processes.

The precision of an individual data point of the MIPAS HCFC-22 measurement in the altitude region of the Asian monsoon tropopause is 7 to 8 pptv in terms of measurement noise. Parameter errors contribute to a total uncertainty of about 15 pptv in this region for each data point. Thus, the scatter of the HCFC-22 data points (e.g. as shown in Fig. 4) is consistent with the total error. Further it has to be noted that tropical HCFC-22 profiles from MIPAS seem to have a high bias below 30 km, that, however, is constant with altitude (Chirkov et al., 2016); thus, it does not affect the comparisons made here. The horizontal resolution (in terms of the full width at half maximum of the horizontal averaging kernel) increases from 300 km at 15 km altitude to 600 km at 20 km altitude. In general, the limited vertical resolution of satellite remote sensing instruments like MIPAS needs to be taken into account in comparisons to model results. According to Chirkov et al. (2016), the vertical resolution (in terms of the full width at half maximum of the vertical averaging kernel) increases from about 3.3 km at 12 km to

5.5 km at 20 km altitude (see Fig. 2 in Chirkov et al., 2016). Given the rather smooth profiles expected in this study, however, the limited altitude resolution has a minor effect only; in contrast, it turns out to be crucial when highly structured profiles, such as typically occur at the edge of the polar vortex, are analysed.

## 3    Results

### 3.1    Impact of emission tracers of the Summer 08 pulse

#### 3.1.1    Contribution of different emission tracers to the top of the Asian monsoon anticyclone

It is known that the Asian monsoon anticyclone has a strong horizontal transport barrier at about 380 K (e.g., Ploeger et al., 2015), however this transport barrier is not well defined at higher levels of potential temperature. The weaker transport barrier at higher levels has consequences for the vertical transport at the top of the anticyclone. Before the transport at the top is discussed we show the horizontal distribution of different emission tracers at 360 K and then their subsequent transport to the top of the anticyclone up to 460 K. Vogel et al. (2015) showed that the emission tracer for India/China is a good proxy for the location and shape of the Asian monsoon anticyclone using pattern correlations with potential vorticity (PV), and MLS $O_3$ and CO satellite measurements between 360 K and 400 K. Therefore here we use the India/China tracer as a proxy for the location of the anticyclone.

To analyse the transport pathways at the top of the Asian monsoon anticyclone during the monsoon season 2008, we use only the tracers of air mass origin for the time pulse for Summer 08 (started on 1 May 2008, running through the end of October 2008). The geographic position and shape of the Asian monsoon anticyclone show strong day-to-day variability (e.g., Garny and Randel, 2013; Ploeger et al., 2015; Vogel et al., 2015). In this paper, we focus on 18 August 2008 during the monsoon season 2008 as a case study. On that day, the anticyclone has two modes, the western mode located over the Near East and the eastern Mediterranean Basin and the eastern mode over India and western China as shown in Fig. 2a. Further, a smaller anticyclone (eddy shedding event) is found over the northwestern Pacific. We selected 18 August 2008 for this study because first this day is dynamically very interesting and second on this day there is very good data coverage of the MIPAS HCFC-22 measurements.

To analyse the transport of young air masses to the top of the Asian monsoon anticyclone, we distinguish between air masses that experienced strong upward transport mainly inside (India/China) and mainly outside (tropical adjacent regions) of the Asian monsoon anticyclone. Fig. 2a/b shows the horizontal distribution of the fraction of the emission tracer for India/China (Fig. 2a) and for tropical adjacent regions (Fig. 2b) at 360 K potential temperature. It is evident in Fig. 2 (top) that at 360 K the CLaMS model simulates very strong horizontal tracer gradients between the tracer for India/China and that for the tropical adjacent regions at the edge of the anticyclone. High fractions of air from India/China up to 90% and low fractions below 10% from the tropical adjacent regions are found in the core of the Asian monsoon anticyclone at 360 K potential temperature. Highest fractions from the tropical adjacent regions of about 40% are found in a belt around the edge of the anticyclone. Towards the north this belt is separated by the subtropical jet from the northern lower stratosphere visible as a very sharp

gradient. To the south, air masses from the tropical adjacent regions do not show a strong gradient with air masses within the tropics and therefore are not separated by a strong transport barrier from the TTL at a level of potential temperature of 360 K (e.g., Ploeger et al., 2015; Santee et al., 2017).

Fig. 2c/d shows the longitude–theta cross sections at 25°N of the fraction from India/China (Fig. 2c) and from the tropical adjacent regions (Fig. 2d) on 18 August 2008. We would like to emphasise the horizontal transport of air masses with high contributions from India/China (40%–90%) from the eastern part of the anticyclone to both the western part and into the eddy over the western Pacific between ≈340 K and ≈380 K. Correspondingly, low fractions from the tropical adjacent regions (0%–30%) are simulated in these regions. The horizontal transport of air masses from the eastern to the western mode of the anticyclone indicated by the India/China tracer is consistent with simulations of CO using the Whole-Atmosphere Community Climate Model (WACCM4-SD) (Pan et al., 2016).

Fig. 2 shows the latitude–theta cross sections in the eastern mode of the anticyclone at 90°E (Fig. 2e/f) and in the western mode of the anticyclone at 30°E (Fig. 2g/h) on 18 August 2008. Below 360 K, high fractions of air from India/China up to 90% and low fractions from the tropical adjacent regions lower than 5% are found in the eastern mode of the anticyclone (Fig. 2e/f). In the western mode there is still a large contribution from the India/China tracer between 20% and 60% and lower fractions of about 10%–40% from the tropical adjacent regions (Fig. 2g/h inside the thick white line). Below the western mode, in the tropics below ≈330 K at around 10°N fractions from the tropical adjacent regions (in that case from Northern Africa) are up to 90% caused by local upward transport (Fig. 2h).

Further, a strong vertical gradient of the India/China tracer is found at about 360 K. This level is below the thermal tropopause, which is located at around 380 K (≈100 hPa) over the eastern mode of the Asian monsoon anticyclone at 90°E (Fig. 2e). At 90°E, there is a layer of young air masses with enhanced fractions from India/China (up to ≈20%) above the thermal tropopause up to about 420 K potential temperature. An obvious explanation for this model result would be that the tropopause over the Asian monsoon is not a strict vertical transport barrier and weak vertical cross-tropopause transport occurred, in contrast to the extra-tropics (air masses at the polar side of the subtropical jet) where the tropopause acts as a chemical transport boundary.

Below 360 K in the region with high values of the India/China tracer, the fractions from the tropical adjacent regions are below 10%, however above 360 K around the tropopause the fractions are much higher, up to about 30%, and up to about 15% around 420 K (see Fig. 2d/f). Thus, at 420 K the contributions of young air masses are about 20% from India/China and 15% from the tropical adjacent regions. From this result the question arises, 'how can air masses from the outer edge of the anticyclone be transported from 360 K into the lower stratosphere above?' A straight vertical cross-tropopause transport cannot be the explanation because inside the anticyclone the fraction from the tropical adjacent regions is much lower.

Note that in Fig. 2 the same data range is used for all colour bars for a better comparability between the horizontal and different vertical cross sections. Therefore some features in the horizontal cross section at 360 K are not too prominent, for example the thin filament at around 50°E between 40°N and 60°N in Fig. 2a (see next section Fig. 3).

### 3.1.2 Emission tracer for India/China vs MIPAS HCFC-22

To demonstrate that the spatial distribution of tracers of air mass origin found in the CLaMS model in the region of the Asian monsoon anticyclone is consistent with observations of chemical tracers, we analyse HCFC-22 measurements of the MIPAS instrument (Chirkov et al., 2016). As described in Sect. 1, HCFC-22 is emitted in locally restricted regions in continental eastern Asia, in particular in China, and in the Near East.

Fig. 3 and Fig. 4a show the horizontal distribution for the India/China tracer and of HCFC-22 measurements synoptically interpolated to 18 August 2008 12:00 UTC at 380 K potential temperature (for details see Sect. 2.4); also shown are the longitude–theta cross section at 25°N (Fig. 4b) as well as the latitude–theta cross sections at 90°E ± 10° (Fig. 4c) and at 30°E ± 10° (Fig. 4d). The contour line of 20% of the India/China tracer is marked on the cross sections for better comparison with the CLaMS results shown in Fig. 2.

The horizontal spatial distributions of the India/China tracer (Fig. 3) and HCFC-22 at 380 K (Fig. 4a) show good overall agreement. In particular the strong gradient at the northern flank of the Asian monsoon anticyclone is evident both in the model and in HCFC-22 observations. Enhanced HCFC-22 values up to 240 pptv compared to the stratospheric background (of around 180 to 200 pptv at 16 km altitude in JAS derived from MIPAS measurements (Chirkov et al., 2016)) are found in both the eastern and western mode of the anticyclone, in the smaller eddy at the northeastern flank of the anticyclone as well as in the thin filament at around 50°E between 40°N and 60°N. Further, as in the CLaMS model in the observations there is no sharp gradient at the southern flank of the anticyclone, which separates anticyclonic air from the surrounding tropics.

The vertical HCFC-22 distributions (Fig. 4c/d) within the eastern and western modes of the anticyclone are broadly consistent with the vertical distribution of the India/China tracer (see Fig. 2e/g). The highest mixing ratios of HCFC-22 are found within the anticyclone below the thermal tropopause. However, also at the top of the anticyclone above the tropopause enhanced HCFC-22 values are measured in agreement with the CLaMS tracer for India/China. Thus, measurements of HCFC-22 are consistent with our model result that young air masses from the region of the Asian monsoon are transported to the top of the anticyclone above the tropopause.

In addition, the vertical HCFC-22 distribution (Fig. 4d) for the western mode shows a very steep gradient in the upper troposphere between 350 K and 360 K in agreement with the spatial distribution of the India/China tracer (see Fig. 2g). Thus below 350–360 K smaller mixing ratios of HCFC-22 are measured than above, indicating that below the western mode of the anticyclone there exists no upward transport from boundary sources for HCFC-22. The enhanced values of HCFC-22 within the western mode and below the thermal tropopause along the longitude–theta cross sections at 25°N (Fig. 4b) confirm the horizontal westward transport within the Asian monsoon anticyclone as found for the India/China tracer in the CLaMS simulations (see Sect. 3.1.1 and Fig. 2c), which is consistent with CO simulations by Pan et al. (2016).

### 3.1.3 Impact of young air masses on the top of the AMA

In Sect. 3.1.1, it is shown that enhanced fractions of both tracers for India/China and tropical adjacent regions are found above the thermal tropopause at the top of the Asian monsoon anticyclone. Fig. 5 shows the horizontal distribution of the fraction of air originating in India/China (left) and in tropical adjacent regions (right) at different levels of potential temperature between 380 K and 460 K.

Young air masses (age < 6 months) from both India/China and tropical adjacent regions are found up to ≈460 K. It was shown earlier that the horizontal distribution of the India/China tracer is a good proxy for the location of the anticyclone (Vogel et al., 2015). The comparison between the horizontal distribution of the tropical adjacent regions and the India/China tracer strongly differs depending on the level of potential temperature from a nearly disjoint distribution at 360 K (see Fig. 2) to a more coincident distribution from 400 K to 460 K (see Fig. 5).

At 380 K, the highest fractions from tropical adjacent regions are found at the edge of the anticyclone, while at 400 K they are found within the anticyclone. Above 400 K both the India/China and the tropical adjacent region tracers show a similar horizontal distribution. We emphasise that at these levels of potential temperature the tracer distributions have the shape of rotating filaments in contrast to the more compact distribution at lower levels. The variation of the distribution of the tracer for the tropical adjacent regions with altitude is an indication that the upward transport of young air masses at the top of the anticyclone occurred more towards the edge and less inside the anticyclone itself.

### 3.2 Backward trajectory calculations

### 3.2.1 40-day backward trajectories at the top of the anticyclone and beyond

To analyse the transport pathways to the top of the anticyclone in more detail, 40-day backward trajectories are calculated starting in the western (20–50°N,0–70°E) and eastern (20–50°N,70–140°E) modes of the anticyclone. The trajectories are started at the position of the air parcels from the 3-dimensional CLaMS simulation at different levels of potential temperature ($\Theta = 380, 400, 420, 440$ K $\pm 0.25$ K) on 18 August 2018. Note that the air parcels in the 3-dimensional CLaMS simulation are distributed on an irregular grid. To take into account the distribution of the boundary emission tracer at the top of the Asian monsoon anticyclone, only air parcels are selected with contributions of young air masses (age < 6 months, Summer 08) larger than 70% (380 K), 50% (400 K), 20% (420 K), and 5% (440 K) (not all levels of potential temperature are presented here). The percentages are chosen in a way to obtain a number of trajectories (less than 30) that can be reasonably visualised. The results of the 40-day backward trajectories are similar at different levels of potential temperature; therefore we show a selection of trajectories to demonstrate the main transport pathway to the top of the Asian monsoon. A larger set of 20-day backward trajectories analysed statistically will be discussed below in Section 3.2.2.

Fig. 6 shows trajectories started in the eastern and western part of the Asian monsoon anticyclone around the thermal tropopause at 380 K on 18 August 2018. Air masses are uplifted to approximately 360 K very rapidly by various convective events occurring at different times and locations. Our 40-day backward trajectories show that preferred regions for fast uplift are continental Asia (mainly the region of the south slope of Himalayas and the Tibetan Plateau) and the western Pacific

(not shown here). A lower fraction of trajectories originates in the free troposphere. The trajectories in Fig. 6 demonstrating convection below 380 K are only a snapshot for 18 August 2018. There are several previous studies (e.g., Randel and Park, 2006; Park et al., 2007, 2009; Wright et al., 2011; Chen et al., 2012; Bergman et al., 2013; Fadnavis et al., 2014; Tissier and Legras, 2016) quantifying the contribution of different source regions to the composition of the Asian monsoon anticyclone during the course of the monsoon season (see discussion in Sect. 4). The backward trajectories demonstrate that above 360 K potential temperature the air masses circulate around the anticyclone in a large-scale upward spiral extending from northern Africa to the western Pacific. Here in the upward spiralling range, the vertical transport is much slower than in the convective range.

Fig. 7 shows trajectories started in the western and eastern parts of the Asian monsoon anticyclone above the thermal tropopause at 400 K and at 440 K. The slow upward transport up to $\approx 1$ to 1.5 K per day in a large-scale upward spiral is evident in both the western and eastern parts of the anticyclone. The 40-day backward trajectories demonstrate that at the top of the anticyclone the upward transport of air masses occurs along a large-scale upward spiral and therefore no straight vertical transport from the upper tropopause into the lower stratosphere takes place. The higher above the thermal tropopause the larger is the contribution of trajectories from outside the Asian monsoon anticyclone coming into the upward spiralling flow above 360 K. Trajectories at other levels of potential temperature (not shown) both in the western and in eastern parts of the anticyclone have a similar behaviour as shown in Figs. 6 and 7 and therefore confirm the presented results.

In general, trajectory calculations have limitations due to trajectory dispersion by errors through interpolation of the wind data to the position of the air parcel at a specific time. Over the timescales in question, mixing can also be relevant (e.g., McKenna et al., 2002b). These errors can accumulate depending on the trajectory length over the course of the simulation. However, the frequently employed trajectory length to study transport processes in the Asian monsoon region ranges from a couple of weeks to a few months (e.g., Chen et al., 2012; Bergman et al., 2013; Vogel et al., 2014; Garny and Randel, 2016; Müller et al., 2016; Li et al., 2017). In our trajectory analysis, the focus is to demonstrate the large-scale transport pathways of the air parcels at the top of the anticyclone, therefore small changes of the trajectory position will not affect our findings.

### 3.2.2 Global 20-day backward trajectories in the region of the Asian monsoon anticyclone

In the previous section, the transport pathways for a restricted number of trajectories in the region of the Asian monsoon anticyclone were discussed. Here, for a broader view 20-day backward trajectories between 0–160°E and 10–60°N are presented including the entire region of the Asian monsoon anticyclone. In this longitude-latitude region (0–160°E and 10–60°N), backward trajectories are calculated on a $1.0° \times 0.5°$ longitude-latitude-grid at 360 K, 380 K, 400 K, 420 K, and 440 K starting on 18 August 2008. Each point in Fig. 8 indicates the location of the start position of a 20-day backward trajectory colour-coded by the change in potential temperature ($\Delta\Theta$) during the last 20 days.

At 360 K, air parcels that experienced very strong upward transport by up to $\approx 60$ K within the last 20 days were found inside the western and eastern modes of the anticyclone, within the eddy over the Pacific, and within the tropics south of the anticyclone. The patterns of $\Delta\Theta$ at 360 K within the anticyclone and in the tropics are very patchy, reflecting that the strong

upward transport in this region is caused by single convective events.

Above 360 K, air parcels that experienced strong upward transport larger than 20–30 K within 20 days (corresponding to a mean value of 1–1.5 K per day) are largely found in the region of the anticyclone. This rate of upwelling is much slower
compared to convective upwelling shown at 360 K. Air parcels that experienced strong upward transport are mainly grouped in curved elongated filaments, reflecting a rotating movement of the air parcels at the top of the anticyclone. Often air parcels with strong $\Delta\Theta$ above 360 K are located more at the edge of the eastern and western modes of the anticyclone and at the edge of the eastward-migrating eddy at the eastern flank of the anticyclone. Thus the upward transport in the region of the anticyclone is not homogeneously distributed over the entire anticyclone at a certain day. In previous studies (e.g., Randel et al., 2010;
Ploeger et al., 2017) climatological mean values (over the monsoon season of several years) are presented which can not be used to analyse the inhomogeneity of the upward transport in the region of the anticyclone at a certain point in time. This inhomogeneity is consistent with results presented above in Sect. 3.2.1 demonstrating that for single selected trajectories the transport at the top of the Asian monsoon anticyclone is a slow upward transport of about 1–1.5 K per day in a large-scale spiral above the anticyclone caused by diabatic heating. In the backward trajectory calculations mixing processes are not included,
however the results of the trajectory calculations are consistent with patterns found in the 3-dimensional CLaMS simulation including mixing as discussed in Sect. 3.1.3, demonstrating that young air masses above 400 K are found at the edge of the anticyclone. Above 400 K, air masses in the tropics also experienced upward transport, but the vertical uplift is in general lower than 20 K within 20 days, (i.e. lower than 1 K per day).

In Appendix A, results of global 20-day backward trajectories demonstrate that during the monsoon season above 360 K an
uplift of air parcels of about 1–1.5 K per day occurred only in the region of the Asian monsoon anticyclone compared to the rest of the tropics. Even in the tropics the uplift is in general slower at these levels of potential temperature. Further the seasonal variability of this upwelling above the Asian monsoon anticyclone from monsoon-onset until post-monsoon 2008 is discussed in Appendix A.

## 3.3   Results from the three-pulse-approach

In the previous sections using CLaMS model simulations and MIPAS HCFC-22 measurements, we could show that the circulation of the Asian monsoon is effective in transporting very young air masses ($<$ 6 months) from the surface into the lower stratosphere up to $\approx$ 460 K. Here, we discuss subsequent transport pathways of air masses from the region of the Asian monsoon and Southeast Asia into the tropical pipe (middle stratosphere).

### 3.3.1   Transport within the tropical pipe

Fig. 9 shows latitude–theta cross sections at 90°E for the fraction from India/China from the start of the simulation on 1 May 2007 until 18 August 2008, which is a sum of the contributions of each of the three time pulses Summer 07, Winter 07/08, and Summer 08 each set for a time period of 6 months. A signal with enhanced fractions from India/China (up to 10%) is found at

around 550 K within the tropics which is from the Summer 07 pulse. This shows that air masses from boundary layer regions in India/China are transported into the middle stratosphere within the tropical pipe within a time period of one year. Fractions of air from the India/China tracer for the time pulse for Winter 07/08 are below 2.4 %, indicating that during winter in the absence of the Asian monsoon anticyclone the transport of boundary layer emissions from India/China into the stratosphere is insignificantly weak.

Fig. 10 shows the same images as Fig. 9, but for the fractions from tropical adjacent regions (TAR). Because of the Summer 08 pulse, high fractions from TAR are found at the edge of or outside the Asian monsoon anticyclone below 360 K. At altitudes up to ≈460 K, enhanced fractions are found over the region of the Asian monsoon, but also above the entire tropics. For the Summer 07 pulse, an enhanced signal from TAR (up to 25 %) is found at around 550 K within the tropics similar to that for India/China tracer. This model result shows that also air masses from outside the Asian monsoon anticyclone (Summer 07 pulse) are transported into the middle stratosphere within the tropical pipe within one year. The fraction of air from TAR (up to 25 %) is even larger than that from India/China (up to 10 %). In contrast to the India/China tracer, the TAR tracer shows that also emissions mainly from Southeast Asia and the tropical Pacific (for details see Sect. 3.3.2) released during winter (Winter 07/08 pulse) are transported into the lower stratosphere via the tropical pipe.

Fig. 11 shows that the upward spiralling transport above 360 K occurs in a field of radiative heating above the Asian monsoon, in particular above the western mode of the anticyclone. Thus, the combination of the anticyclonic flow and the uplift by radiative heating results in an upward spiralling transport like a 'Spiral Staircase'. Positive radiative heating rates are found in Era-Interim reanalyses on top of the Asian monsoon anticyclone as well as over the subtropics in the southern hemisphere (see Fig. 11). In these regions, positive radiative heating rates were also reported in a study by Park et al. (2007), using a free-running climate model. They argue that the positive radiative heating rates are the response to very low temperatures around the tropopause, which in turn are the response to convective heating near the equator (e.g., Gill, 1980; Highwood and Hoskins, 1998). It is known that the radiative heating rates in the tropical UTLS are different in current reanalysis models (e.g., Wright and Fueglistaler, 2013) and are most likely overestimated in ERA-Interim (e.g., Ploeger et al., 2012; Schoeberl et al., 2012). Therefore, the rates of diabatic heating in the upward spiralling range found in our study are most likely somewhat too high, however slow upward transport in the UTLS in the region of the Asian monsoon anticyclone associated with positive heating has been addressed previously (e.g., Park et al., 2007; Bergman et al., 2012; Garny and Randel, 2016; Ploeger et al., 2017). The focus of our study is to demonstrate that in the upward spiralling range (above 360 K) a slow upward transport is found over the region of entire anticyclone (west and east mode) with diabatic heating rates of up to 1–1.5 K inferred from ERA-interim. Our 40-day backward trajectory calculations (see Fig. 6 and 7) demonstrate that a diabatic heating above 360 K is found at both the western and eastern side of the anticyclone during August 2008.

Fig. 12 shows MIPAS HCFC-22 measurements at the same latitude–theta cross sections as Fig. 10 and Fig. 9. The transport pathway of HCFC-22 within the tropical pipe is evident. The transport of HCFC-22 is similar to that of the emission tracer for tropical adjacent regions, in contrast to the signal for the India/China tracer, which results from a combination of just two signals, one from the Summer 07 pulse and another from the Summer 08 pulse. It was shown in previous studies (Chirkov et al., 2016; Vogel et al., 2016) that HCFC-22 is enhanced in the region of the Asian monsoon anticyclone. There are also HCFC-

22 source regions outside the Asian monsoon region in continental eastern Asia (Fortems-Cheiney et al., 2013), therefore the upward transport of HCFC-22 in the troposphere can also occur during winter. In boreal winter, efficient transport into the stratosphere is found over the west Pacific and Maritime Continent caused by strong convection and in addition by the ascending branch of the Walker Circulation located over the Maritime continent (e.g., Bergman et al., 2012; Hosking et al.,

2012). In addition, stronger heating rates (vertical velocities) are found in the TTL during winter compared to summer (e.g., Bergman et al., 2012).

Figs. 9, 10, and 12 further show that in addition to the upward transport into the tropical pipe, transport of air masses out of the region of the Asian monsoon occurs into the northern lower stratosphere and to a much lower extent into the southern hemisphere. Details of the transport into the northern lower stratosphere are further discussed in Sect. 4.

### 3.3.2   Transport times and origin of air within the tropical pipe

On 18 August 2008, enhanced signals for the tracers from India/China and tropical adjacent regions from the Summer 07 pulse are found at around 550 K in the tropics between 30°S and 30°N (see Fig. 9 and Fig. 10). To analyse in more detail the transport times from the model boundary layer into the tropical pipe the contributions of the three different time pulses from different

emission tracers are calculated within the tropical pipe at 550 K potential temperature. We use the following approach: for each day between 1 May 2007 and 31 October 2008, a mean value for each emission tracer of all CLaMS air parcels is calculated between 30°S and 30°N at 550 K ($\pm$ 0.5 K). This is done for the sum of all three time pulses Summer 07, Winter 07/08, and Summer 08 and for each pulse individually.

Fig. 13 (top) shows the contribution of the three different time pulses ($\Omega_{S07}$, $\Omega_{W07}$, $\Omega_{S08}$) for the entire Earth's surface

between 1 October 2007 until the end of the simulation period (31 October 2008). The time period between 1 May 2007 and 1 October 2007 is not shown because here the contribution of all emission tracers is zero caused by the fact that the air masses need a certain transport time to reach the level of 550 K potential temperature within the tropical pipe. Air masses from the Summer 07 pulse reach this level first in November/December 2007; between January and May there is a steep increase of air masses from the Summer 07 pulse with a maximum in June 2008. Air masses from the Winter 07 pulse reach the 550 K level

of potential temperature during May 2008 and are increasing until the end of the simulation period. No contributions from the Summer 08 pulse are found at 550 K, since the transport times are too short to reach the tropical pipe within the simulation period.

The largest contributions to the Summer 07 pulse are from tropical Pacific, India/China, Southeast Asia, and northern Africa (Fig. 13 (top)). To the Winter 07 pulse contribute mainly the tropical Pacific and Southeast Asia. Here, the contribution of

India/China and northern Africa is insignificant, demonstrating that during the absence of the monsoon anticyclone in winter no vertical transport into the tropical pipe occurred.

Fig. 13 (bottom) shows that on the 550 K level, the contributions of all emission tracers (age < 18 months) to the tropical pipe is up to 55% by the end of October 2008. The contributions from Southeast Asia and the tropical Pacific are 16% and 15%, respectively. The contribution from India/China, which is mostly monsoon air, is much lower, around 6%. Smaller fractions are

estimated for the tropical Indian Ocean (4%), Northern Africa (3%), and South America (4%). The contributions of all other regions of the Earth's surface are each smaller than 3% and are summarised in the tracer referred to as residual surface (8%) shown in grey.

The black and the white lines in Fig. 13 (bottom) mark the contribution of the Summer 07 and Winter 07/08 pulses for each emission tracer, respectively (as shown in Fig. 13 (top)). For the tracer from India/China as well as for northern Africa, the black and the white lines are overlapping, implying that there is no contribution from India/China or northern Africa from the Winter 07/08 pulse in the tropical pipe (as shown above). Further, our findings show that air masses from India/China, thus mainly from the Asian monsoon anticyclone (see Vogel et al., 2015), contribute to a smaller fraction of the composition of air within the tropical pipe at 550 K; the major part is from Southeast Asia and the tropical Pacific. We would like to point out that the CLaMS emission tracer for Southeast Asia includes both the land masses of Southeast Asia and parts of the western Pacific including the Maritime Continent, which is known to also have intense deep convection during summer (e.g., Pan and Munchak, 2011).

## 4 Discussion

It is well known that the composition of the Asian monsoon anticyclone is strongly affected by convection over continental Asia (e.g. the south slope of the Himalayas and the Tibetan Plateau), Bay of Bengal, and the western Pacific, (e.g., Randel and Park, 2006; Park et al., 2007, 2009; Wright et al., 2011; Chen et al., 2012; Bergman et al., 2013; Fadnavis et al., 2014; Tissier and Legras, 2016). However there is debate about the contribution of different source regions to the composition of the Asian monsoon anticyclone. Further, there are differences in the conclusions in the literature about the contribution of different source regions depending on the reanalysis data used (e.g., Wright et al., 2011; Bergman et al., 2013). Findings by Vogel et al. (2015) show that there is a strong intraseasonal variability in the contributions of different boundary layer source regions to the composition of the Asian monsoon anticyclone during a particular monsoon season. We would like to emphasise that the trajectories presented in Fig. 6 demonstrating convection below 380 K are only a snapshot for 18 August 2018 with convection over the western Pacific and continental Asia mainly in the region of the south slope of the Himalayas and the Tibetan Plateau.

Here, in contrast to earlier studies, we focus on transport at the top of the anticyclone at altitudes greater than 380 K potential temperature ($\approx$100 hPa) reaching up to 460 K ($\approx$60 hPa). Further, going a step beyond previous studies (e.g., Garny and Randel, 2016; Ploeger et al., 2017), we relate the transport of air masses from inside the Asian monsoon anticyclone to air masses uplifted outside the anticyclone. Subsequently these air masses are jointly transported upwards to the top of the anticyclone at $\approx$460 K.

### 4.1 Transport pathways at the top of the anticyclone

It has been demonstrated by satellite observations that the Asian monsoon anticyclone affects the transport of sulphate aerosol and its precursor $SO_2$ injected by volcanic eruptions within the adjacent regions of the Asian monsoon (e.g., Bourassa et al.,

2012; Griessbach et al., 2013; Fairlie et al., 2014; Fromm et al., 2014; Wu et al., 2017). However, there is a debate about the exact transport mechanism. The observed transport into the stratosphere of air masses injected into the upper troposphere by volcanoes located at the edge of the Asian monsoon (e.g., Griessbach et al., 2013; Fairlie et al., 2014; Fromm et al., 2014; Wu et al., 2017) can be explained by the vertical transport concept of upward spiralling above the Asian monsoon, proposed here. It has been found in several satellite measurements (Griessbach et al., 2013; Fromm et al., 2014; Fairlie et al., 2014) that polluted air masses from the Nabro eruption in Eritrea, northeastern Africa, in June 2011 are transported around the anticyclone and at higher levels are found above the anticyclone, consistent with the CLaMS tracer for the tropical adjacent regions. Wu et al. (2017) show that the Asian monsoon anticyclone causes the transport of air masses emitted by the June 2009 eruption of the volcano Sarychev around the anticyclone southwards into the TTL, isolating aerosol-free air masses inside the anticyclone from aerosol-rich air outside.

Furthermore, the concept of the upward spiralling range transporting air masses from the adjacent regions of the Asian monsoon can also explain the impact of air masses uplifted by tropical cyclones in the western Pacific ocean on air masses observed by balloon measurements over Lhasa (China) (Li et al., 2017). Tropical cyclones rapidly loft marine air masses outside the Asian monsoon into the upper troposphere. The interplay between the location of the Asian monsoon antiyclone and the tropical cyclone controls the transport pathways of the air parcels from the boundary layer. Air parcels injected by a tropical cyclone into the outer edge of the anticyclone follow the flow around the anticyclone.

Moreover, the concept of the upward spiralling range including air mass transport from sources in the tropical adjacent regions could be a transport pathway to understand the formation of the Asian tropopause aerosol layer (ATAL) observed above the Asian monsoon region (e.g., Vernier et al., 2015, 2018; Brunamonti et al., 2018).

## 4.2 Contributions to the tropical pipe

It has been proposed that the Asian monsoon constitutes an effective transport pathway from the surface, through the Asian monsoon anticyclone, and deep into the tropical pipe based on satellite observations of hydrogen cyanide (HCN) (Randel et al., 2010). HCN is a tropospheric pollutant produced mainly by biomass burning with a strong sink on ocean surfaces. Therefore tropical ocean regions cannot be the source for HCN found in the tropical pipe. Ploeger et al. (2017) addressed this issue using CLaMS simulations marking air masses within the Asian monsoon anticyclone by a PV-gradient criterion (Ploeger et al., 2015). They find that the air mass fraction from the anticyclone correlates well with satellite measurements of HCN within the tropical pipe.

In our study, we found a similar behaviour for contributions of the India/China tracer within the tropical pipe as Randel et al. (2010) for HCN and Ploeger et al. (2017) for the simulated anticyclone air mass fraction. Ploeger et al. (2017) found maximum anticyclone air mass fractions of around 5% in the tropical pipe using 3-dimensional CLaMS simulations for 2010-2013. This is consistent with our simulations finding about a 6 % contributions of the India/China tracer within the tropical pipe at 550 K in 2008.

However, going beyond the results of Randel et al. (2010) and Ploeger et al. (2017), we show that the contributions from emissions from Southeast Asia and the tropical Pacific during summer are larger than the contribution from India/China within the tropical pipe. This demonstrates that the Asian monsoon anticyclone is a more effective transport pathway for the tropical adjacent regions than for air masses from inside the anticyclone itself (India/China). From the tropical adjacent regions air masses can be transported to the edge of the Asian monsoon anticyclone and then further into the tropical pipe.

### 4.3 Transport to the northern extratropical lower stratosphere

We would like to emphasise that the vertical transport of air masses at the top of the anticyclone is less effective than the horizontal isentropic transport by eastward eddy shedding events transporting air from the Asian monsoon anticyclone and from the tropical adjacent regions into the northern lower stratosphere or westward shedding transporting air into the TTL (e.g., Orbe et al., 2015; Garny and Randel, 2016; Pan et al., 2016; Vogel et al., 2016; Ploeger et al., 2017). The impact of the quasi-horizontal transport on the chemical composition of the northern lower stratosphere has already been discussed in detail in previous studies (e.g., Dethof et al., 1999; Ploeger et al., 2013, 2017; Garny and Randel, 2016; Müller et al., 2016; Vogel et al., 2014, 2016; Rolf et al., 2018).

Vogel et al. (2016) performed a CLaMS simulation for the year 2012 using similar tracers of air mass origin as in this work and found a flooding of the northern extratropical lower stratosphere with young air masses from the region of the Asian monsoon anticyclone. The transport of young air masses (age $< 6$ months) into the northern extratropical lower stratosphere was calculated to result in up to 44% young air at 360 K (up to 35% at 380 K) at the end of October 2012, with the highest contribution from India/China up to 15% (14%) (see Fig. 14 in Vogel et al. (2016)). Here, the same analysis is performed for the simulation for 2008 and a slightly higher impact on the northern extratropical lower stratosphere is found for the year 2008, up to 48% young air at 360 K (up to 41% at 380 K) at the end of October 2008 and up to 18% (16%) from India/China (see Appendix B). The differences between 2008 and 2012 are most likely caused by the interannual variability of the monsoon system. However, within the tropical pipe at 550 K, in 2008 the contributions from India/China are about 6%, demonstrating that the transport of air masses from the Asian monsoon anticyclone into the northern extratropical lower stratosphere during boreal summer and fall is more effective than the vertical transport into the tropical pipe during the course of one year. This is consistent with Ploeger et al. (2017), who found maximum anticyclone air mass fractions around 5% in the tropical pipe and 15% in the northern extratropical lower stratosphere using 3-dimensional CLaMS simulations for 2010-2013. In a study releasing trajectories within the Asian monsoon anticyclone, Garny and Randel (2016) found a similar value of 15% of trajectories released in the anticyclone that reach the northern extratropical lower stratosphere after 60 days (for 2006).

### 5 Conclusions

In this paper, the transport mechanisms of young air masses at the top of the Asian monsoon and subsequent transport into the tropical pipe are analysed using 3-dimensional CLaMS simulations as well as CLaMS backward trajectory calculations.

Tracers of air mass origin are introduced in the 3-dimensional CLaMS simulation to trace back the origin of young air masses (age < 6 months) found at the top of the Asian monsoon anticyclone and of air masses found within the tropical pipe (6 months < age < 18 months).

Young air masses with an age lower than 6 months are found at the top of the Asian monsoon anticyclone up to altitudes of about 460 K ($\approx 60$ hPa) by the end of August 2008. MIPAS satellite measurements of HCFC-22, a regional tracer emitted especially in continental eastern Asia and in the Near East, confirm our model result that young air masses from the region of the Asian monsoon are transported to the top of the anticyclone above the tropopause up to around 460 K.

A trajectory analysis in combination with the 3-dimensional CLaMS simulation shows that air masses from the lower troposphere are rapidly lofted by convective events up to approximately 360 K potential temperature (**convective range**) (see Fig. 14) consistent with previous results (e.g., Randel and Park, 2006; Park et al., 2007, 2009; Chen et al., 2012; Bergman et al., 2013; Fadnavis et al., 2014; Tissier and Legras, 2016; Li et al., 2017). Strong convection occurs both directly below the Asian monsoon anticyclone in continental Asia (India and China) and outside the anticyclone region over Southeast Asia and the Pacific Ocean (see Fig. 15).

Above 360 K and up to $\approx 460$ K, our findings from trajectory analyses demonstrate that air masses are uplifted in an anticlonic large-scale upward spiral around the anticyclone extending from northern Africa to the western Pacific by diabatic heating of $\approx 1$–1.5 K per day (**upward spiralling range**) (see Figs. 14 and 15). A straight homogeneous vertical cross-tropopause transport is not found in our backward trajectories, rather a very inhomogeneous horizontal distribution of tracers of air mass origin is found within and at the edge of the anticyclone, resulting in filamentary structures. Within the upward spiralling range strong horizontal transport of air masses from the eastern mode to the western mode of the anticyclone occurs. The thermal tropopause is located within this upward spiralling range. Thus, air masses in the upward spiralling range are uplifted by diabatic heating across the (lapse rate) tropopause, which does not act as a transport barrier against this diabatic vertical transport process. This transport across the tropopause is consistent with previous studies (e.g., Bergman et al., 2012; Garny and Randel, 2016; Ploeger et al., 2017). Diabatic heating rates within the tropical transition layer have strong seasonal variability. Diabatic heating rates of up to $\approx 1$–1.5 K per day are in particular found between July and October 2008 over the region of the Asian monsoon anticyclone. In other regions within the tropical transition layer the heating rates are in general smaller ($\approx 0.5$–1.0 K per day).

Above 380 K, within the upward spiralling range above the anticyclone, young air masses from along the edge of the anticyclone originating in the tropical adjacent regions are mixed with air masses from inside the anticyclone mainly originating in India/China (see Fig. 15). Therefore, a significant fraction of air masses from the tropical adjacent regions is found within a widespread area around the anticyclone and above caused by the large-scale anticyclonic flow in this region, acting as a large-scale stirrer. This transport pattern up to 460 K is consistent with previous results focused on lower levels of potential temperature up to $\approx 400$ K (Vogel et al., 2014, 2016; Li et al., 2017).

Further between 420 K and 440 K, highest contributions from young air masses are found around the edge of the anticyclone, indicating a spatially strongly inhomogeneous ascent in the monsoon with strongest ascent at the edge. The higher in the upward spiralling range the more air masses from the stratospheric background are mixed with the young air masses transported

upwards within this upward spiralling range (see Fig. 15). Thus in this paper, we answer the question of what are the transport pathways of young air masses at the top of the Asian monsoon into the stratosphere by the concept of the upward spiralling range.

To answer our second question of how boundary layer source regions in Asia affect the composition of the middle strato-
sphere within the tropical pipe at 550 K, the transport times from the Earth's surface up to this level of potential temperature need to be taken into account. In a two-monsoon-season simulation tracing back air masses that are released at the Earth's surface since 1 May 2007 (age < 18 months) only air masses older than 6 months are found within the tropical pipe at 550 K. Consequently, fresh emissions from the Asian monsoon season 2008 do not contribute to the distribution within the tropical pipe at 550 K before October 2008. Air masses released during the monsoon season 2007 and during Winter 2007/2008 con-
tribute up to 55% to the signal within the tropical pipe at 550 K in October 2008. Air masses from India/China, mainly from inside the Asian monsoon anticyclone, contribute a minor fraction (6%) to the composition of air within the tropical pipe at 550 K; the major fraction is from Southeast Asia (16%) and the tropical Pacific (15%).

In summary, the transport of air masses from the region of the Asian monsoon into the tropical pipe occurs in three distinct steps: first, very fast uplift within the convective range up to $\approx 360$ K within the Asian monsoon anticyclone and outside
in the tropical adjacent regions (within a few days); second, uplift above 360 K up to $\approx 460$ K within the upward spiralling range (within a few months); and third, above 460 K transport within the tropical pipe associated with the large-scale Brewer-Dobson circulation (within $\sim$ one year). Furthermore, we emphasise that in addition to air masses from the Asian monsoon region, a substantial percentage of air masses from the tropical adjacent regions (Southeast Asia/tropical Pacific/northern Africa/northwestern Pacific) is transported by this pathway into the tropical pipe.

*Data availability.*  The complete MIPAS data are available at http://www.imk-asf.kit.edu/english/308.php. The CLaMS model data may be requested from Bärbel Vogel (b.vogel@fz-juelich.de).

**Appendix A:  Intraseasonal variability of upwelling above Asian monsoon anticyclone**

Global 20-day backward trajectories are calculated at 360 K, 380 K, 400 K, 420 K, 440 K potential temperature on a $2.5° \times 2.0°$ longitude-latitude-grid from 1 June until 31 October 2008 covering the time period from monsoon-onset until post-monsoon
to discuss the intraseasonal variability of upward transport at the top of the Asian monsoon anticyclone. Some chosen days (1 June, 1 July, 1 August, 1 September, and 1 October 2008) are shown in Fig. A1. During the monsoon season in July, August and September an upward transport of about 1–1.5 K per day is mainly found in the region of the Asian monsoon anticyclone, in contrast to monsoon-onset in June. In June, an upward transport of about 1–1.5 K per day is found in the entire tropics, however the patterns are very inhomogeneous; in particular maximum values are found in the region of the northern subtropical jet
stream.

## Appendix B: Impact on the northern extratropical lower stratosphere

The accumulation of young air masses (age $<$ 6 months) from the region of the Asian monsoon since 1 May 2008 in the northern extratropical lower stratosphere is calculated using a climatological isentropic barrier derived by Kunz et al. (2015) that separates tropical tropospheric air from extratropical stratospheric air depending on potential temperature and season. To calculate the percentages of different emission tracers within the extratropical lower stratosphere at 360 K and at 380 K, the mean value for each emission tracer of all CLaMS air parcels is calculated for PV values larger than those of the transport barrier (5.5 PVU at 360 K±0.5 K and 7.2 PVU at 380 K±0.5 K) and poleward of 30°N for each day between 1 May and 31 October 2008. Fig. A2 shows that at the end of October 2008, the contributions of young air masses to the composition of northern extratropical lower stratosphere is up to 48% at 360 K (up to 41% at 380 K not shown here). The contribution from India/China is up to 18% (16%) and from tropical Pacific and Southeast Asia up to 18% (16%).

*Competing interests.* The authors declare that they have no conflict of interest.

*Acknowledgements.* The authors sincerely thank William J. Randel (NCAR, Boulder), and Michael Volk (University Wuppertal) for very helpful discussions. We gratefully acknowledge the European Centre of Medium-Range Weather Forecasts (ECMWF) for providing the ERA-Interim reanalysis data. Our activities were partly funded by the European Community's Seventh Framework Programme (FP7/2007-2013) under the project StratoClim (grant agreement no. 603557) and by the German Science Foundation (Deutsche Forschungsgemeinschaft, DFG) under the DFG project AMOS (HALO-SPP 1294/VO 1276/5-1). The authors would also like to thank Michelle Santee (JPL, Pasadena) and two other anonymous reviewers for their very helpful comments.

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

**Table 1.** Latitude and longitude range of artificial boundary layer sources in the CLaMS model, also referred to as "emission tracers". The geographical position of each emission tracer is shown in Fig. 1 (adapted from Vogel et al. (2015, 2016))

| Emission tracer ($\Omega_i$) | Latitude | Longitude |
|---|---|---|
| Northern India (NIN) | 20–40° N | 55–90° E |
| Southern India (SIN) | 0–20° N | 55–90° E |
| Eastern China (ECH) | 20–40° N | 90–125° E |
| Southeast Asia (SEA) | 12° S–20° N | 90–155° E |
| Northwestern Pacific (NWP) | 20–40° N | 125–180° E |
| Siberia (SIB) | 40–75° N | 55–180° E |
| Europe (EUR) | 45–75° N | 20° W–55° E |
| Mediterranean (MED) | 35–45° N | 20° W–55° E |
| Northern Africa (NAF) | 0–35° N | 20° W–55° E |
| Southern Africa (SAF) | 36° S–0° N | 7–42° E |
| Madagascar (MDG) | 27–12° S | 42–52° E |
| Australia (AUS) | 40–12° S | 110–155° E |
| North America (NAM) | 15–75° N | 160–50° W |
| South America (SAM) | 55° S–15° N | 80–35° W |
| Tropical Pacific Ocean (TPO) | 20° S–20° N | see Fig. 1 |
| Tropical Atlantic Ocean (TAO) | 20° S–20° N | see Fig. 1 |
| Tropical Indian Ocean (TIO) | 20° S–20° N | see Fig. 1 |
| Background | remaining surface | see Fig. 1 |
| ALL ($\Omega = \sum_{i=1}^{n} \Omega_i$) | entire surface | |
| India/China | NIN+SIN+ECH | |
| tropical adjacent regions (TAR) | SEA+TPO+NAF+NWP | |

—————————————————————-

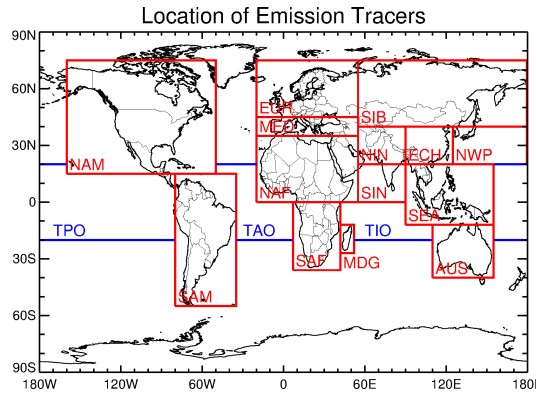

**Figure 1.** Global geographical location of artificial boundary layer source regions in the CLaMS model, also referred to as 'emission tracers' adapted from Vogel et al. (2015). The latitude and longitude range for each emission tracer is listed in Table 1.

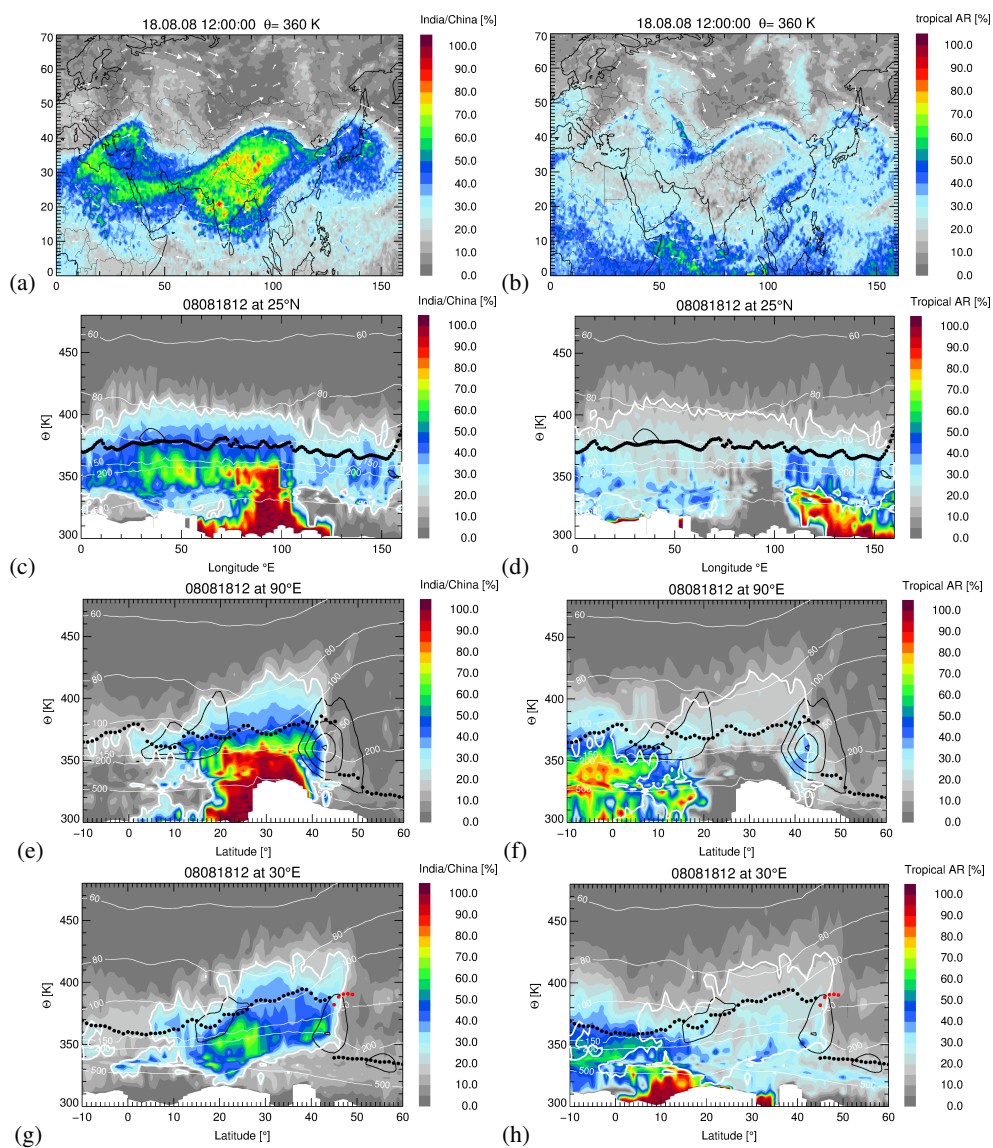

**Figure 2.** Horizontal distribution of the fraction of air originating in India/China (left) and in tropical adjacent regions (right) at 360 K potential temperature, longitude–theta cross sections at 25°N and latitude–theta cross sections at 90°E (eastern part of the anticyclone) and 30°E (western part of the anticyclone) on 18 August 2008. The horizontal winds are indicated by white arrows (maps) or by black thin lines (cross sections) (shown are 30, 40, 50, and 60 m/s). The primary thermal tropopause is marked by black dots, the secondary thermal tropopause by red dots, and the levels of pressure are marked by thin white lines. The contour line of 20% of the India/China tracer is shown by thick white lines (cross sections).

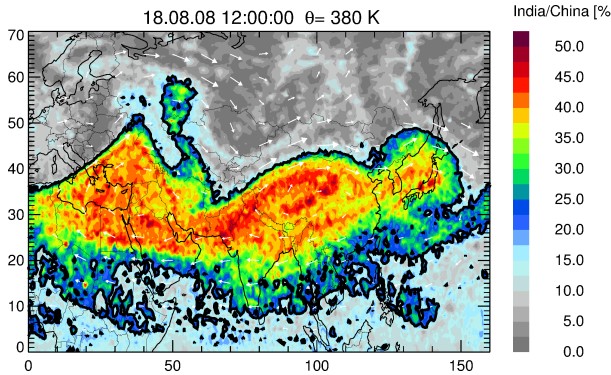

**Figure 3.** Horizontal distribution of the fraction of air originating in India/China at 380 K potential temperature. The contour line of 20% of the India/China tracer is shown by thick black line.

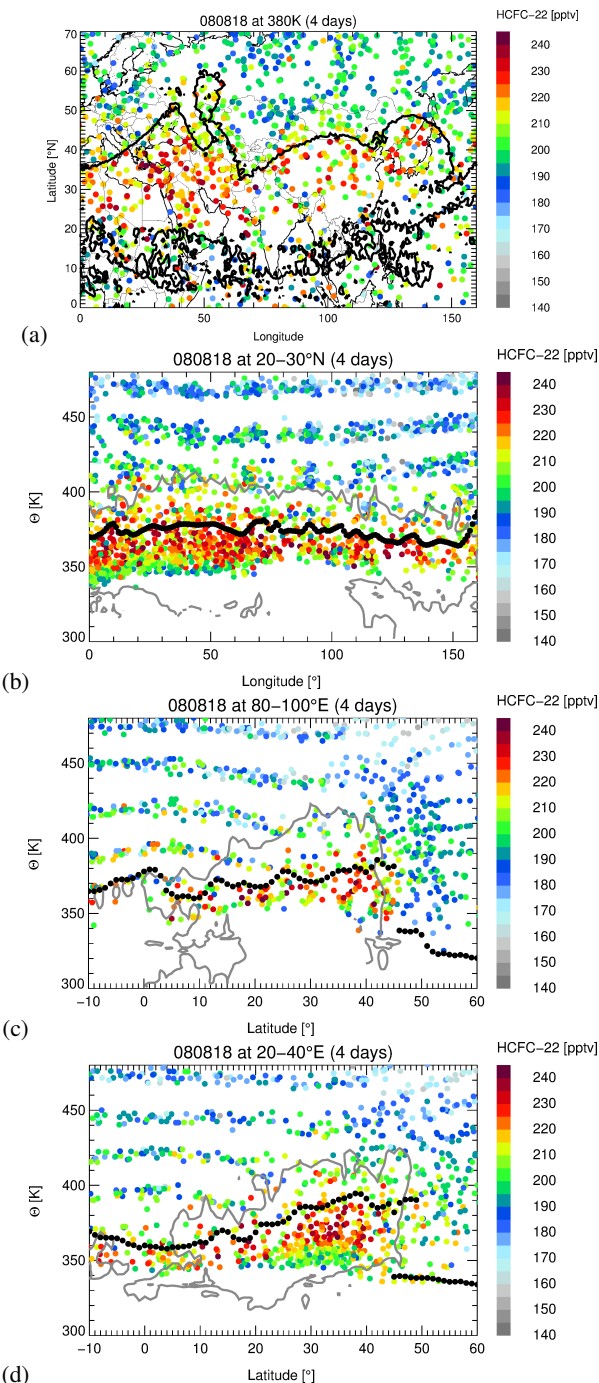

**Figure 4.** Horizontal distribution of MIPAS HCFC-22 measurements at 380 K potential temperature (a). The MIPAS measurements are synoptically interpolated within 4 days (for details see Sect. 2.4). Longitude–theta cross section at 25°N (b) is shown as well as latitude–theta cross sections at 90°E (eastern part of the anticyclone) (c) and at 30°E (western part of the anticyclone) (d) on 18 August 2008. The contour line of 20% of the India/China tracer is shown by thick black (maps) or grey (cross sections) lines as shown in Figs. 3 and 2. The thermal tropopause is marked by black dots.

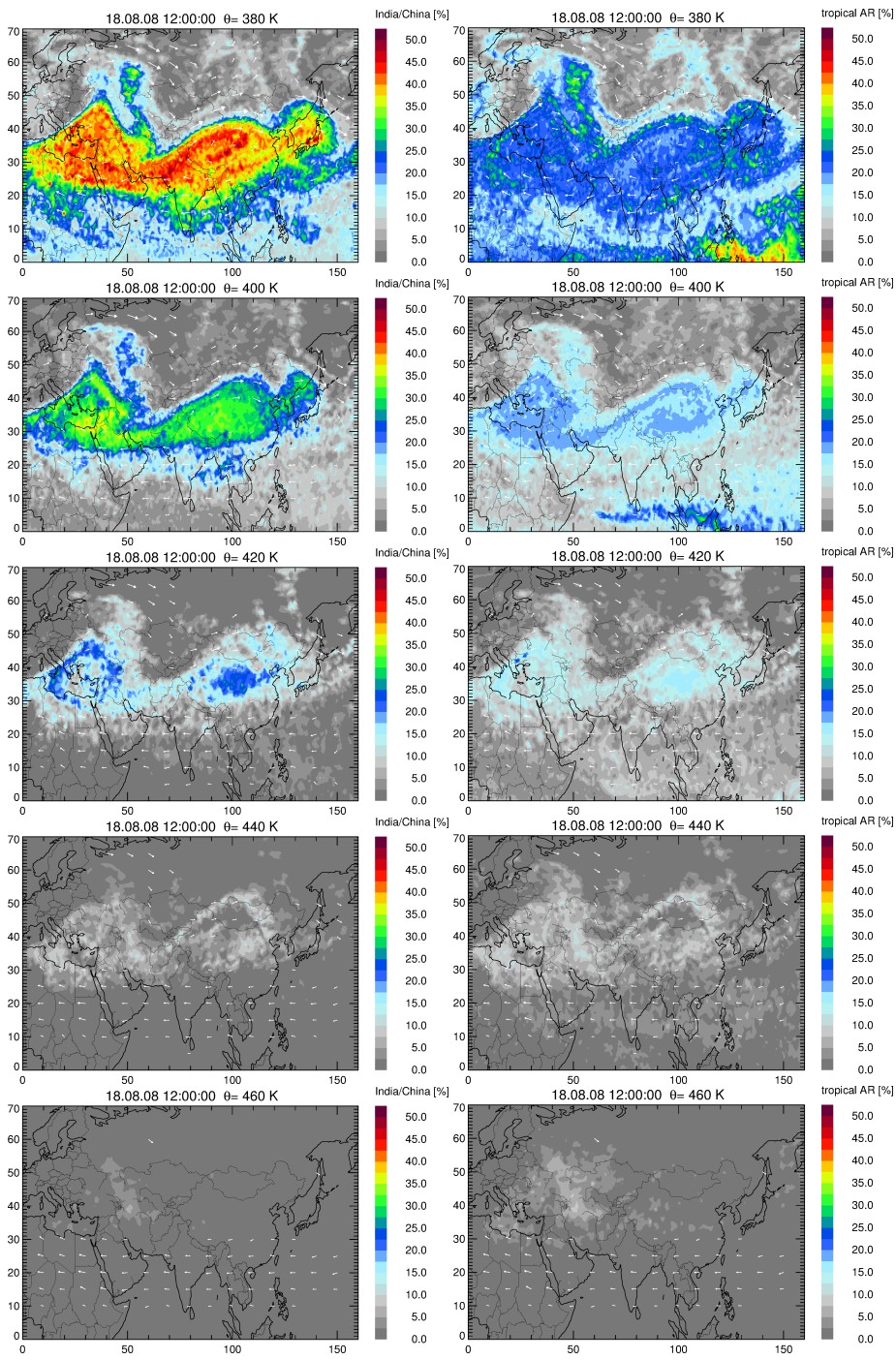

**Figure 5.** Horizontal distribution of the fraction of air originating in India/China (left) and in tropical adjacent regions (right) at 380 K, 400 K, 420 K, 440 K, and 460 K potential temperature.

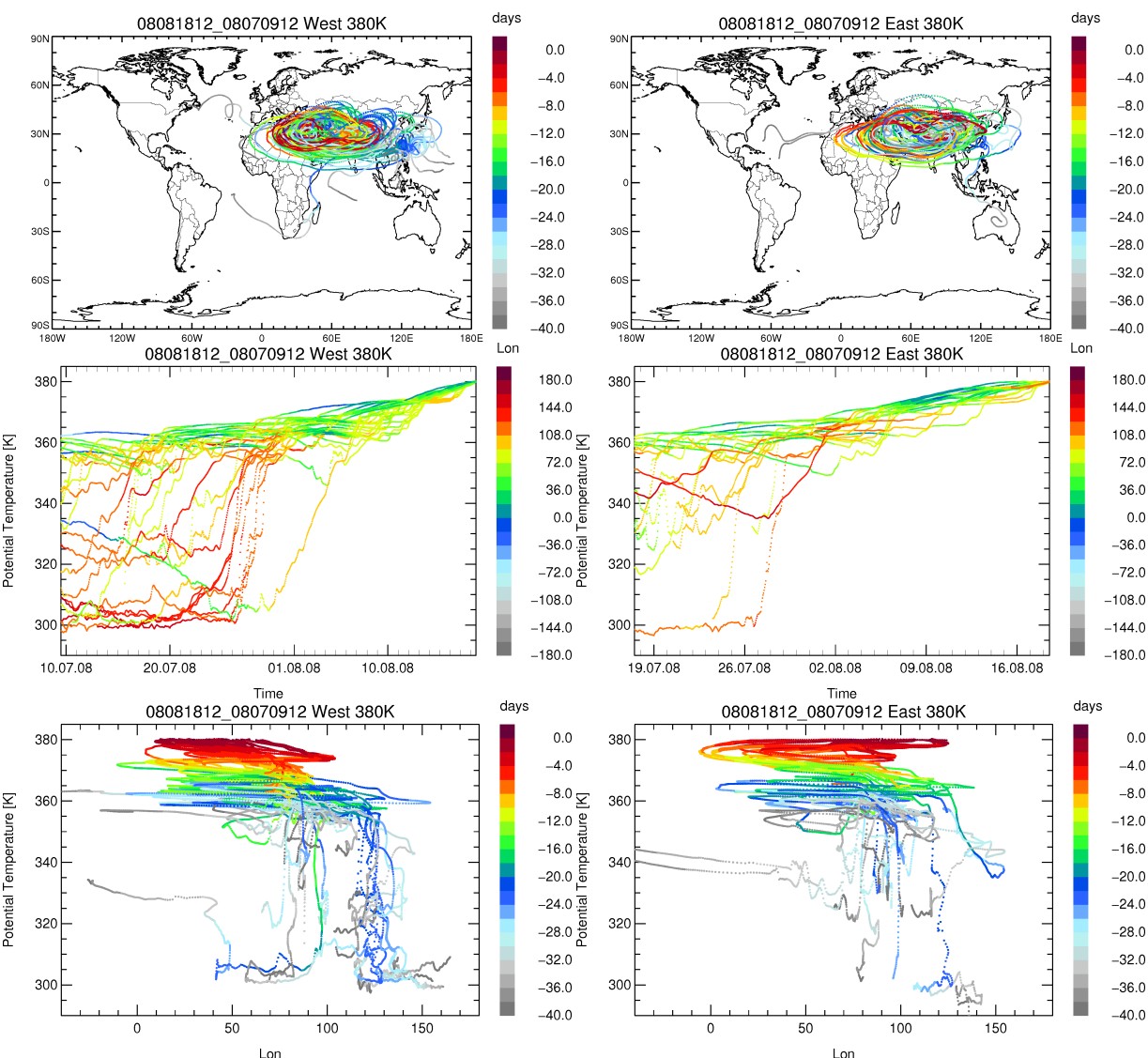

**Figure 6.** Different 40-day backward trajectories started at 380 K in the western (left) and eastern (right) mode of the Asian monsoon anticyclone are shown colour-coded by days back from 18 August 2008 (top). Further, potential temperature versus time (in UTC) along 40-day backward trajectories colour-coded by longitude (middle) and potential temperature versus longitude colour-coded by days back from 18 August 2008 (bottom) are shown. The trajectory positions are plotted every hour (coloured dots). Large distances between successive positions indicate rapid uplift.

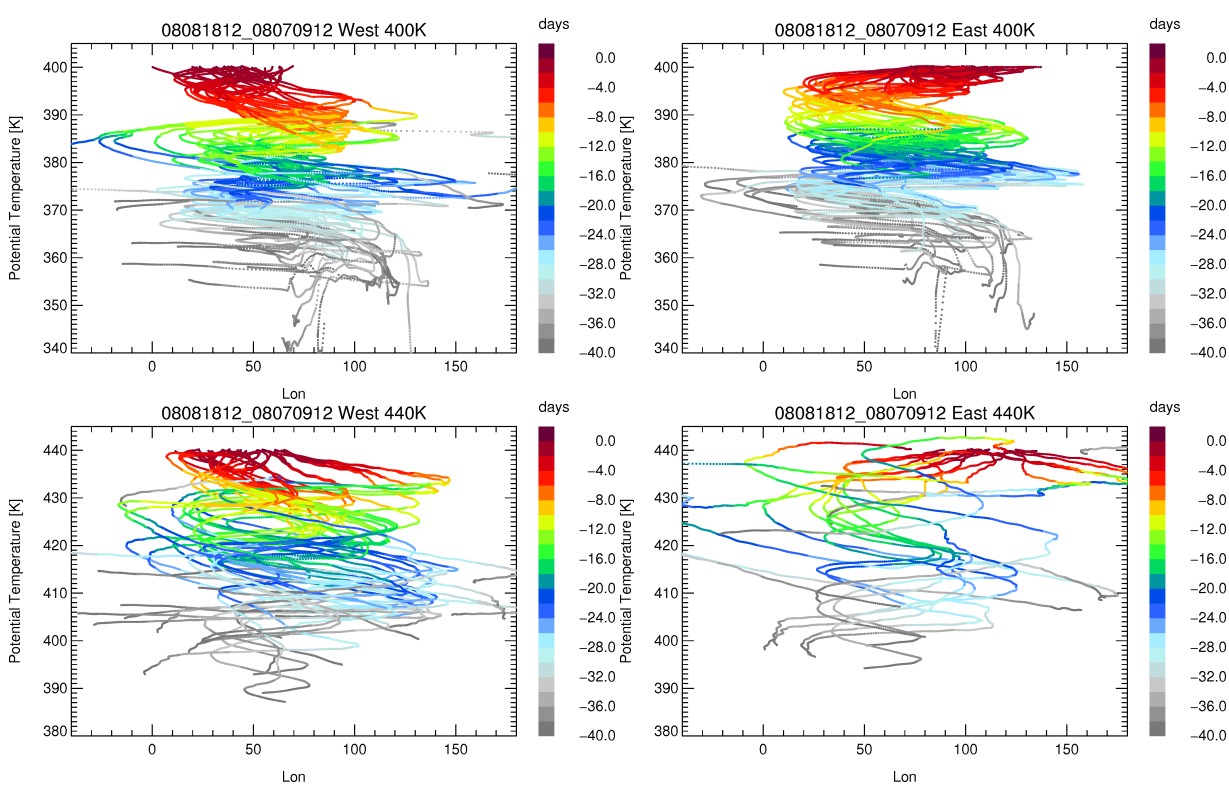

**Figure 7.** Potential temperature of different 40-day backward trajectories started at 400 K (top) and 440 K (bottom) in the western (left) and eastern (right) mode of the Asian monsoon anticyclone are shown versus longitude colour-coded by days back from 18 August 2008

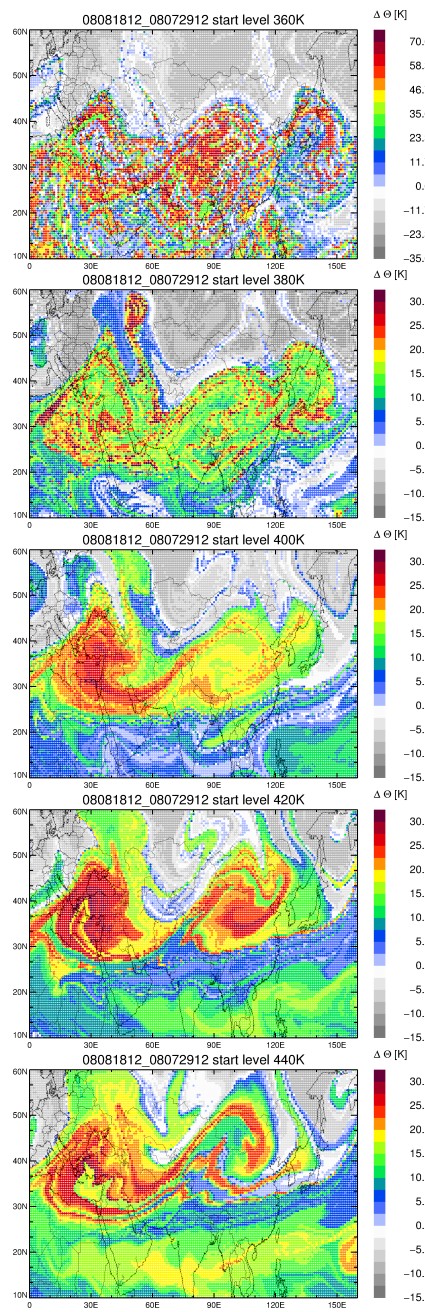

**Figure 8.** The change in potential temperature ($\Delta\Theta$) along 20-day backward trajectories initialised on 18 August 2008 is shown for different levels of potential temperature (360 K, 380 K, 400 K, 420 K, and 440 K). Note that the range of the colour bar in the 1st panel is much larger than in the other panels. At the lower potential temperature levels (360 K, 380 K), some 20-day backward trajectories exist that reach the model boundary layer within a time period shorter than 20 days (in cases of very strong uplift by convection). For these trajectories $\Delta\Theta$ is shown for the shorter time period.

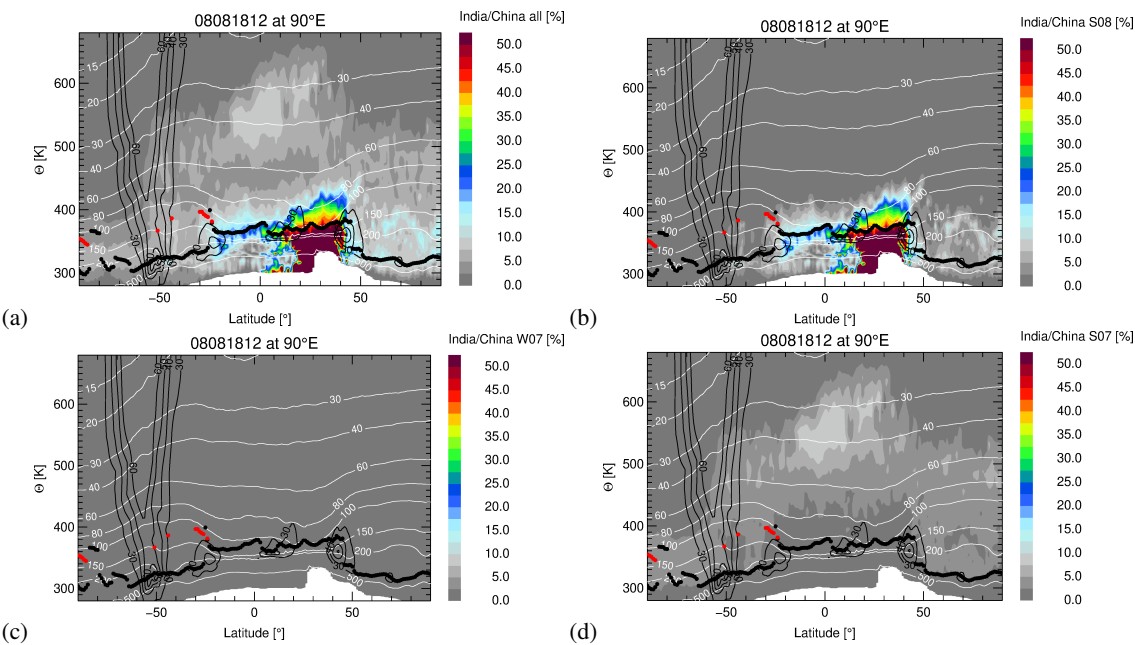

**Figure 9.** Latitude–theta cross sections at 90°E for the fraction of the India/China tracer for the simulation period (1 May 2007 - 18 August 2008 labeled as 'all') (a), for the Summer 08 (S08) pulse (b), for the Winter 07/08 (W07) pulse (c), and for the Summer 07 (S07) pulse (d) on 18 August 2008. The thermal tropopause (primary in black dots and secondary in red dots) and absolute horizontal winds (black lines for 30, 40, 50, and 60 m/s) are shown. The levels of pressure are marked by thin white lines. Note that the maximum value for the Winter 07/08 (W07) pulse (c) is 2.4%.

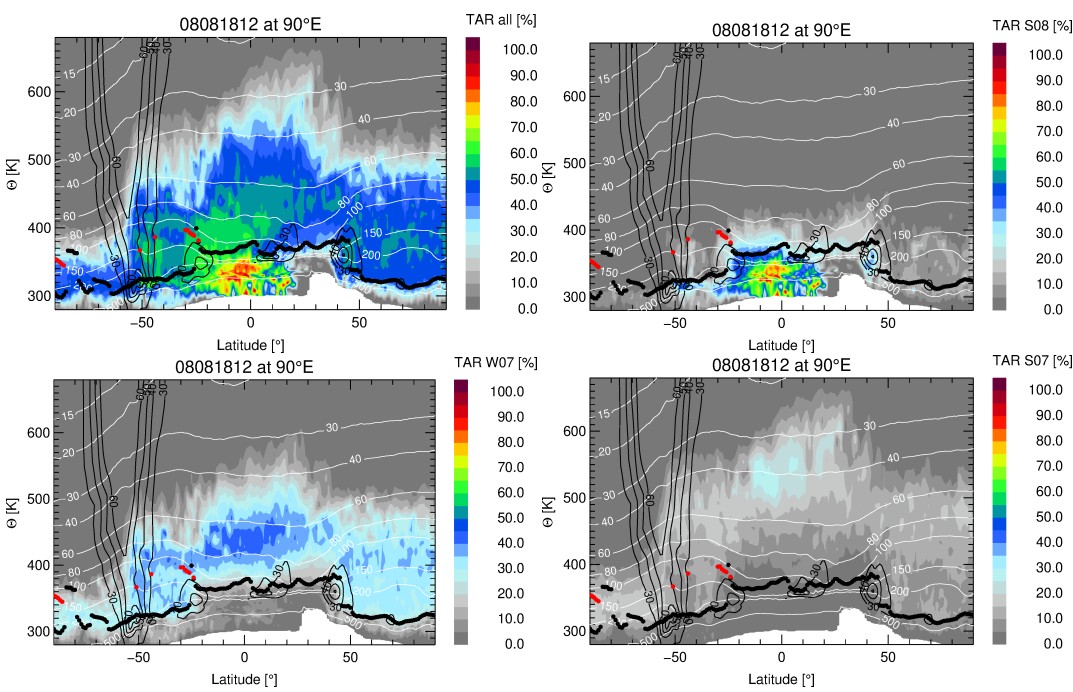

**Figure 10.** As Fig. 9 but for the fraction of the tracer for tropical adjacent regions (TAR)

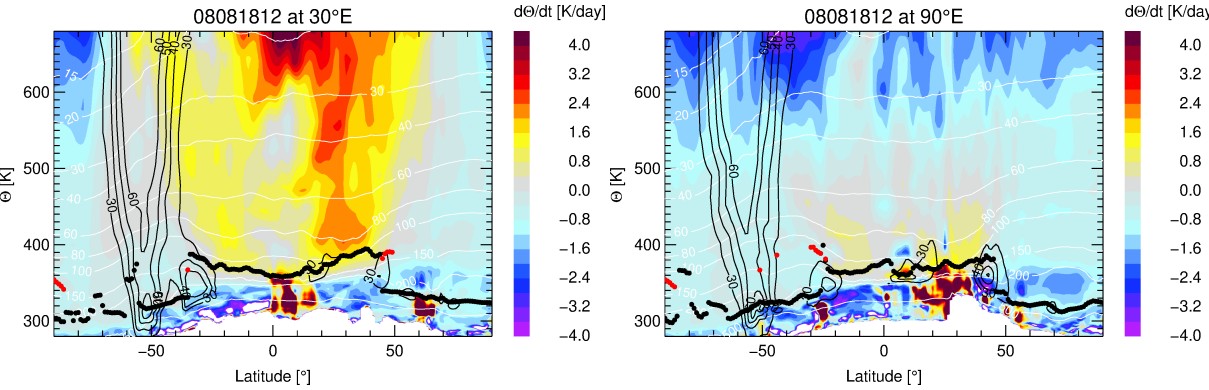

**Figure 11.** Latitude–theta cross section of dΘ/dt showing the radiative heating above the Asian monsoon anticyclone for the western (30°E) and eastern mode (90°E) of the anticyclone on 18 August 2008. The thermal tropopause (primary in black dots and secondary in red dots) and absolute horizontal winds (black lines for 30, 40, 50, and 60 m/s) are shown. The pressure levels are marked by thin white lines.

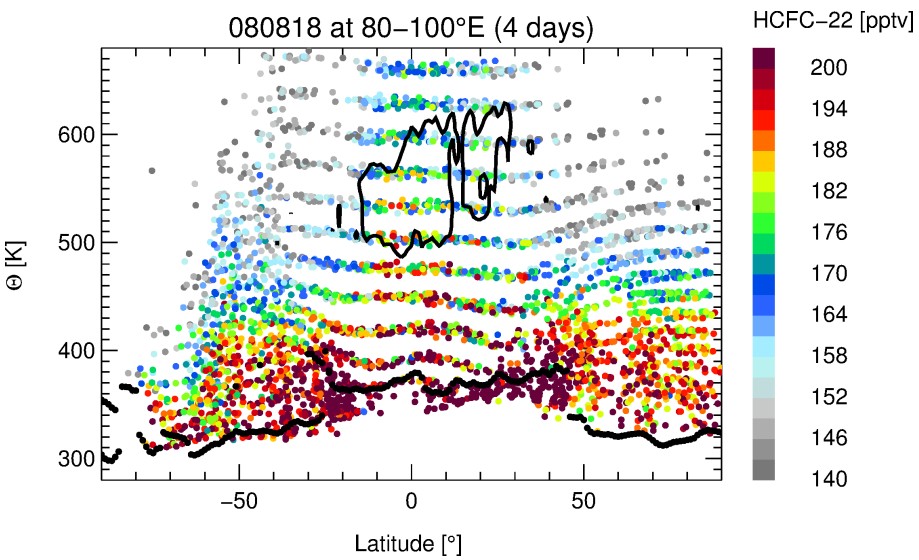

**Figure 12.** Latitude–theta cross sections of MIPAS HCFC-22 measurements crossing the eastern (80°E-100°E) mode of the anticyclone up to 680 K potential temperature on 18 August 2008. The thermal tropopause is indicated by black dots. The contour line of 6% of the India/China tracer for the Summer 07 pulse is shown by thick black lines as shown in Fig. 9. To highlight the MIPAS HCFC-22 signal within the tropical pipe the plot range of the data is up to 200 pptv. Note that the maximum values of MIPAS HCFC-22 in the troposphere are higher than 200 pptv.

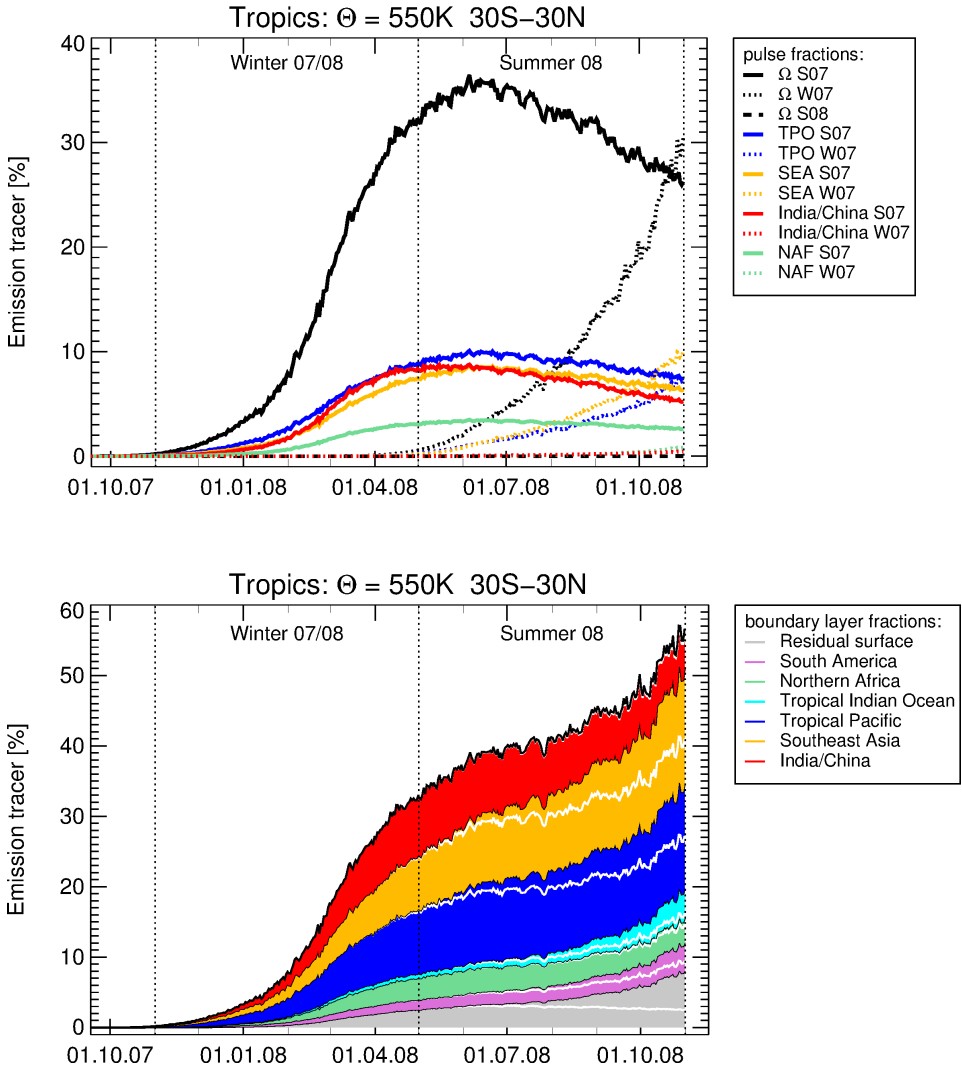

**Figure 13.** Top: The contribution of the three different time pulses S07, W07, and S08 (each set for a time period of 6 months marked by the vertical dotted lines) for the entire Earth's surface ($\Omega_{S07}$, $\Omega_{W07}$, $\Omega_{S08}$) to the tropical pipe between 30°S and 30°N at 550 K potential temperature from 1 October 2007 until the end of the simulation period (31 October 2008) (top, black lines). Emission tracers released during S08 have no contribution to the tropical pipe at 550 K by October 2008. The contribution of several tracers of air mass origin to the individual pulses are shown in colours.

Bottom: Contribution of different emission tracers for the tropical Pacific Ocean, Southeast Asia, India/China, South America, Northern Africa, tropical Indian Ocean, and the residual surface (all other regions of the Earth's surface) to the tropical pipe between 30°S and 30°N at 550 K potential temperature from 1 October 2007 until the end of the simulation period. The contribution of the time pulses for the different emission tracers are marked by thin white lines for S07 and by thin black lines for W07.

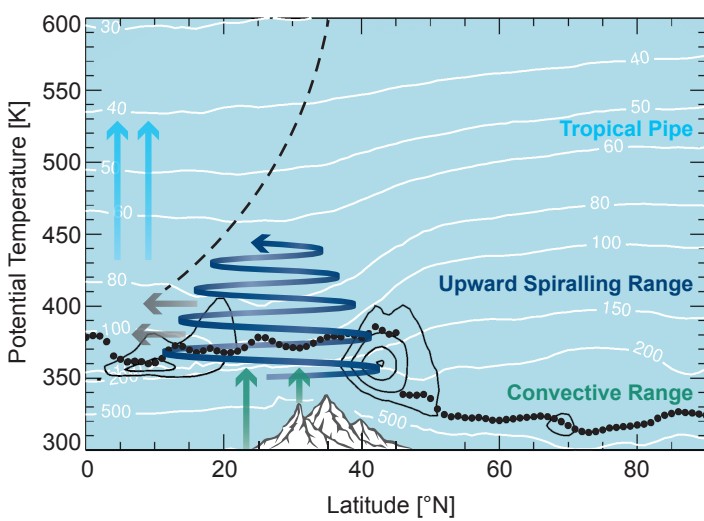

**Figure 14.** Latitude-theta cross section at about 90°E: The transport of air masses from the region of the Asian monsoon into the tropical pipe occurs in three distinct steps: first, very fast uplift within the convective range up to ≈ 360 K within the Asian monsoon anticyclone and outside in the tropical adjacent regions (within a few days); second, uplift above 360 K (up to ≈ 460 K) within the upward spiralling range (within a few months), and third, transport within the tropical pipe to altitudes higher than 460 K associated with the large-scale Brewer-Dobson circulation (within ∼ one year). The thermal tropopause (black dots) and horizontal winds (black lines) are shown. The horizontal winds mark the edge of the anticyclone at its northern and southern flank. The levels of pressure are marked by thin white lines. Large grey arrows indicate isentropic transport from the Asian monsoon anticyclone into the tropics. The dashed line marks the tropical pipe, which largely isolates tropical air masses from isentropic mixing with mid-latitude air (e.g., Plumb, 1996; Volk et al., 1996).

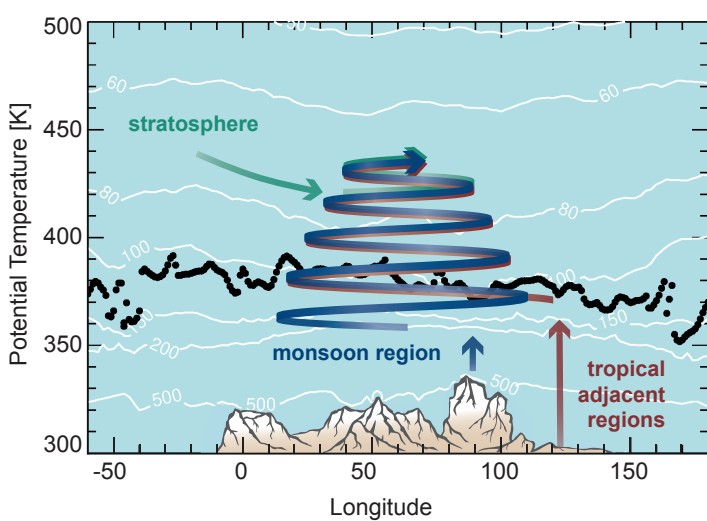

**Figure 15.** Longitude-theta cross section at about 30°N: At the top of the Asian monsoon anticyclone (above ≈360 K) air masses circulate around the anticyclone in a large-scale upward spiral extending from northern Africa to the western Pacific. In the upward spiralling range air masses from inside the Asian monsoon anticyclone (shown in blue) are mixed with air masses convectively uplifted outside the core of the Asian monsoon anticyclone in the tropical adjacent regions e.g. uplifted by tropical cyclones in the western Pacific ocean (shown in red). The higher above the thermal tropopause the larger is the contribution of air masses from outside the Asian monsoon anticyclone from the stratospheric background coming into the upward spiralling flow (shown in green). The levels of pressure are marked by thin white lines and the thermal tropopause is shown by black dots.

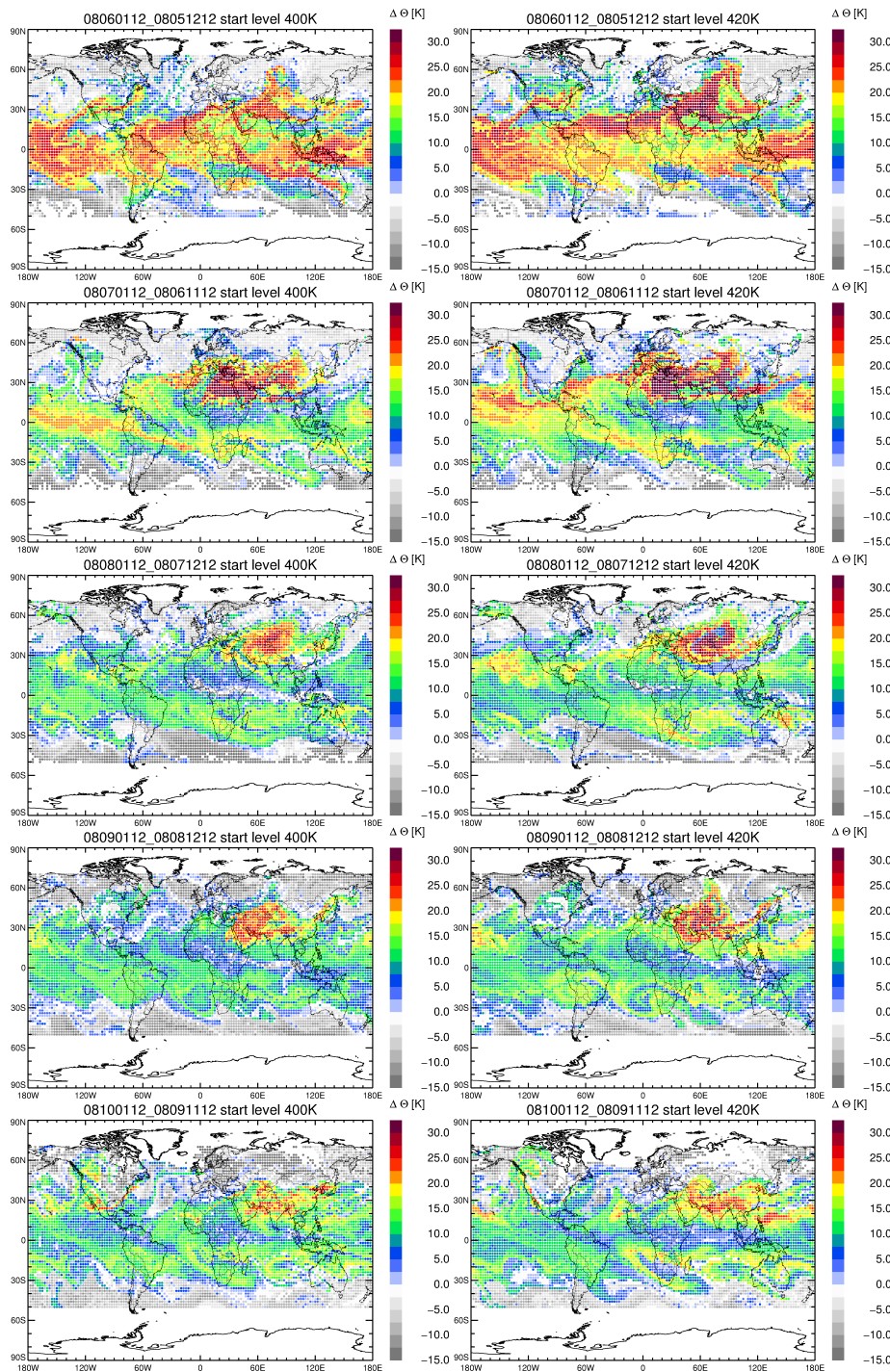

**Figure A1.** The change in potential temperature (ΔΘ) along 20-day backward trajectories are shown for different levels of potential temperature (400 K and 420 K) for different days from monsoon-onset until post-monsoon (1st June, 1st July, 1st August, 1st September, and 1st October 2008).

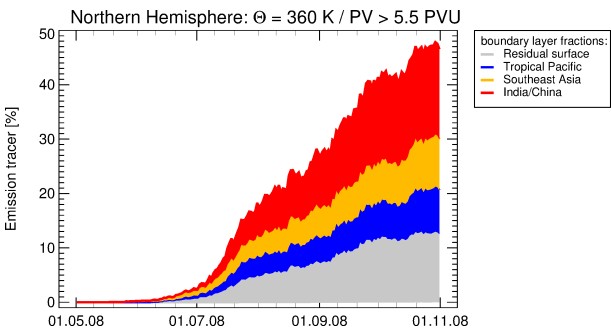

**Figure A2.** Contribution of different emission tracers from India/China, southeast Asia, the tropical Pacific Ocean, and residual surface to the northern lower stratosphere at 360 K from May to October 2008.