# Peer review of "Lagrangian simulations of the transport of young air masses to the top of the Asian monsoon anticyclone and into the tropical pipe"

_Atmospheric Chemistry and Physics, 2018_

## Referee Comment (RC1) · Anonymous Referee #1 · 14 Aug 2018

This manuscript uses back trajectories and full 3D CLaMS simulations in conjunction with MIPAS HCFC-22 measurements to elucidate the transport pathway of air masses emitted in defined boundary layer regions through the Asian summer monsoon anticyclone and into the tropical pipe. The modeling tools and measurements are well suited to the investigation, the analysis is generally well thought out and well executed, and the findings will certainly be of interest to the journal readership. I do, however, have a number of substantive comments that I would like to see addressed before the paper is accepted for publication in ACP.

Specific substantive comments and questions:

[Figure]

Sections 2.2 and 3.2.1: 40 days seems like a very long period for trajectory calculations. I realize that CLaMS 40-day trajectories have been published previously, but nevertheless I think that a sentence or two on how much error has accumulated over the course of such long trajectory calculations would be appropriate, either in Section 2.2 or in Section 3.2.1.

Section 2.4: I miss in the description of the MIPAS HCFC-22 any information about the accuracy, precision, or horizontal or vertical resolution of the measurements. Some discussion of the data quality is warranted to help evaluate the comparisons with CLaMS results later in the manuscript. This information may be contained in the paper by Chirkov et al., but some basic data quality information needs to be included here as well for the convenience of the reader. See related comment below.

P7, L29 – P8, L11: These paragraphs are confusing, because the first sentence (P7, L29), as well as the subsection title, refer to transport of emission tracers to "the top of the Asian monsoon anticyclone", yet Figure 2 (top row) and the related discussion focus on 360 K, which is obviously not at the top of the anticyclone. It may be that the discussion begins with 360 K because that level is where regions "inside" and "outside" the anticyclone are defined, which seems to be what is implied by the sentence in P8, L9-10, but if so then that motivation needs to come earlier in the paragraph to set the stage. Moreover, if that is the case, then I am confused by that as well – why define inside/outside the anticyclone at a single level, rather than at each considered level, since the shape of the anticyclone changes considerably with height? And Fig. 3 defines the anticyclone by the 20% contour of the India/China tracer at 380 K (not 360 K). So this entire discussion needs to be clarified.

P8, L15: To my eye, it looks as though fractions as high as 40% extend lower than 350 K, down to at least 340 K, if not lower.

P8, L20-21: It is not clear exactly which regions are being referred to for these values; in particular, in some areas (∼310-330 K, 10N) fractions from the tropical adjacent

regions much higher than 10%-40% are seen.

P8, L28-29: First, the region "inside the anticyclone" is referred to here, but it is not possible for the reader to identify where the anticyclone boundary falls at different altitudes in the cross sections of Fig. 2. The authors should think about how to convey information about the approximate location of the anticyclone in these panels. Second, it is stated that near the tropopause the fraction from the tropical adjacent regions reaches as high as 35%, but I am not sure exactly where is being referred to, as most TAR fractions in the vicinity of the monsoon in Fig. 2d are no larger than 25-30%.

P9, L11-25: I agree that the HCFC-22 data show good agreement with the India/China emission tracer and that they are a very useful element of the analysis. However, Fig. 3 reveals quite a few stray data points well outside the anticyclone that also have elevated HCFC-22 abundances. As mentioned earlier, the precision of an individual data point should be given so that the agreement in Fig. 3 can be fully evaluated. It seems to me that the enhancement in the thin filament (L15) does not particularly stand out in the measurements; indeed, in the absence of the CLaMS results to guide the eye, it likely would be overlooked altogether. Likewise, the measured enhancements at the top of the anticyclone above the tropopause (L20) are also fairly modest; in fact, they are not much different from other high MIPAS points well away from where CLaMS indicates a signal (e.g., at the EQ at 370 K, at 5N at 420 K, and at 30N at 430 K). It might help to also overlay on these plots (both the map and the cross sections) a solid contour highlighting a selected HCFC-22 mixing ratio. Although the "dot plots" are very valuable for representing the sampling of the MIPAS measurements, they do make it more difficult to get an impression of the overall morphology. Overlaying one specific contour from a gridded HCFC-22 field might strengthen the case for good agreement with the modeled tracer. Finally, although I do see a steep vertical gradient in the HCFC-22 data from $\sim$350 to 360 K in the $\sim$25-40N region, I do not see a corresponding signature in the India/China tracer in that region (L25); there is a steep gradient in that tracer in Fig. 2g, but at altitudes below 350 K, so the patterns in the 350-360 K region

are not really that similar.

P10, L6-11: Again, this discussion refers to "within" and "in the core of" the anticyclone, so some means of delineating exactly where that region is at each level is needed. In Fig. 3, the 20% contour for the India/China tracer is used to approximate the boundary of the anticyclone at 380 K, but what about at the higher levels shown in Fig. 4? How is the reader to gauge that the largest contributions of both emission tracers are found within the anticyclone at 400 K but around its edge at 420-460 K, as stated here? In fact, I am not convinced that either statement is true: the eastern lobe of the anticyclone (∼100E) shows the largest fractions of the TAR tracer along what looks to me more like the edge of the anticyclone at 400 K, whereas the largest values of both tracers seem to be concentrated in the core region at that longitude at 420 K.

P10, L16-19: How were the percentages of young air masses for the selected air parcels chosen? In the absence of any explanation these values seem arbitrary. Are these trajectories initiated from the entire region within the defined lat/lon boxes? I'm wondering if these percentages can be related to the values shown for the India/China tracer in Fig. 4.

P10, L22-23: How consistent are the trajectory results, which indicate that the Tibetan Plateau and the western Pacific are preferred regions for fast uplift, with prior studies (in other words, some citations would be appropriate here).

P10, L24-26: Is there a reason that the corresponding plots for the eastern lobe of the anticyclone were not shown in Fig. 5, as they were in Fig. 6? I would have thought that they would be relevant to the discussion here.

P11, L14-26: What exactly is meant by "substantial" upward transport (L14)? Does "substantial" mean 0.5 K/day, 1 K/day, or?? It would be better to be more quantitative. In addition, here the discussion is cast in terms of heating rate (K per day), whereas Fig. 7 and Fig. A1 show the change in potential temperature (in K) along 20-day trajectories, making the reader do the (admittedly easy) math. Once the meaning of

"substantial" is established, it would be better to qualify the transport experienced by air parcels grouped in filaments as being "substantial" or "strong" (L16) – filamentary structure is not present everywhere that air parcels have experienced some uplift. I also think it would be better to say "largely", rather than "only", in L26 because there are red dots outside the monsoon region, especially in July and August.

P12, L11: It might be good to explain why the emphasis has shifted from the tropical adjacent regions examined in previous figures to Southeast Asia specifically in Fig. 9, especially since Fig. 12 shows that the TPO also makes a substantial contribution to the air at 550 K.

P12, L14-17: It is stated that an enhanced signal from Southeast Asia of up to 25% (L14 and L17) is seen around 550 K for the S07 pulse, but as far as I can tell from Fig. 9, the largest S07 enhancement (at ∼10S) is only ∼12%, not 25%.

P12, L27-28: I do think it is important to point out the uncertainties in the reanalysis heating rates, as done in these lines. However, the way this paragraph ends leaves the reader hanging a bit. What is the take-away message? Can we trust the results in Fig. 10 or not? What are the possible implications for the 'upward spiraling range'?

P12, L29: It is stated that Fig. 11 shows the same cross sections as Figs. 8 and 9. The latter two figures, however, show results only for the eastern lobe (90E), whereas Fig. 11 also shows the cross section for 30E. Although we have some information about S08 in that region from Fig. 2, we do not get the full picture from that figure, and thus we have little to compare to the left panel of Fig. 11. I note that, in terms of major features, the HCFC-22 results look quite similar at 30E and 90E. Is that also the case for the CLaMS results, that is, do the corresponding plots at 30E look similar to those in Figs. 8 and 9? If so, then that should be mentioned, and perhaps the left panel of Fig. 11 should also be omitted. If not, discussion of the differences should be included.

P16, L3-4: Has evidence for a coherent signature of the existence of the anticyclone and influence of monsoon air up to altitudes as high as 460 K been reported previously?

It seems to me that this may be an important finding that has been underemphasized in this manuscript.

P26, Fig. 2: Perhaps it would make the maps in the top row too cluttered, but I think it would be helpful to draw on them a horizontal line at 25N and vertical lines at 30E and 90E to orient the reader for the cross sections in the bottom panels. In addition, I understand that a common color bar is used for all panels in this figure, and I agree that that is probably the best approach, and I further agree that extending the color bar to 100% is appropriate for the cross sections. However, I note that employing such a color bar renders some of the features in the maps less prominent. For example, the filament at 50E seen so clearly at 380 K in Fig. 3, where the tracer color bar extends only to 50%, is nearly invisible in Fig. 2 but might show up well if the color bar range were reduced. I am not suggesting that the color bar should necessarily be changed, merely pointing out the issue.

P27, Fig. 3: I found the figure layout and accompanying discussion hard to follow. Here the latitude-theta cross section at 30E comes first, then the one at 90E, and finally the longitude-theta cross section, which is essentially opposite to the order followed in Fig. 2. It would make it easier to compare the CLaMS and MIPAS results if Fig. 3 were configured as a single-column figure following the same layout as Fig. 2 (with an extra panel at the top for the India/China tracer and the MIPAS panels corresponding to those in Fig. 2 below). In addition, I do not understand why only in Fig. 2 are the panels labelled. Panel labels would be helpful in Fig. 3 and all other multi-panel figures as well. This would simplify referencing the figures in the text, eliminating the need to always point to top, middle, bottom, left, right, etc.

P31, Fig. 7: Again, I think this figure would work better laid out in a single column. In addition, I find the transition between upwelling and downwelling in these maps awkward – the zero value of delta(theta) lies between two pale blue colors, and thus cannot be readily identified.

Minor points of clarification, wording suggestions, and grammar / typo corrections:

P1, L9: To avoid any possibility of confusion, I think "boundary sources" should be "boundary layer sources"; also in this line "transport pathway" should just be "transport"

P2, L1: I think it would be better to add "and is" between "summer" and "associated"

P2, L9: a large variability –> large variability; anticyclone reaching –> anticyclone, which reaches

P2, L13: referred –> referred to

P3, L7-8: relation . . . influence –> relationship . . . influences

P3, L11-12: with observations of global . . . measurements of the –> with global . . . measurements from the

P3, L25: the the –> the; between 360 K –> from 360 K

P3, L33: as –> us

P4, L5: at top –> at the top

P5, L27: having "Tropical AR" in quotes and bold font gives the reader the impression that this is an important acronym that will be used again, whereas "tropical adjacent regions" is always written out in full in the text. "Tropical AR" seems to be used only in figure labels; in Table 1 this area is referred to as "TAR". It would be better to be more consistent in the usage.

P6, L5: an added –> added

P6, L22-23: associated to –> associated with; delete "anymore"

P6, L26: Asia –> Asian

P7, L6 and also L8: synoptical –> synoptic

P7, L26: the 18 August –> 18 August

[Figure]

P8, L1: and for the –> and that for the

P8, L6-8: the lack of strong tracer gradients on the equatorward side of the anticyclone has been noted several times, so some references to previous work would be appropriate here.

P8, L15: low values of the tropical adjacent regions –> low fractions from the tropical adjacent regions

P8, L24-25: this wording is confusing. It would be clearer to say: "At 90E (Fig. 2c), a layer of young air masses with enhanced India/China fractions extends well above the thermal tropopause, with values as high as 20% up to 420 K."

P9, L4-5: "in particular" is repeated twice in these lines, thus "restricted regions, such as" would be better. In addition, this point was made previously not only in Section 1 as noted, but also in Section 2.4 (P6). It may not be necessary to provide this information three times, so the authors might consider deleting it from Section 2.4.

P9, L9: delete "percentages of"; are marked –> is marked on the cross sections

P9, L18: mode –> modes; also, I feel it would be more appropriate to say "broadly consistent"

P9, 26: it would be clearer to say "smaller" rather than "lower" mixing ratios (since this sentence also talks about "below" and "above")

P10, L6-8: Restructuring this sentence would make it easier to interpret: "At 380 K, the highest fractions of air from India/China and from the tropical adjacent regions are found in the core of the anticyclone and at its edge, respectively."

P10, L10: "vertical upward" is redundant in this context; use one or the other, not both

P10, L15: started –> starting; mode –> modes

P10, L26: upward transport –> vertical transport (to avoid repeating "upward")

P10, L27: part –> parts

P10, L29: western and eastern part –> western and eastern parts of the anticyclone

P10, L30: "vertical upward" – same comment as above

P11, L10: mode –> modes

P11, L11: to what does "in this region" refer? The tropics, or 360 K, or ???

P11, L18: mode –> modes

P11, L29: the Appendix A –> Appendix A

P12, L7: boundary regions –> boundary layer regions

P12, L9: winter time –> winter

P12, L10: boundary emissions –> boundary layer emissions

P12, L17: larger as –> larger than; also delete "(Winter 07/08 pulse)" after "tracer"

P12, L18: winter time –> winter

P12, L23: analysis –> reanalysis

P12, L26: tropopause which again are –> tropopause, which in turn are

P12, L29: longitude –> latitude

P12, L32: from Summer –> from the Summer

P12, L34: Asia –> Asian

P12, L31: it might be good to add "just" here: "a combination of just two signals", to make a stronger contrast

P13, L1: winter time –> winter

P13, L4: velocity –> velocities; summer time –> summer

P13, L6: exists; that –> exist that

P13, L11: This point was already made in Figs. 8 and 9; thus it would be better to refer back to those figures here than to point ahead to Fig. 12, which is not introduced until the following paragraph.

P13, L12: boundary –> boundary layer

P13, L26: highest –> largest

P13, L30: boundary emission –> boundary layer emission

P14, L3: the the –> the

P14, L7: contribute to a lower –> contribute a smaller

P14, L8: CLaMS –> the CLaMS; include –> includes

P14, L30: the the –> the

P15, L17: is already –> has already been

P16, L20: 1 K per day - 1.5 K per day –> 1-1.5 K per day (as done everywhere else in the paper)

P16, L22-30: these lines are a bit garbled. First, it is odd to have a 1-sentence paragraph (L22-24). Second, L30 starts with "Further" but then repeats verbatim the sentence in L23-24. These sentences need to be merged / rearranged / rewritten. Third, the sentence in L25-26 is hard to read. It would be clearer to say: "Thus, within the upward spiralling range above the anticyclone, young air masses from along its edge originating in the tropical adjacent regions are mixed with air masses from inside the anticyclone mainly originating in India/China." Finally, L29: consisted –> consistent

P17, L2-5: It is stated that fresh emissions from the 2008 monsoon season do not contribute to the distribution within the tropical pipe at 550 K. However, emissions from that season would eventually reach 550 K, so this statement needs to be qualified

in some way (for example, by adding "before October 2008" or something similar). Similarly, it might be good to add "in October 2008" after "550 K" in L5.

P17, L8: here too I think it would be better to delete "pathway"

P17, L13-14: region air masses from the tropical adjacent regions (Southeast Asia/tropical Pacific/northern Africa/northwestern Pacific) are transported in a substantial percentage by this pathway into the tropical pipe –> region, a substantial percentage of air masses from the tropical adjacent regions (Southeast Asia/tropical Pacific/northern Africa/northwestern Pacific) is transported by this pathway into the tropical pipe

P26, Fig. 2 caption: Rather than "first" and "second", it may be better to refer to the tropopauses as "primary" and "secondary". Also, in the last sentence, "percentages" should be deleted, and "(cross sections)" should be added after "white lines".

P27, Fig. 3 caption: thick black or grey lines –> thick black (maps) or grey (cross sections) lines

P29, Fig. 5 caption: reversed –> back; single –> successive

P30, Fig. 6 caption: reversed –> back

P31, Fig. 7 caption: are shown –> is shown; 1st row –> 1st panel; rows –> panels

P32, Fig. 8 caption: again, "primary" and "secondary" may be better than "first" and "second"

P33, Fig. 10 caption: again, "primary" and "secondary" may be better than "first" and "second"

P34, Fig. 11 caption: eastern mode (80E-100E) –> eastern (80E-100E) mode

P35, Fig. 12 caption: (1) this would be easier to read if "(top)" were moved to before "The contribution" in L1 and "(bottom)" were moved to before "The contribution" in L5.

(2) "by October 2008" should be added at the end of "550 K" in L4. (3) "The contribution of the three time pulses" –> The contributions of the time pulses" (it is confusing to say three since only two are shown). (4) in the legend of the figure itself, "Residual" should be "Residual surface" to be consistent with the text.

P36, Fig. 13 caption: transport pathway –> transport

P38, Fig. A2: In the legend, Residual –> Residual surface

---

## Referee Comment (RC2) · Anonymous Referee #2 · 14 Aug 2018

The study "Lagrangian simulations of the transport of young air masses to the top of the Asian monsoon anticyclone and into the tropical pipe" investigates transport processes from the boundary layer to the monsoon anticyclone and further into the stratosphere by employing 3-D CLaMS simulations with mixing and additional backtrajectory data. Further, comparisons with satellite data (MIPAS) are included, which increase the confidence in the presented results. Overall, the manuscript is well written and the figures and analyses are well composed. Further, the study contains interesting results suitable for publication in ACP. Nevertheless, I think that the following comments need to be addressed before the manuscript can be published. In particular, the manuscript would benefit from (and in my opinion needs) an extended discussion of the presented

results with respect to previous publications.

General comments:

In the abstract and in the text, you distille the transport processes from the boundary layer to the tropical stratosphere into 3 separate regimes. In my opinion, this result is in agreement with previous notions on transport of ASM air massse, i.e. I think it is known that the upper troposphere in the Asian summer monsoon region is strongly affected by convection (e.g. Randel and Park, 2006, show a strong impact from convection at 350-360K), slow upward movement within the anticyclone is addressed e.g. in Park et al. (2007, 2009; see also the schematic Fig. 14 in the latter publication). Further, Ploeger et al. (2017) show slow upward transport of Asian summer monsoon air masses in the tropical pipe (cf. also Randel et al. 2010) and also presents an overview of transport processes in the ASM region in its introduction. There are also other studies addressing transport in the Asian monsoon and some that also mention slow ascent in the UT in the monsoon region: e.g. Wright et al. (2011), Bergman et al. (2012) and Garny and Randel (2016). Nevertheless, it is indeed interesting to have a single study (and model) that shows all of the transport regimes and your study includes additional information. Please, relate your results to these previous findings or suggestions and carve out how your study differs/agrees with the processes described there. How do your results complement these previous suggestions/findings? Maybe you could commet also on the influence of extremely deep (or even overshooting) convection on air masses within the UTLS in the Asian monsoon region.

Related to this issue, you state a convective regime and I wonder how convection is treated in your simulations and backward trajectory calculations. Please incorporate some notion on how the setup of your simulations/trajectories will affect your results.

P.8 L.31-31: "...how can air masses...?": you pose this question, however, to me it is not clear where it is answered. Are you thinking about inmixing from the outside to the inside/edge of the AC and subsequent vertical transport. Please connect to the

parts in the text where this question is answered and/or e.g. repeat the question and give the answer to it in the conclusion. Would it be possible to include the transport of air masses from adjacent regions above 380K also in your Fig. 13?

Most of the analysis are focused on one day (18 August 2008), only. For some of the analyses this might not be important, however, other analyses might depend on the specific conditions (e.g. the split of the anticyclone) during that date/period as for example the trajectory analysis in Fig. 5. Please include some additional discussion regarding that issue. Partly, you have already addressed this issue, e.g. to complement Fig. 7 you additionally include Figure A1. I would guess that in particular the backward trajectories results are affected by the choice of the starting date and might vary throughout the monsoon season. This issue also extends to the comparison with MIPAS data and to the inferred transport on the eastern/western side of the anticyclone.

Fig. 8 in Garny and Randel (2016) shows kinematic and diabatic vertical velocities and Fig. 12 a) in Park et al. (2007) shows pressure tendencies. These figures show ascent on the eastern side of the anticyclone and descent on the western side at levels close to (but still mostly below) the tropopause. How do your statements and your Fig. 10 relate to that? How does the climatological picture of Fig. 10 look like? Is there always (i.e. on a climatological basis) stronger heating above the western side of the anticyclone above the tropopause but cooling below? How is it on the eastern side?

Additionally, I think it would be very helpful if you relate the results of your tracer pulses shown in Figs. 8 and 9 to the results in Ploeger et al. (2017). In particular with respect to transport of air masses from the anticyclone to the deep stratosphere.

Specific comments:

P1 L19-20: Regarding the effectiveness of horizontal mixing and vertical transport, Garny and Randel (2016) seem to come to a different conclusion. Please discuss (e.g. in Sect. 4) how your results agree and differ. In case of the latter please also discuss why they differ.

P3 L7-10: Regarding the connection of El Nino and La Nina with the Indian summer monsoon. You argue that 2008 was (in terms of rainfall) normal because of La Nina in the winter before, although, in the previous sentence you claim that El Nino and La Nina events tend to be connected to unusual rainfall in the following Indian summer monsoon season. This seems contradictory to me! Further, Kumar et al. (2006) show a relation of concurrent SSTs with rainfall in India (during a quick search I could not find that they are stating a connection with previous winter SSTs). Also, in Webster et al. (1998) I could not easily find to which SST anomaly they refer, i.e. previous/following winter or concurrent summer. Please comment on this and revise if necessary.

P5 L3-5: Is the sum of the tracers for all parcels in the boundary layer really always equal to 1 as you describe on page 5 L3-4. What if unmarked parcels from above the BL are transported into the BL? Are they removed? Otherwise, they might not be marked in the BL as marking takes place every 24h, only. Does the time step of 24h release play an important role? As an example, Bergman et al (2013) use backward-trajectories started every 6 hours. Why don't you mark/emmit the tracer "continuously", i.e. at every time step of the simulation? Is there a scientific or technical reason for this setup.

P6 L1: Please add whether the trajectories described in this section are calculated using heating rates (as I would assume) or kinematic vertical velocities.

P7 L6-9: Has the same method for interpolating MIPAS HCFC-22 data been used in Vogel et al. 2016? Then you could add a note so it is clear.

P8 L16 and following as well as P9 L26-27: Either in the description of Fig. 2 2nd row and Fig. 3 2nd row left or in the discussion you should draw a relation to Pan et al. (2016), who showed that upward transport (e.g. of CO) is mainly focused on the eastern side of the AC.

P10 L13: Please state that you are starting trajectories only on 18 August 2018 for the analyses in Sect. 3.2.1. Or have you analysed other dates as well?

P10 L22-23: How do you know that the transport occurs above the Tibetan Plateau? From Fig. 5 only the longitudinal range is visible but not where in latitude the parcels ascend. If you have made additional analyses to check that they are indeed from the Tibetan Plateau just note that you have analysed this but chose to not include a figure or the analysis here.

P10 L32: At some instances (e.g. here at P10 L32) you refer to inside or outside the anticyclone but do not give a reliable definition or state what you consider as inside or ouside. Would it be an option to include PV contours for that purpose? Also on P11 L5 you should probably rephrashe to "entire Asian monsoon region" because you do not start only within the anticyclone.

P11 L14: Maybe you should rephrase this part stating "At 380..." instead of "Above 380K,..." because at 400K the structures are not as inhomogeneous anymore and above 400K there is also considerable upward transport in the tropics.

P. 12 L14 and L17: Two times 25% instead of ∼15% is mentioned, as I assume would be correct. If I am correct, the 25% are the contribution of the winter pulse (W07) at ∼450K, right?

P14 L6-8: Do you really mean "Asian monsoon air masses from the anticyclone" or rather air masse from your India/China tracer? I think your findings show the claimed relation only for the latter.

P14 L30-31: I think slow upward transport has been proposed earlier (see my general comment). Please clarify if you are referring to some specific point of the upward transport process that was not published earlier.

I think it would be good to label all panels of all figures with (a), (b), (c) and so on as you do for example in Fig. 3 but not in Figs. 4, 5 etc. This is just a suggestion, but would definitely help to increase the readability. Then you could refer directly to the individual panels of the figures and it would be consitent throughout the manuscript.
Also, consider to add additional references to the individual panels in the text when you draw a conclusion or describe something that is based on the respective panel.

Minor suggestions/corrections:

P1 L11: Either change to "Second, these air masses..." or "Second, air masses are uplifted within the anticyclone..." or something similar.

P1 L14: As before, maybe clarify by changing your sentence to something like: "Third, transport of air masses affected by the Asian monsoon (anticyclone)..." or something similar.

P2. L1: This probably needs some additional restriction to where the the Asian monsoon is the "most pronounced circulation pattern". Do you refer here to the tropospheric flow or the UTLS anticyclone?

P2 L21-22: Order references according to year of publication.

P3 L1: Would it be better to change "defined regions" to "specific regions"?

P3 L2: Shouldn't this read: "covering Earth's entire surface". Then it would need to be changed throughout the manuscript.

P3 L32-33: Maybe change to "...a total simulation period of 18 months)."

P4 L2-6: The two sentences starting with "With this approach..." and "This model setup..." seem somehow repetitive. If they are not, please try to clarify.

P9 L4-5: Repetition of "in particular". Please rephrase.

P13 L6: "exists" should be "exist". Also consider to rephrase, e.g. to "... pathways exist. On these horizontal pathways, air masses are transported isentropically..."

P13 L10: I would suggest to shift the first sentence of the paragraph ("On 18 August....") behind the current second sentence ("To analyse...") or/and adapt as it seems to be doubled at the moment.
[Figure]

P13 L33: Probably this should be "...from the tropical...".

P14 L13: Are "Asian monsoon anticyclone" and "Asian monsoon" switched here?

P32: In the caption of Fig. 8 it should state "...(1 May 2007 - 18 August 2008)..." instead of "...(1 May 2007 - 31 October 2008)...", because you show the tracer distribution on 18 August 2008. This is also how you describe the figure in the text.

References:

Bergman, J. W., Jensen, E. J., Pfister, L., and Yang, Q.: Seasonal differences of vertical-transport efficiency in the tropical tropopause layer: On the interplay between tropical deep convection, large-scale vertical ascent, and horizontal circulations, J. Geophys. Res., 117, https://doi.org/10.1029/2011JD016992, 2012.

Bergman, J. W., Fierli, F., Jensen, E. J., Honomichl, S., and Pan, L. L.: Boundary layer sources for the Asian anticyclone: Regional contribu- tions to a vertical conduit, J. Geophys. Res., 118, 2560–2575, https://doi.org/10.1002/jgrd.50142, 2013.

Garny, H. and Randel, W. J.: Transport pathways from the Asian monsoon anticyclone to the stratosphere, Atmos. Chem. Phys., 16, 2703– 2718, https://doi.org/10.5194/acp-16-2703-2016, 2016.

Kumar, K. K., Rajagopalan, B., Hoerling, M., Bates, G., and Cane, M.: Unraveling the Mystery of Indian Monsoon Failure During El Niño, Science, 314, 115–119, https://doi.org/10.1126/science.1131152, 2006.

Park, M., Randel, W. J., Gettleman, A., Massie, S. T., and Jiang, J. H.: Transport above the Asian summer monsoon anticyclone inferred from Aura Microwave Limb Sounder tracers, J. Geophys. Res., 112, D16309, https://doi.org/10.1029/2006JD008294, 2007.

Park, M., Randel, W. J., Emmons, L. K., and Livesey, N. J.: Transport pathways of carbon monoxide in the Asian summer monsoon diagnosed from Model of Ozone and Related Tracers (MOZART), J. Geophys. Res., 114, D08303,

https://doi.org/10.1029/2008JD010621, 2009.

Ploeger, F., Konopka, P., Walker, K., and Riese, M.: Quantifying pollution transport from the Asian monsoon anticyclone into the lower strato- sphere, Atmos. Chem. Phys., 17, 7055–7066, https://doi.org/10.5194/acp-17-7055-2017, https://www.atmos-chem-phys.net/17/7055/ 2017/, 2017.

Randel, W. J. and Park, M.: Deep convective influence on the Asian summer monsoon anticyclone and associated tracer variability observed with Atmospheric Infrared Sounder (AIRS), J. Geophys. Res., 111, D12314, https://doi.org/10.1029/2005JD006490, 2006.

Vogel, B., Günther, G., Müller, R., Jens-Uwe Grooß, A. A., Bozem, H., Hoor, P., Krämer, M., Müller, S., Riese, M., Rolf, C., Spelten, N., Stiller, G. P., Ungermann, J., and Zahn, A.: Long-range transport pathways of tropospheric source gases originating in Asia into the northern lower stratosphere during the Asian monsoon season 2012, Atmos. Chem. Phys., 16, 15 301–15 325, https://doi.org/10.5194/acp- 16-15301-2016, http://www.atmos-chem-phys.net/16/15301/2016/, 2016.

Webster, P. J., Magaña, V. O., Palmer, T. N., Shukla, J., Tomas, R. A., Yanai, M., and Yasunari, T.: Monsoon: Processes, predictability, and the prospects for prediction, J. Geophys. Res., 103, 14.451–14.510, 1998.

Wright, J. S., R. Fu, S. Fueglistaler, Y. S. Liu, and Y. Zhang (2011), The influence of summertime convection over Southeast Asia on water vapor in the tropical stratosphere, J. Geophys. Res., 116, D12302, doi: 10.1029/2010JD015416.

---

## Referee Comment (RC3) · Anonymous Referee #3 · 25 Sep 2018

The manuscript "Lagrangian simulations of the transport of young air masses to the top of the Asian monsoon anticyclone and into the tropical pipe" investigates transport pathways in the monsoon region from the boundary layer into the stratosphere. The authors use both Lagrangian backward trajectory calculations and three-dimensional simulations including irreversible mixing with the Lagrangian transport model CLaMS. Artificial tracers of air mass origin are compared to measurements of chlorodifluoromethane (HCFC-22; $CHClF_2$) by the Michelson Interferometer for Passive Atmospheric Sounding (MIPAS). The methods are similar to those in Vogel et al. (2016) but in addition to horizontal transport they address vertical transport pathways out of the Asian monsoon anticyclone and subsequent upward transport into the lower strato-

sphere. The chosen period is a normal monsoon season in terms of medium rainfall over India in summer 2008. The paper presents an interesting description of the "spiralling staircase". Consistency with MIPAS data supports the model results. The paper is well written and I support publication after a few comments have been addressed. The heavy focus on a particular day is didactical, but the analysis of different meteorological situations would provide stronger evidence. In certain passages the phrasing could be improved in order to more clearly distinguish what is the particular contribution of this study to the "longstanding debate about the transport mechanisms at the top of the Asian monsoon anticyclone and beyond into the stratosphere". The debate is mentioned again in the discussion but the different arguments about "the exact transport mechanism" could be more clearly stated. Otherwise it is difficult to distinguish if some very general assertions have been stated before in the literature or are novel to this work. The figures could be improved and the description thereof clarified in the text.

Minor comments:

p1 l19 "However, this upward transport": Make "this" more clear (the upward spiralling range).

p2 l20 Briefly mention the sides in the debate and state where the authors stand.

p5 l3 $\zeta < 120K$? p5 l 17 "is quantified." What will be the quantitative measure?

p6 l8 Dee et al. (2011) already cited in the first reference to ERAI in p4 l11

p7 l20 make more explicit the dates of the monsoon and the pulse releases to clarify the 6 month age.

l22 "have strong variability from day to day" please rephrase

p8 l1 mention the 360 theta level before in the text to streamline the reading.

p8 l10 "simulated horizontal gradients": is there an objective metric or visual inspection? How is the top of the asian monsoon precisely defined?

p8 l17 Fig. 2 (3rd row), to be consistent with previous paragraph.

p10 l15 When are the trajectories started? How many? What release pattern? It would help for understanding what was done to state this clearly in the text. 40 days could be analysed statistically, bur for individual trajectories is a little bit too long.

p10 l20 describe the panels of fig 5 in the text.

p10 last paragraph. At this point the reader would feel satisfied with a statistical analysis of a larger number of days to support the case study results. Maybe some additional results such as those later presented in A1 could be mentioned.

p11 l4 what do you mean with "single selected trajectories"?

p11 l32 "In the previous sections, we could show that the Asian monsoon is an effective circulation pattern in the UTLS that transports very young air masses (< 6 months) from the surface into the lower stratosphere up to $\approx$ 460 K." This may have been mentioned before in the literature. You could rephrase this as "we could show how the effective circulation pattern in the UTLS can be seen with CLaMS and MIPAS data, for example. Also it seems that all your conclusions will be drawn from a single day case study. Additional statistical evidence from more modelling cases could help.

12 4 is it a CLaMS simulation?

Fig 8. The figure is difficult to read. The label at the color bar is very small and takes time to find. You could replace the title of the subplots "08081812 at 90 E" that is the same for all and put is in the cation. "horizontal winds (black lines)", do you mean horizontal wind absolute value isolines? "corresponding levels of pressure " corresponding to what? "Pressure levels as white lines" would be better. Are the values really zero in W07 (lower left panel) or is the color scale?

"demonstrating" may sound a bit strong for this context. "Suggesting" or "indicating" could fit better.

p12 l14 If still describing Fig. 9 better to keep the same paragraph.

Figure 10: again, wind speed contours, "pressure levels" instead of "corresponding levels of pressure"

p13 l11: attract the attention of the reader to the vertical dashed line immediately here.

Fig 13 what is the dashed line?

p15 l12 please refer to the published version.

p16 l17 "Thus, air masses in the upward spiral range are uplifted by diabatic heating across the (lapse rate) tropopause, which does not act as a transport barrier against this diabatic vertical transport process.: This assertion is likely to be "consistent with previous studies".

p16 l 22 occurs where?

The last paragraph is a nice summary of the mechanism but it undoubtedly draws from the conclusions of many previous studies. This should somehow be acknowledged.

---

## Author Comment (AC1) · 23 Nov 2018

**Author Comment to Referee #1**

ACP Discussions doi: 10.5194/acp-2018-724 (Editor - Peter Haynes) 'Lagrangian simulations of the transport of young air masses to the top of the Asian monsoon anticyclone and into the tropical pipe'

We thank Referee #1 for further guidance on how to to revise our paper. Following the reviewers advice we revised some parts of the paper for the purpose of clarification. In particular, we want to thank the reviewer for the elaborate language corrections. This was a very great support. Our reply to the reviewer comments is listed in detail below. Questions and comments of the referee are shown in italics. Passages from the revised version of the manuscript are shown in blue.

This manuscript uses back trajectories and full 3D CLaMS simulations in conjunction with MIPAS HCFC-22 measurements to elucidate the transport pathway of air masses emitted in defined boundary layer regions through the Asian summer monsoon anticyclone and into the tropical pipe. The modeling tools and measurements are well suited to the investigation, the analysis is generally well thought out and well executed, and the findings will certainly be of interest to the journal readership. I do, however, have a number of substantive comments that I would like to see addressed before the paper is accepted for publication in ACP.

**Specific substantive comments and questions:**

Sections 2.2 and 3.2.1: 40 days seems like a very long period for trajectory calculations. I realize that CLaMS 40-day trajectories have been published previously, but nevertheless I think that a sentence or two on how much error has accumulated over the course of such long trajectory calculations would be appropriate, either in Section 2.2 or in Section 3.2.1.

We agree and have thus added a few sentences in Section 3.2.1. to discuss the errors of the trajectory length as follows.

In general, trajectory calculations have limitations due to trajectory dispersion by errors through interpolation of the wind data to the position of the air parcel at a specific time. Over the timescales in question, mixing can also be relevant (e.g., McKenna et al., 2002). These errors can accumulate depending on the trajectory length over the course of the simulation. However, the frequently employed trajectory length to study transport processes in the Asian monsoon region is ranging from a couple of weeks to a few months (e.g., Chen et al., 2012; Bergman et al., 2013; Vogel et al., 2014; Garny and Randel, 2016; Müller et al., 2016; Li et al., 2017). In our trajectory analysis, the focus is to demonstrate the large-scale transport pathways of the air parcels at the top of the anticyclone, small changes of the trajectory position will therefore not affect our findings.

Section 2.4: I miss in the description of the MIPAS HCFC-22 any information about the accuracy, precision, or horizontal or vertical resolution of the measurements. Some discussion of the data quality is warranted to help evaluate the comparisons with CLaMS results later in the manuscript. This information may be contained in the paper by Chirkov et al., but some basic data quality information needs to be included here as well for the convenience of the reader. See related comment below.

As suggested we provide some additional information about the MIPAS HCFC-22 data quality in Section 2.4 as follows.

The limited vertical resolution of the MIPAS HCFC-22 data needs to be taken into account in comparisons to model results. The precision of an individual data point in the altitude region of the Asian monsoon tropopause is 7 to 8 pptv in terms of measurement noise. Parameter errors contribute to a total uncertainty of about 15 pptv in this region for each data point. Thus, the scatter of the HCFC-22 data points (e.g. as shown in Fig. 3) is consistent to the total error. According to Chirkov et al. (2016), the vertical resolution (in terms of the full width at half maximum of the vertical averaging kernel) increases from about 3.3 km at 12 km to 5.5 km at 20 km altitude (see Fig. 2 in Chirkov et al., 2016). The horizontal resolution (in terms of the full width at half maximum of the horizontal averaging kernel) increases from 300 km

at 15 km altitude to 600 km at 20 km altitude. Given the rather smooth profiles expected in this study, the limited altitude resolution has a minor effect only; in contrast, it turns out to be crucial when highly structured profiles, such as typically occur at the edge of the polar vortex, are analysed. Further it has to be noted that tropical HCFC-22 profiles from MIPAS seem to have a high bias below 30 km, that, however, is broadly constant with altitude (Chirkov et al., 2016); thus, it does not affect the comparisons made here.

P7, L29 - P8, L11: These paragraphs are confusing, because the first sentence (P7, L29), as well as the subsection title, refer to transport of emission tracers to 'the top of the Asian monsoon anticyclone', yet Figure 2 (top row) and the related discussion focus on 360 K, which is obviously not at the top of the anticyclone. It may be that the discussion begins with 360 K because that level is where regions 'inside' and 'outside' the anticyclone are defined, which seems to be what is implied by the sentence in P8, L9-10, but if so then that motivation needs to come earlier in the paragraph to set the stage. Moreover, if that is the case, then I am confused by that as well - why define inside/outside the anticyclone at a single level, rather than at each considered level, since the shape of the anticyclone changes considerably with height? And Fig. 3 defines the anticyclone by the 20% contour of the India/China tracer at 380 K (not 360 K). So this entire discussion needs to be clarified.

We agree that here we have to provide a better motivation to explain our analysis. We introduced a motivation/introduction to Sect. 3.1.1 as follows.

It is known that the Asian monsoon anticyclone has a strong horizontal transport barrier at about 380 K (e.g., Ploeger et al., 2015), however this transport barrier is not well defined at higher levels of potential temperature. The less strong transport barrier at higher levels has consequences on the vertical transport at the top of the anticyclone. Before the transport at the top is discussed we show the horizontal distribution of different emission tracers at 360 K and then their subsequent transport to the top of the anticyclone up to 460 K. Vogel et al. (2015) showed that the emission tracer for India/China is a good proxy for the location and shape of the Asian monsoon anticyclone using pattern correlations with potential vorticity (PV), and MLS  $O_3$  and CO satellite measurements between 360 K and 400 K. Therefore here we use the India/China tracer as proxy for the location of the anticyclone.

P8, L15: To my eye, it looks as though fractions as high as 40% extend lower than 350 K, down to at least 340 K, if not lower.

Yes, we agree the fractions as high as 40% extend lower down to  $\approx 340$  K. We revised the paragraph as follows.

We would like to emphasise the horizontal transport of air masses with high contributions from India/China (40%–90%) from the eastern part of the anticyclone to both the western part and into the eddy over the western Pacific between  $\approx 340$  K and  $\approx 380$  K.

P8, L20-21: It is not clear exactly which regions are being referred to for these values; in particular, in some areas ( $\approx 310-330$  K, 10N) fractions from the tropical adjacent regions much higher than 10%-40% are seen.

We revised this paragraph to be more precise as follows.

In the western mode there is still a high contribution from the India/China tracer between 20% and 60% and lower fractions about 10%-40% from the tropical adjacent regions (Fig. 2g/h inside the thick white line). Below the western mode, in the tropics below  $\approx 330$  K at around  $10^{\circ}$ N fractions from the tropical adjacent regions (in that case from Northern Africa) are up to 90% caused by local upward transport.

P8, L28-29: First, the region 'inside the anticyclone' is referred to here, but it is not possible for the reader to identify where the anticyclone boundary falls at different altitudes in the cross sections of Fig. 2. The authors should think about how to convey information about the approximate location of the anticyclone in these panels. Second, it is stated that near the tropopause the fraction from the tropical adjacent regions reaches as high as 35%, but I am not sure exactly where is being referred to, as most TAR fractions in the vicinity of the monsoon in Fig. 2d are no larger than 25-30%.

A clear definition of the edge anticyclone over a large range of different altitudes is not possible such as a PV-based criterion (e.g., Ploeger et al., 2015). We use here as a proxy the distribution of the India/China tracer. Further, we agree the TAR fractions are to about 30%. We revised the respective sentence as follows.

Below 360 K in the region with high values of the India/China tracer, the fractions from the tropical adjacent regions are below 10%, however above 360 K around the tropopause the fractions are much higher, up to about 30%, and up to about 15% around 420 K (see Fig. 2d/f).

P9, L11-25: I agree that the HCFC-22 data show good agreement with the India/China emission tracer and that they are a very useful element of the analysis. However, Fig. 3 reveals quite a few stray data points well outside the anticyclone that also have elevated HCFC-22 abundances. As mentioned earlier, the precision of an individual data point should be given so that the agreement in Fig. 3 can be fully evaluated. It seems to me that the enhancement in the thin filament (L15) does not particularly stand out in the measurements; indeed, in the absence of the CLaMS results to quide the eye, it likely would be overlooked altogether. Likewise, the measured enhancements at the top of the anticyclone above the tropopause (L20) are also fairly modest; in fact, they are not much different from other high MIPAS points well away from where CLaMS indicates a signal (e.g., at the EQ at 370 K, at 5N at 420 K, and at 30N at 430 K). It might help to also overlay on these plots (both the map and the cross sections) a solid contour highlighting a selected HCFC-22 mixing ratio. Although the 'dot plots' are very valuable for representing the sampling of the MIPAS measurements, they do make it more difficult to get an impression of the overall morphology. Overlaying one specific contour from a gridded HCFC-22 field might strengthen the case for good agreement with the modeled tracer. Finally, although I do see a steep vertical gradient in the HCFC-22 data from  $\approx 350$  to 360 K in the  $\approx 25$ -40N region, I do not see a corresponding signature in the India/China tracer in that region (L25); there is a steep gradient in that tracer in Fig. 2g, but at altitudes below 350 K, so the patterns in the 350-360 K region are not really that similar.

We thank the reviewer for the advice to modify Fig. 3 by overlaying a solid contour highlighting a selected HCFC-22 mixing ratio. We tested different plotting types as shown in Fig. 1 of this author comment. Because of the coarse vertical resolution of the MIPAS measurements a solid contour depends strongly on the grid used for interpolation. Therefore, we prefer to show the plots without a solid contour line for a selected HCFC-22 mixing ratio.

Figure 1: Latitude–theta cross sections at 30°E (western part of the anticyclone) on 18 August 2008: contour plot, contour plot with a resampled finer grid (congrid), plotcell, and xy-plot. The contour line of 200 pptv HCFC-22 is shown in black. The contour line of 20% of the India/China tracer is shown by thick grey lines.

P10, L6-11: Again, this discussion refers to 'within' and 'in the core of' the anticyclone, so some means of delineating exactly where that region is at each level is needed. In Fig. 3, the 20% contour for the India/China tracer is used to approximate the boundary of the anticyclone at 380 K, but what about at the higher levels shown in Fig. 4? How is the reader to gauge that the largest contributions of both emission tracers are found within the anticyclone at 400 K but around its edge at 420-460 K, as stated here? In fact, I am not convinced that either statement is true: the eastern lobe of the anticyclone ( $\approx 100E$ ) shows the largest fractions of the TAR tracer along what looks to me more like the edge of the anticyclone at 400 K, whereas the largest values of both tracers seem to be concentrated in the core region at that longitude at 420 K.

We agree that this discussion is a bit confusing. We revised this paragraph as follows.

Young air masses (age < 6 months) from both India/China and tropical adjacent regions are found up to  $\approx 460$  K. It was shown earlier that the horizontal distribution of the India/China tracer is a good proxy for the location of the anticyclone (Vogel et al., 2015). The horizontal distribution of the tropical adjacent regions compared to the horizontal distribution of the India/China tracer strongly differs depending on the level of potential temperature from a nearly disjoint distribution at 360 K (see Fig. 2) to a more coincident distribution from 400 K to 460 K (see Fig. 5).

At 380 K, the highest fractions from tropical adjacent regions are found at the edge of the anticyclone and at 400 K within the anticyclone. Above 400 K both tracers India/China and tropical adjacent regions show a similar horizontal distribution. We emphasise that at these levels of potential temperature the tracer distributions have the shape of rotating filaments in contrast to the more compact distribution at lower levels. The variation of the distribution of the tracer for the tropical adjacent regions with altitude is an indication that the upward transport of young air masses at the top of the anticyclone occurred more towards the edge and less inside the anticyclone itself.

P10, L16-19: How were the percentages of young air masses for the selected air parcels chosen? In the absence of any explanation these values seem arbitrary. Are these trajectories initiated from the entire region within the defined lat/lon boxes? I'm wondering if these percentages can be related to the values shown for the India/China tracer in Fig. 4.

We agree that the description of the initialisation procedure of the trajectories is a bit short. We revised this paragraph as follows.

To analyse the transport pathways to the top of the anticyclone in more detail, 40-day backward trajectories are calculated starting in the western  $(20-50^{\circ}\text{N},0-70^{\circ}\text{E})$  and eastern  $(20-50^{\circ}\text{N},70-140^{\circ}\text{E})$  modes of the anticyclone. The trajectories are started at the position of the air parcels from the 3-dimensional CLaMS simulation at different levels of potential temperature ( $\Theta = 380, 400, 420, 440 \text{ K} \pm 0.25 \text{ K}$ ) on 18 August 2018. Note that the air

parcels in the 3-dimensional CLaMS simulation are distributed on an irregular grid. To take into account the distribution of the boundary emission tracer at the top of the Asian monsoon anticyclone, only air parcels are selected with contributions of young air masses (age < 6 months, Summer 08) larger than 70% (380 K), 50% (400 K), 20% (420 K), and 5% (440 K) (not all levels of potential temperature are presented here). The percentages are chosen in a way to obtain a number of trajectories (less than 30) that can be reasonably visualised. The results of the 40-day backward trajectories are similar at different levels of potential temperature; therefore we show a selection of trajectories to demonstrate the main transport pathway to the top of the Asian monsoon. A larger set of 20-day backward trajectories analysed statistically will be discussed below in Section 3.2.2.

P10, L22-23: How consistent are the trajectory results, which indicate that the Tibetan Plateau and the western Pacific are preferred regions for fast uplift, with prior studies (in other words, some citations would be appropriate here).

We revised this paragraph in Sect. 3.2.1 as follows and added a small discussion within Sect. 4 about convective source regions contributing to the composition of the Asian monsoon anticyclone following the advice by reviewer #2.

Fig. 5 shows trajectories started in the eastern and western part of the Asian monsoon anticyclone around the thermal tropopause at 380 K on 18 August 2018. Air masses are uplifted to approximately 360 K very rapidly by various convective events occurring at different times and locations. Our 40-day backward trajectories show that preferred regions for fast uplift are continental Asia (mainly the region of the south slope of the Himalayas and the Tibetan Plateau) and the western Pacific (not shown here). A lower fraction of trajectories originates in the free troposphere. The trajectories in Fig. 2 demonstrating convection below 380 K are only a snapshot for 18 August 2018. There are several previous studies (e.g., Randel and Park, 2006; Park et al., 2007, 2009; Wright et al., 2011; Chen et al., 2012; Bergman et al., 2013; Fadnavis et al., 2014; Tissier and Legras, 2016) quantifying the contribution of different source regions to the composition of the Asian monsoon anticyclone during the course of the monsoon season (see discussion in Sect. 4).

We added in the Discussion Section (Sect. 4) the following discussion.

It is well known that the composition of the Asian monsoon anticyclone is strongly affected by convection over continental Asia (e.g. the south slope of the Himalayas and the Tibetan Plateau), Bay of Bengal, and the western Pacific, (e.g., Randel and Park, 2006; Park et al., 2007, 2009; Wright et al., 2011; Chen et al., 2012; Bergman et al., 2013; Fadnavis et al., 2014; Tissier and Legras, 2016). However there is a debate about the contribution of different source regions to the composition of the Asian monsoon anticyclone. Further, there are differences in the conclusions in the literature about the contribution of different source regions depending on the used reanalysis data (e.g., Wright et al., 2011; Bergman et al., 2013). Findings by Vogel et al. (2015) show that there is a strong intraseasonal variability of boundary source regions to the composition of the Asian monsoon anticyclone during a particular monsoon season. We would like to emphasise that the trajectories presented in Fig. 6 demonstrating convection below 380 K are only a snapshot for 18 August 2018 with convection over the western Pacific and continental Asia mainly in the region of the south slope of the Himalayas and the Tibetan Plateau.

P10, L24-26: Is there a reason that the corresponding plots for the eastern lobe of the anticyclone were not shown in Fig. 5, as they were in Fig. 6? I would have thought that they would be relevant to the discussion here.

We didn't show the eastern mode of the anticyclone because the trajectories show similar results as for the western mode. We did not want to extend the paper to much. However, we decided to show also the plots for the eastern mode as shown in Fig. 2 (of this author comment) and Fig. 5 in the revised version of the manuscript to avoid the reader is confused why we didn't show the plots for the eastern mode.

P11, L14-26: What exactly is meant by 'substantial' upward transport (L14)? Does 'substantial' mean 0.5 K/day, 1 K/day, or?? It would be better to be more quantitative. In addition, here the discussion is cast in terms of heating rate (K per day), whereas Fig. 7 and Fig. A1 show the change in potential temperature (in K) along 20-day trajectories, making the reader do the (admittedly easy) math. Once

---

## Author Comment (AC2) · 23 Nov 2018

**Author Comment to Referee #2**

**ACP Discussions doi: 10.5194/acp-2018-724**
**(Editor - Peter Haynes)**
**'Lagrangian simulations of the transport of young air masses to the top of the Asian monsoon anticyclone and into the tropical pipe'**
* * *
We thank Referee #2 for further guidance on how to revise our paper. Following the reviewers advice we have elaborate the relation of our findings to previously published work and introduced an extended discussion of the presented results with respect to previous publications. Our reply to the reviewer comments is listed in detail below. Questions and comments of the referee are shown in italics. Passages from the revised version of the manuscript are shown in blue.

*The study "Lagrangian simulations of the transport of young air masses to the top of the Asian monsoon anticyclone and into the tropical pipe" investigates transport processes from the boundary layer to the monsoon anticyclone and further into the stratosphere by employing 3-D CLaMS simulations with mixing and additional backtrajectory data. Further, comparisons with satellite data (MIPAS) are included, which increase the confidence in the presented results. Overall, the manuscript is well written and the figures and analyses are well composed. Further, the study contains interesting results suitable for publication in ACP. Nevertheless, I think that the following comments need to be addressed before the manuscript can be published. In particular, the manuscript would benefit from (and in my opinion needs) an extended discussion of the presented results with respect to previous publications.*

**General Comments**

*In the abstract and in the text, you distille the transport processes from the boundary layer to the tropical stratosphere into 3 separate regimes. In my opinion, this result is in agreement with previous notions on transport of*

*ASM air massse, i.e. I think it is known that the upper troposphere in the Asian summer monsoon region is strongly affected by convection (e.g. Randel and Park, 2006, show a strong impact from convection at 350- 360K), slow upward movement within the anticyclone is addressed e.g. in Park et al. (2007, 2009; see also the schematic Fig. 14 in the latter publication). Further, Ploeger et al. (2017) show slow upward transport of Asian summer monsoon air masses in the tropical pipe (cf. also Randel et al. 2010) and also presents an overview of transport processes in the ASM region in its introduction. There are also other studies addressing transport in the Asian monsoon and some that also mention slow ascent in the UT in the monsoon region: e.g. Wright et al. (2011), Bergman et al. (2012) and Garny and Randel (2016). Nevertheless, it is indeed interesting to have a single study (and model) that shows all of the transport regimes and your study includes additional information. Please, relate your results to these previous findings or suggestions and carve out how your study differs/agrees with the processes described there. How do your results complement these previous suggestions/findings? Maybe you could commet also on the influence of extremely deep (or even overshooting) convection on air masses within the UTLS in the Asian monsoon region.*

Following the reviewer's advice we introduced an extended discussion of the presented results with respect to previous publications within the Discussion Section 4 in the revised version of the manuscript (see below). Further, we introduced in the Conclusion Section 5 additional references to better link to previous published work.

It is well known that the composition of the Asian monsoon anticyclone is strongly affected by convection over continental Asia (e.g. the south slope of the Himalayas and the Tibetan Plateau), Bay of Bengal, and the western Pacific, (e.g., Randel and Park, 2006; Park et al., 2007, 2009; Wright et al., 2011; Chen et al., 2012; Bergman et al., 2013; Fadnavis et al., 2014; Tissier and Legras, 2016). However there is a debate about the contribution of different source regions to the composition of the Asian monsoon anticyclone. Further, there are differences in the conclusions in the literature about the contribution of different source regions depending on the used reanalysis data (e.g., Wright et al., 2011; Bergman et al., 2013). Findings by Vogel et al. (2015) show that there is a strong intraseasonal variability of boundary source regions to the composition of the Asian monsoon anticyclone during

a particular monsoon season. We would like to emphasise that the trajectories presented in Fig. 6 demonstrating convection below 380 K are only a snapshot for 18 August 2018 with convection over the western Pacific and continental Asia mainly in the region of the south slope of the Himalayas and the Tibetan Plateau.

Here, in contrast to earlier studies, we focus on transport at the top of the anticyclone at altitudes greater than 380 K potential temperature ($\approx$100 hPa) reaching up to 460 K ($\approx$60 hPa). Further, in addition to previous studies (e.g., Garny and Randel, 2016; Ploeger et al., 2017), we relate the transport of air masses from inside the Asian monsoon anticyclone to air masses uplifted outside the anticyclone. Subsequently these air masses are jointly transported upwards to the top of the anticyclone at $\approx$460 K. ... (further see revised version of the manuscript).

We added in Sect. 3.3.1 the following discussion:

It is known that the radiative heating rates in the tropical UTLS are different in current reanalysis models (e.g., Wright and Fueglistaler, 2013) and are most likely overestimated in ERA-Interim (e.g., Ploeger et al., 2012; Schoeberl et al., 2012). Therefore, the rates of diabatic heating in the upward spiralling found in our study are most likely somewhat too high, however slow upward transport in the UTLS in the region of the Asian monsoon anticyclone associated with positive heating has been addressed previously (e.g., Park et al., 2007; Bergman et al., 2012; Garny and Randel, 2016; Ploeger et al., 2017).

Regarding the influence of extremely deep convection, note that in CLaMS convection is driven by ERA-Interim reanalysis data. We introduced the following paragraph within Section 2 to explain convection in CLaMS in more detail. Small-scale overshooting convection is not included in ERA-Interim, however the focus of our paper is to understand the main transport pathways at the top of the anticyclone higher than 380 K up to 460 K ($\approx$100-60 hPa) which is above the main level of tropical deep convection (e.g., Devasthale and Fueglistaler, 2010; Bergman et al., 2012).

The upward transport and convection in CLaMS (in both three-dimensional simulations as well as in trajectory calculations) is driven by ERA-Interim

reanalysis data in which changes are implemented to improve deep and mid-level convection compared to previous reanalysis data (Dee et al., 2011). However, small-scale rapid uplift in convective cores is not included. Therefore convection over Asia is most likely underestimated in ERA-Interim. However, the focus of our paper is to understand the main transport pathways at the top of the anticyclone greater than $380\,\text{K}$ and up to $460\,\text{K}$ ($\approx$100-60 hPa) which is above the main level of tropical deep convection (e.g., Devasthale and Fueglistaler, 2010; Bergman et al., 2012). Further, previous studies demonstrated that the vertical transport in CLaMS allows the spatio-temporal distribution of CO within the Asian monsoon anticyclone measured by the Aura Microwave Limb Sounder (MLS) to be reproduced (Vogel et al., 2015; Ploeger et al., 2017).

*Related to this issue, you state a convective regime and I wonder how convection is treated in your simulations and backward trajectory calculations. Please incorporate some notion on how the setup of your simulations/trajectories will affect your results.*

We revised this paragraph in Sect. 2 regarding convection in CLaMS as already stated above (see previous comment).

*P.8 L.31-31: "...how can air masses...?": you pose this question, however, to me it is not clear where it is anserwered. Are you thinking about inmixing from the outside to the inside/edge of the AC and subsequent vertical transport. Please connect to the parts in the text where this question is answered and/or e.g. repeat the question and give the answer to it in the conclusion. Would it be possible to include the transport of air masses from adjacent regions above 380K also in your Fig. 13?*

We agree it would be helpful to revisit to these questions posed within the introduction. Therefore, we refer to these two questions within the conclusions as follows.

Further between $420\,\text{K}$ and $460\,\text{K}$, highest contributions from young air masses are found around the edge of the anticyclone, indicating a spatially strongly inhomogeneous ascent in the monsoon with strongest ascent at the edge. The higher in the upward spiralling range the more air masses from the stratospheric background are mixed with the young air masses transported

upwards within this upward spiralling range. Thus in this paper, we could answer the question of what are the transport pathways of young air masses at the top of the Asian monsoon into the stratosphere by the concept of the upward spiralling range.

To answer our second question of how boundary layer source regions in Asia affect the composition of the middle stratosphere within the tropical pipe at $550\,K$, the transport times from the Earth's surface up to this level of potential temperature need to be taken into account. In a two-monsoon-season simulation...

Moreover, following the reviewer advice we added Fig. 1 (= Fig. 15 of the revised version of the manuscript) to our manuscript.

*Most of the analysis are focused on one day (18 August 2008), only. For some of the analyses this might not be important, however, other analyses might depend on the specific conditions (e.g. the split of the anticyclone) during that date/period as for example the trajectory analysis in Fig. 5. Please include some additional discussion regarding that issue. Partly, you have already addressed this issue, e.g. to complement Fig. 7 you additionally include Figure A1. I would guess that in particular the backward trajectories results are affected by the choice of the starting date and might vary throughout the monsoon season. This issue also extends to the comparison with MIPAS data and to the inferred transport on the eastern/western side of the anticyclone.*

We agree that the most of the analysis is focused on 18 August 2008 as a case study. However, the results of the 3-dimensional CLaMS simulation for 18 August 2008 is a result of the interplay between convection, large-scale upward transport (driven by radiative heating), and the anticyclonic flow in the UTLS during the last weeks of the simulation. The same is true for the 40-day and 20-day backward trajectories as well as for the MIPAS measurements. Thus our results are representative for August 2008. To give a broader view, we already include 20-day backward trajectories showing different days during the monsoon season 2008 within the Appendix.

*Fig. 8 in Garny and Randel (2016) shows kinematic and diabatic vertical velocities and Fig. 12 a) in Park et al. (2007) shows pressure tendencies.*

*These figures show ascent on the eastern side of the anticyclone and descent on the western side at levels close to (but still mostly below) the tropopause. How do your statements and your Fig. 10 relate to that? How does the climatological picture of Fig. 10 look like? Is there always (i.e. on a climatological basis) stronger heating above the western side of the anticyclone above the tropopause but cooling below? How is it on the eastern side?*

Fig. 10 (= Fig. 11 of the revised version of this manuscript) shows in agreement with Garny and Randel (2016) and Park et al. (2007) downward transport (negative radiative heating) below ≈360 K in the western mode of the anticyclone and upward transport (positive radiative heating) above ≈360 K in the eastern mode on 18 August 2008. Fig. 8 in Garny and Randel (2016) shows monthly mean vertical velocity for July 2006 at 360 K using ERA-Interim data. Fig. 12a in (Park et al., 2007) shows July–August average ERA40 vertical velocity for 2000-2002 at 104 hPa. Further, also Fig. 10a in Pan et al. (2016) confirm ascent on the eastern side of the anticyclone and descent on the western side showing June–July–August vertical velocity from WACCM4-SD at 100 hPa for 2014. Thus, ascent in the eastern side of anticyclone and descent on the western side in the upper troposphere is a common feature found in our case study for the 18 August 2008 as well as in a more climatological picture reported in the literature (Park et al., 2007; Garny and Randel, 2016; Pan et al., 2016).

The focus of our study is to demonstrate that in the upward spiralling range (above 360 K) a slow upward transport is found over the region of entire anticyclone (west and east mode) with diabatic heating rates of up to 1–1.5 K inferred from ERA-interim. Our 40-day backward trajectory calculations (see Fig. 6 and 7 of the revised version of this manuscript) demonstrate that a diabatic heating above 360 K is found at both the western and eastern side of the anticyclone during August 2008. A broader climatological analysis about differences in heating rates between the western and eastern side of the anticyclone above the tropopause would be an additional project and is therefore not included in this paper.

*Additionally, I think it would be very helpful if you relate the results of your tracer pulses shown in Figs. 8 and 9 to the results in Ploeger et al. (2017). In particular with respect to transport of air masses from the anticyclone to the deep stratosphere.*

We extended the discussion regarding the paper by Ploeger et al. (2017) within the Discussion Section 4 as follows below. Our results agree in general with findings by Ploeger et al. (2017), however in our paper in addition the contribution of the tropical adjacent regions to the tropical pipe are quantified. Further, the tracer approach is different between Ploeger et al. (2017) and our paper.

In Ploeger et al. (2017) the anticyclone tracer is initialised with unity inside the PV contour enclosing the anticyclone core in the 370-380 K layer on each day during July-August of the years 2010-2013 and is advected as an inert tracer during the following year. On 1 July of the year thereafter, the tracer is set to zero everywhere and is then reinitialised for the following monsoon season. Thus, the anticyclone tracer is set to zero during the monsoon season in July.

In our paper, the boundary emission tracers are released within the model boundary layer each day during the course of the two-monsoon-season simulation from 1 May 2007 until 1 November 2008. Thus, the transport from the troposphere through the Asian monsoon anticyclone into the tropical pipe is continuously covered over two succeeding years.

It has been proposed that the Asian monsoon constitutes an effective transport pathway from the surface, through the Asian monsoon anticyclone, and deep into the tropical pipe based on satellite observations of hydrogen cyanide (HCN) (Randel et al., 2010). HCN is a tropospheric pollutant produced mainly by biomass burning with a strong sink on ocean surfaces. Therefore tropical ocean regions cannot be the source for HCN found in the tropical pipe. Ploeger et al. (2017) addressed this issue using CLaMS simulations marking air masses within the Asian monsoon anticyclone by a PV-gradient criterion (Ploeger et al., 2015). They find that the air mass fraction from the anticyclone correlates well with satellite measurements of HCN within the tropical pipe.

In our study, we found a similar behaviour for contributions of the India/China tracer within the tropical pipe as Randel et al. (2010) for HCN and Ploeger et al. (2017) for the simulated anticyclone air mass fraction. Ploeger et al. (2017) found a maximum anticyclone air mass fractions around 5% in the tropical pipe using 3-dimensional CLaMS simulations for 2010-2013. This is consistent to our simulations finding about 6 % contributions of the India/China tracer within the tropical pipe at 550 K in 2008.

However in addition to Randel et al. (2010) and Ploeger et al. (2017), we

show that the contributions from emissions from Southeast Asia and the tropical Pacific during summer are larger than the contribution from India/China within the tropical pipe. This demonstrates that the Asian monsoon anticyclone is a more effective transport pathway for the tropical adjacent regions than for air masses from inside the anticyclone itself (India/China). From the tropical adjacent regions air masses can be transported to the edge of the Asian monsoon anticyclone and then further into the tropical pipe.

**Specific comments**

*P1 L19-20: Regarding the effectiveness of horizontal mixing and vertical transport, Garny and Randel (2016) seem to come to a different conclusion. Please discuss (e.g. in Sect. 4) how your results agree and differ. In case of the latter please also discuss why they differ.*

Many thanks for this comment. This shows that we have to be more precise in our formulations to avoid any misunderstandings. Our statement on P1 L19-20 is related to the vertical transport of air masses into the tropical pipe compared to the transport from the monsoon anticyclone into the northern extratropical lower stratosphere. In contrast, Garny and Randel (2016) compares the difference of vertical transport into the lower tropical stratosphere in the anticyclone directly above the tropopause with the isentropic transport from the monsoon anticyclone into the northern extratropical lower stratosphere. Garny and Randel (2016) found only a few percent 3%/8% ($360\,K/380\,K$) by isentropic transport, however 15% of trajectories from the Asian monsoon anticyclone reach the northern extratropical lower stratosphere after 60 days. Most of them by upward transport into the tropical stratosphere and subsequent transport into the northern extratropical lower stratosphere. In Vogel et al. (2018), we found a contribution of the India/China tracer up to 16% (at $380\,K$) in the northern extratropical lower stratosphere during fall 2008. The transport of the India/China tracer within the 3-dimensional CLaMS simulations includes both transport pathways: direct isentropic transport as well as vertical transport in the region of the anticyclone into the tropical stratosphere and subsequent northward transport. Therefore, the value of 15% in Garny and Randel (2016) is comparable with the value of 16% (at $380\,K$) in Vogel et al. (2018) and therefore in good agreement.

We revised the abstract as follows.

In the upward spiralling range, air masses are uplifted by diabatic heating across the (lapse rate) tropopause, which does not act as a transport barrier under these conditions. Further, in the upward spiralling range air masses from inside the Asian monsoon anticyclone are mixed with air masses convectively uplifted outside the core of the Asian monsoon anticyclone in the tropical adjacent regions. Further, the vertical transport of air masses from the Asian monsoon anticyclone into the tropical pipe is weak in terms of transported air masses compared to the transport from the monsoon anticyclone into the northern extratropical lower stratosphere. Air masses from the Asian monsoon anticyclone (India/China) contribute a minor fraction to the composition of air within the tropical pipe at $550\,\mathrm{K}$ (6%), the major fractions are from Southeast Asia (16%) and the tropical Pacific (15%).

The paragraph related to the issue within the Discussion Section 4 is revised as follows.

Vogel et al. (2016) performed a CLaMS simulation for the year 2012 using similar tracers of air mass origin as in this work and found a flooding of the northern extratropical lower stratosphere with young air masses from the region of the Asian monsoon anticyclone. The transport of young air masses (age $< 6\,\mathrm{months}$) into the northern extratropical lower stratosphere is calculated, resulting in up to 44% at $360\,\mathrm{K}$ (up to 35% at $380\,\mathrm{K}$) end of October 2012, with the highest contribution from India/China up to 15% (14%) (see Fig. 14 in Vogel et al. (2016)). Here, the same analysis is performed for the simulation for 2008 and a slightly higher impact on the northern extratropical lower stratosphere is found for the year 2008, up to 48% young air at $360\,\mathrm{K}$ (up to 41% at $380\,\mathrm{K}$) end of October 2008 and up to 18% (16%) from India/China compared to 2012 (see Appendix B). This difference is most likely caused by the interannual variability of the monsoon system. However, within the tropical pipe at $550\,\mathrm{K}$, in 2008 the contributions from India/China are about $6\,\%$, demonstrating that the transport of air masses from the Asian monsoon anticyclone into the northern extratropical lower stratosphere during boreal summer and fall is more effective than the vertical transport into the tropical pipe during the course of one year. This is consistent with Ploeger et al. (2017), who found maximum anticyclone air mass fractions around 5%

in the tropical pipe and 15% in the northern extratropical lower stratosphere using 3-dimensional CLaMS simulations for 2010-2013. In a study releasing trajectories within the Asian monsoon anticyclone, Garny and Randel (2016) found a similar values of 15% of trajectories released in the anticyclone that reach the northern extratropical lower stratosphere after 60 days (for 2006).

*P3 L7-10: Regarding the connection of El Nino and La Nina with the Indian summer monsoon. You argue that 2008 was (in terms of rainfall) normal because of La Nina in the winter before, although, in the previous sentence you claim that El Nino and La Nina events tend to be connected to unusual rainfall in the following Indian summer monsoon season. This seems contradictory to me! Further, Kumar et al. (2006) show a relation of concurrent SSTs with rainfall in India (during a quick search I could not find that they are stating a connection with previous winter SSTs). Also, in Webster et al. (1998) I could not easily find to which SST anomaly they refer, i.e. previous/following winter or concurrent summer. Please comment on this and revise if necessary.*

Thanks for the comment, we revised these sentences as follows and introduced a further reference to the connection between ENSO to Indian rainfall.

Further, in 2008 there was a normal monsoon season in terms of normal rainfall over India in summer 2008[1]. It is established that the Indian monsoon is influenced by the El Niño Southern Oscillation (ENSO) (e.g., Kumar et al., 2006). There is evidence that a strong La Niña in winter (e.g. 2007/08 (DJF) according to the Oceanic Niño Index[2]) in combination with La Niña conditions during the subsequent summer (as in 2008) is correlated with normal rainfall over India with a certain variability in precipitation between different Indian regions (e.g., Chakraborty, 2018).

*P5 L3-5: Is the sum of the tracers for all parcels in the boundary layer really always equal to 1 as you describe on page 5 L3-4. What if unmarked parcels from above the BL are transported into the BL? Are they removed? Otherwise, they might not be marked in the BL as marking takes place every 24h, only. Does the time step of 24h release play an important role? As*
* * *
[1] see e.g. http://www.tropmet.res.in/~kolli/mol/Monsoon/Historical/air.html
[2] see e.g. http://ggweather.com/enso/oni.htm

*an example, Bergman et al (2013) use backward- trajectories started every 6 hours. Why don't you mark/emmit the tracer "continuously", i.e. at every time step of the simulation? Is there a scientific or technical reason for this setup*

Thanks for the comment, in fact we have to be more precise at this point and revised the sentence as follows. We adapted also Table 1 by introducing the emission tracer for the background.

Within the model boundary layer, the sum of all the different emission tracers ($\Omega_i$) including the emission tracer for the background (remaining surface) is equal to 1 ($\Omega = \sum_{i=1}^{n} \Omega_i = 1$, see Table 1) .

Air parcels in the free troposphere and stratosphere are not unmarked. They are marked with 'zero' in contrast to air parcels in the model boundary that are marked with 'one'. If air parcels from the free troposphere are transported downward into the model boundary layer, they will be overwritten by the boundary conditions every 24 hours. The setting of the emission tracers is adjusted to the mixing in CLaMS which is every 24 hours. The mixing in CLaMS is coupled to the integral deformations in the flow over the time step of transport. The critical deformation parameter $\lambda_c$ can also be expressed in terms of a critical Lyapunov exponent $\gamma_c$ (with $\gamma_c = \lambda_c \times \Delta t$) which depends on the advection times step ($\Delta$t). The mixing procedure in CLaMS is optimised using a $\gamma_c$ equal to 1.5 for a $\Delta t = 24$ h (Konopka et al., 2007), therefore we use here a time step of 24 hours to set the boundary emission tracers. It is possible that an air parcel from the free troposphere is transported downwards into the model boundary layer and subsequently upwards out of the model boundary layer into the free troposphere within a time period lower than 24 hours without mixing. In that case the air parcel is not marked by an emission tracer. However, we think the impact of this issue is small compared to the uncertainties of the trajectory calculations itself at the lowest model levels.

*P6 L1: Please add whether the trajectories described in this section are calculated using heating rates (as I would assume) or kinematic vertical velocities.*

Yes, we use heating rates. We revised the sentence to be more precise as follows. Moreover, we added some further information to the vertical transport

in the model within the general CLaMS description in Sect. 2.0 (see above).

Within this study, 20-day and 40-day backward trajectories are calculated driven by wind data (with a horizontal resolution of $1° \times 1°$) from the ERA-Interim reanalysis (Dee et al., 2011) and using the diabatic approach to analyse the transport pathways of air parcels at the top of the Asian monsoon anticyclone and beyond into the tropical pipe.

*P7 L6-9: Has the same method for interpolating MIPAS HCFC-22 data been used in Vogel et al. 2016? Then you could add a note so it is clear.*

The same synoptic interpolation of MIPAS HCFC data has been used in Vogel et al. (2016) (see Fig 13). However, in Fig. 13a in Vogel et al. (2016) three-monthly mean values of HCFC for July, August and September 2008 are shown in contrast to the Figures in Vogel et al. (2018). In Vogel et al. (2018), MIPAS HCFC-22 data are shown synoptically interpolated for 18 August 2018 (see Fig. 3 and Fig. 11). Because of this difference, we think here it it better to make no reference to Vogel et al. (2016) to avoid any misunderstanding.

*P8 L16 and following as well as P9 L26-27: Either in the description of Fig. 2 2nd row and Fig. 3 2nd row left or in the discussion you should draw a relation to Pan et al. (2016), who showed that upward transport (e.g. of CO) is mainly focused on the eastern side of the AC.*

We added the following sentence in Sect. 3.1.1 to the discussion of Fig. 2 (2nd row):

The horizontal transport of air masses from the eastern to the western mode of the anticyclone indicated by the India/China tracer is consistent with simulations of carbon monoxide (CO) using the Whole-Atmosphere Community Climate Model (WACCM4-SD) (Pan et al., 2016).

*P10 L13: Please state that you are starting trajectories only on 18 August 2018 for the analyses in Sect. 3.2.1. Or have you analysed other dates as well?*

In our paper (Vogel et al., 2018), 40-day backward trajectories started on

18 August 2018 are only presented. We add the date (see below). For your information, we also performed for other days 40-day backward trajectories with similar results as for 18 August 2018. However, we decided to only present the results for the 18 August 2018 as a case study.

To analyse the transport pathways to the top of the anticyclone in more detail, 40-day backward trajectories are calculated starting in the western (20–50°N,0–70°E) and eastern (20–50°N,70–140°E) modes of the anticyclone. The trajectories are started at the position of the air parcels from the 3-dimensional CLaMS simulation at different levels of potential temperature ($\Theta = 380, 400, 420, 440\,\mathrm{K}$ $\pm 0.25\,\mathrm{K}$) on 18 August 2018. Note that the air parcels in the 3-dimensional CLaMS simulation are distributed on an irregular grid. To take into account the distribution of the boundary emission tracer at the top of the Asian monsoon anticyclone, only air parcels are selected with contributions of young air masses (age $< 6$ months, Summer 08) larger than 70% (380 K), 50% (400 K), 20% (420 K), and 5% (440 K) (not all levels of potential temperature are presented here). The percentages are chosen in a way to obtain a number of trajectories (less than 30) that can be reasonably visualised. The results of the 40-day backward trajectories are similar at different levels of potential temperature; therefore we show a selection of trajectories to demonstrate the main transport pathway to the top of the Asian monsoon. A larger set of 20-day backward trajectories analysed statistically will be discussed below in Section 3.2.2.

*P10 L22-23: How do you know that the transport occurs above the Tibetan Plateau? From Fig. 5 only the longitudinal range is visible but not where in latitude the parcels ascend. If you have made additional analyses to check that they are indeed from the Tibetan Plateau just note that you have analysed this but chose to not include a figure or the analysis here.*

Fig. 2 (of this author comment), shows the location of the strongest updraft along the 40-day backward trajectories shown in Fig. 5 of the revised version of the manuscript. There is a cluster of trajectories in the region of the south slope of Himalayas and the Tibetan Plateau as well as in the western Pacific. We revised the sentence as follows.

Our 40-day backward trajectories show that preferred regions for fast uplift are continental Asia (mainly the region of the south slope of Himalayas and

the Tibetan Plateau) and the western Pacific (not shown here).

*P10 L32: At some instances (e.g. here at P10 L32) you refer to inside or outside the anticyclone but do not give a reliable definition or state what you consider as inside or ouside. Would it be an option to include PV contours for that purpose? Also on P11 L5 you should probably rephrashe to "entire Asian monsoon region" because you do not start only within the anticyclone.*

It is known that the Asian monsoon anticyclone has a strong horizontal transport barrier at about 380 K (e.g., Ploeger et al., 2015), however this transport barrier is missing at higher levels of potential temperature. Therefore, it is difficult to define inside/outside the anticyclone for all levels above 380 K. Here we use the emission tracer for India/China is a proxy for the location and shape of the Asian monsoon anticyclone as introduced as follows in the manuscript.

Vogel et al. (2015) showed that the emission tracer for India/China is a good proxy for the location and shape of the Asian monsoon anticyclone using pattern correlations with potential vorticity (PV), and MLS $O_3$ and CO satellite measurements between 360 K and 400 K. Therefore here we use the India/China tracer as proxy for the location of the anticyclone.

*P11 L14: Maybe you should rephrase this part stating "At 380..." instead of "Above 380K,..." because at 400K the structures are not as inhomogeneous anymore and above 400K there is also considerable upward transport in the tropics.*

As proposed we revised this paragraph as following including also comments by reviewer #1.

Above 360 K, air parcels that experienced strong upward transport larger than 20–30 K within 20 day (corresponding to a mean value of 1–1.5 K per day) are largely found in the region of the anticyclone. This rate of upwelling is much slower compared to convective upwelling shown at 360 K. Air parcels that experienced strong upward transport are mainly grouped in curved elongated filaments, reflecting a rotating movement of the air parcels at the top of the anticyclone. Often air parcels with strong $\Delta\Theta$ above 360 K are located more at the edge of the eastern and western modes of the anticyclone

and at the edge of the eastward-migrating eddy at the eastern flank of the anticyclone. Thus the upward transport in the region of the anticyclone is inhomogeneous and not homogeneously distributed over the entire anticyclone as suggested from climatolgical studies (e.g., Randel et al., 2010; Ploeger et al., 2017). This is consistent with results presented above in Sect. 3.2.1 demonstrating that for single selected trajectories the transport at the top of the Asian monsoon anticyclone is a slow upward transport of about 1–1.5 K per day in a large-scale spiral above the anticyclone caused by diabatic heating. In the backward trajectory calculations mixing processes are not included, however the results of the trajectory calculations are consistent with patterns found in the 3-dimensional CLaMS simulation including mixing as discussed in Sect. 3.1.3, demonstrating that young air masses above 400 K are found at the edge of the anticyclone. Above 400 K, air masses in the tropics also experienced upward transport, but the vertical uplift is in general lower than 20 K within 20 day, (i.e. lower than 1 K per day).

*P. 12 L14 and L17: Two times 25% instead of 15% is mentioned, as I assume would be correct. If I am correct, the 25% are the contribution of the winter pulse (W07) at 450K, right?*

Yes, we agree. However, we changed Fig. 9 (of the manuscript) showing TAR instead of SEA. For TAR 25% is correct. We changed this sentence as follows (see Figure 9 in the revised version of the manuscript.

Many thanks for this comment. 15 % is correct. We corrected the percentages in the manuscript.

*P14 L6-8: Do you really mean "Asian monsoon air masses from the anticyclone" or rather air masse from your India/China tracer? I think your findings show the claimed relation only for the latter.*

We agree that the India/China tracer and air masses from the anticyclone are not the same, however we found in Vogel et al. (2015), that India/China tracer is a good proxy for the location and shape of the Asian monsoon anticyclone using pattern correlations with potential vorticity (PV), and MLS $O_3$ and CO satellite measurements as explained in Sect. 2.1 in Vogel et al. (2018). We clarified the sentence as follows:

Further, our findings show that air masses from India/China, thus mainly from the Asian monsoon anticyclone, contribute to a smaller fraction of the composition of air within the tropical pipe at 550 K; the major part is from Southeast Asia and the tropical Pacific.

*P14 L30-31: I think slow upward transport has been proposed earlier (see my general comment). Please clarify if you are referring to some specific point of the upward transport process that was not published earlier.*

We revised the Discussion Section 4 as discussed above and clarified that our focus is the relation of transport of air masses from inside the Asian monsoon anticyclone to air masses uplifted outside the anticyclone in an altitude range higher than 380 K potential temperature ($\approx$100 hPa) up to 460 K ($\approx$60 hPa).

Here, in contrast to earlier studies, we focus on transport at the top of the anticyclone at altitudes greater than 380 K potential temperature ($\approx$100 hPa) reaching up to 460 K ($\approx$60 hPa). Further, in addition to previous studies (e.g., Garny and Randel, 2016; Ploeger et al., 2017), we relate the transport of air masses from inside the Asian monsoon anticyclone to air masses uplifted outside the anticyclone. Subsequently these air masses are jointly transported upwards to the top of the anticyclone at $\approx$460 K.

Further we added in Sect. 3.3.1 the following discussion.

It is known that the radiative heating rates in the tropical UTLS are different in current reanalysis models (e.g., Wright and Fueglistaler, 2013) and are most likely overestimated in ERA-Interim (e.g., Ploeger et al., 2012; Schoeberl et al., 2012). Therefore, the rates of diabatic heating in the upward spiralling range found in our study are most likely somewhat too high, however slow upward transport in the UTLS in the region of the Asian monsoon anticyclone associated with positive heating has been addressed previously (e.g., Park et al., 2007; Bergman et al., 2012; Garny and Randel, 2016; Ploeger et al., 2017).

*I think it would be good to label all panels of all figures with (a), (b), (c) and so on as you do for example in Fig. 3 but not in Figs. 4, 5 etc. This is just a suggestion, but would definitely help to increase the readability. Then you could refer directly to the individual panels of the figures and it would be*

*consitent throughout the manuscript.*

done

*Also, consider to add additional references to the individual panels in the text when you draw a conclusion or describe something that is based on the respective panel.*

done

**Minor suggestions/corrections:**

1. *P1 L11: Either change to "Second, these air masses..." or "Second, air masses are uplifted within the anticyclone..." or something similar.*

   done

2. *P1 L14: As before, maybe clarify by changing your sentence to something like: "Third, transport of air masses affected by the Asian monsoon (anticyclone)..." or something similar.*

   done

3. *P2. L1: This probably needs some additional restriction to where the the Asian monsoon is the "most pronounced circulation pattern". Do you refer here to the tropospheric flow or the UTLS anticyclone?*

   We revised the sentence as follows:

   The Asian summer monsoon is associated with deep convection over the Indian subcontinent and is the most pronounced circulation pattern in boreal summer with an anticyclonic flow that extends from the upper troposphere into the lower stratosphere (UTLS) region (e.g., Li

et al., 2005; Randel and Park, 2006; Park et al., 2007).

4. *P2 L21-22: Order references according to year of publication.*

   done

5. *P3 L1: Would it be better to change "defined regions" to "specific regions"?*

   We prefer 'defined regions'.

6. *P3 L2: Shouldn't this read: "covering Earth's entire surface". Then it would need to be changed throughout the manuscript.*

   We don't think so.

7. *P3 L32-33: Maybe change to "...a total simulation period of 18 months)."*

   done

8. *P4 L2-6: The two sentences starting with "With this approach..." and "This model setup..." seem somehow repetitive. If they are not, please try to clarify.*

   done

9. *P9 L4-5: Repetition of "in particular". Please rephrase.*

   done

10. *P13 L6: "exists" should be "exist". Also consider to rephrase, e.g. to "... pathways exist. On these horizontal pathways, air masses are transported isentropically..."*

done

11. *P13 L10: I would suggest to shift the first sentence of the paragraph ("On 18 August....") behind the current second sentence ("To analyse...") or/and adapt as it seems to be doubled at the moment.*

no

12. *P13 L33: Probably this should be "...from the tropical...".*

done

13. *P14 L13: Are "Asian monsoon anticyclone" and "Asian monsoon" switched here?*

done

14. *P32: In the caption of Fig. 8 it should state "...(1 May 2007 - 18 August 2008)..." instead of "...(1 May 2007 - 31 October 2008)...", because you show the tracer distribution on 18 August 2008. This is also how you describe the figure in the text.*

done

[Figure]

Figure 1: Longitude-theta cross section at 30°N: At the top of the Asian monsoon anticyclone (above ≈360 K) air masses circulate around the anticyclone in a large-scale upward spiral extending from northern Africa to the western Pacific. In the upward spiralling range air masses from inside the Asian monsoon anticyclone (shown in blue) are mixed with air masses convectively uplifted outside the core of the Asian monsoon anticyclone in the tropical adjacent regions e.g. uplifted by tropical cyclones in the western Pacific ocean (shown in red). The higher above the thermal tropopause the larger is the contribution of air masses from outside the Asian monsoon anticyclone from the stratospheric background coming into the upward spiralling flow (shown in green). The levels of pressure are marked by thin white lines and the thermal tropopause is shown by black dots.

[Figure]

Figure 2: The geographical position of the strongest updraft along the 40-day backward trajectories in the western and eastern mode of the anticyclone (started at 380 K) shown in Fig. 5 of the revised version of the manuscript.

---

## Author Comment (AC3) · 23 Nov 2018

**Author Comment to Referee #3**

ACP Discussions doi: 10.5194/acp-2018-724
(Editor - Peter Haynes)
**'Lagrangian simulations of the transport of young air masses to the top of the Asian monsoon anticyclone and into the tropical pipe'**
* * *
We thank Referee #3 for further guidance on how to revise our paper. Following the reviewers advice we have elaborated the relation of our findings to previously published work regarding the 'longstanding debate' and introduced an extended discussion of the presented results with respect to previous publications. Our reply to the reviewer comments is listed in detail below. Questions and comments of the referee are shown in italics. Passages from the revised version of the manuscript are shown in blue.

*The manuscript 'Lagrangian simulations of the transport of young air masses to the top of the Asian monsoon anticyclone and into the tropical pipe' investigates transport pathways in the monsoon region from the boundary layer into the stratosphere. The authors use both Lagrangian backward trajectory calculations and three-dimensional simulations including irreversible mixing with the Lagrangian transport model CLaMS. Artificial tracers of air mass origin are compared to measurements of chlorodifluoromethane (HCFC-22; CHClF2) by the Michelson Interferometer for Passive Atmospheric Sounding (MIPAS). The methods are similar to those in Vogel et al. (2016) but in addition to horizontal transport they address vertical transport pathways out of the Asian monsoon anticyclone and subsequent upward transport into the lower stratosphere. The chosen period is a normal monsoon season in terms of medium rainfall over India in summer 2008. The paper presents an interesting description of the 'spiralling staircase'. Consistency with MIPAS data supports the model results. The paper is well written and I support publication after a few comments have been addressed. The heavy focus on a particular day is didactical, but the analysis of different meteorological situations would provide stronger evidence. In certain passages the phrasing could be improved in order to more clearly distinguish what is the particular contribution of this*

*study to the 'longstanding debate about the transport mechanisms at the top of the Asian monsoon anticyclone and beyond into the stratosphere'. The debate is mentioned again in the discussion but the different arguments about 'the exact transport mechanism' could be more clearly stated. Otherwise it is difficult to distinguish if some very general assertions have been stated before in the literature or are novel to this work. The figures could be improved and the description thereof clarified in the text.*

We revised the paper following the referee's advice and added an extended discussion section to the revised manuscript to point out the relation of our findings to the existing literature.

**Minor comments:**

1. *p1 l19 'However, this upward transport': Make 'this' more clear (the upward spiralling range).*

    We revised this part in the abstract as follows.

    Moreover, the vertical transport of air masses from the Asian monsoon anticyclone into the tropical pipe is weak in terms of transported air masses compared to the transport from the monsoon anticyclone into the northern extratropical lower stratosphere.

2. *p2 l20 Briefly mention the sides in the debate and state where the authors stand.*

    We revised this part in the introduction as follows.

    The Asian monsoon circulation provides an effective pathway for tropospheric trace gases such as pollutants, gaseous aerosol precursors, as well as aerosol particles into the lower stratosphere which could play an important role in the formation of ATAL layer (e.g., Vernier et al., 2015, 2018; Höpfner et al., 2016; Brunamonti et al., 2018). There is also export of monsoon air quasi-isentropically out of the monsoon and

a certain fraction of monsoon air may reach greater altitudes in the stratosphere. There is a longstanding debate about the transport mechanisms at the top of the Asian monsoon anticyclone and beyond into the stratosphere (e.g., Bannister et al., 2004; Park et al., 2009; Randel et al., 2010; Bergman et al., 2012, 2013; Randel and Jensen, 2013; Uma et al., 2014; Orbe et al., 2015; Garny and Randel, 2016; Tissier and Legras, 2016; Ploeger et al., 2017). In the literature different aspects of the complex interplay between convection, large-scale upward transport (driven by radiative heating), and the anticyclonic flow in the UTLS are highlighted. Randel et al. (2010) pointed out that the monsoon circulation provides an effective pathway for pollution from Asia to enter the global stratosphere. Vertical upward transport into the deep stratosphere occurs within the tropical pipe, where tropical air masses are isolated to some extent from isentropic mixing with mid-latitude air (e.g., Plumb, 1996; Volk et al., 1996). Bourassa et al. (2012) analysed the eruption of the Nabro volcano in northeastern Africa and reported that the volcanic aerosol enhancement from the Nabro eruption was not injected directly into the stratosphere. They conclude that volcanic aerosol only attained stratospheric altitudes through subsequent transport processes associated with deep convection in the region of the Asian monsoon anticyclone. Pan et al. (2016) highlight that the Asian monsoon anticyclone is an isolated 'bubble' of tropospheric air above the global mean tropical tropopause that isentropically sheds tropospheric air into the stratosphere. Further, they argue that the vertical transport of Asian monsoon air into the deep stratosphere is inefficient during summer..

3. *p5 l3 $\zeta$ < 120K? p5 l 17 'is quantified.' What will be the quantitative measure?*

In CLaMS a pressure-based coordinate system ($\sigma$ coordinates) is used for pressure levels greater than 300 hPa with a hybrid vertical coordinate ($\zeta$) (for more details, see Konopka et al., 2012; Pommrich et al., 2014). The model boundary layer is defined $\approx$ 2–3 km above the surface following orography corresponding to $\zeta$ < 120 K. The emission tracers are set in the model boundary layer every 24 hours and are transported (advection and mixing) into the free troposphere and stratosphere during the course of the simulation. The fraction of different emission

tracers of an air parcel, e. g. in the tropical pipe, is a measure to quantify the transport from the source region into the tropical pipe during the course of the simulation.

4. *p6 l8 Dee et al. (2011) already cited in the first reference to ERAI in p4 l11*

Yes, Dee et al. is cited twice. However, for clarification we think it is good to cite once again Dee et al. in Sect. 2.2.

5. *p7 l20 make more explicit the dates of the monsoon and the pulse releases to clarify the 6 month age.*

We revised the sentence as follows.

To analyse the transport pathways at the top of the Asian monsoon anticyclone during the monsoon season 2008, we use only the tracers of air mass origin for the time pulse for Summer 08 (started on 1 May 2008 until end of October 2008).

6. *l22 'have strong variability from day to day' please rephrase*

The geographic position and shape of the Asian monsoon anticyclone show a strong day-to-day variability.

7. *p8 l1 mention the 360 theta level before in the text to streamline the reading*

Thanks for the this comment. We agree that the 360 K level was introduced too abruptly. We added the following paragraph at the beginning of Sect. 3.1.1.

It is known that the Asian monsoon anticyclone has a strong horizontal transport barrier at about $380\,\mathrm{K}$ (e.g., Ploeger et al., 2015), however this transport barrier is not well defined at higher levels of potential temperature. The less strong transport barrier at higher levels has consequences on the vertical transport at the top of the anticyclone. Before the transport at the top is discussed we show the horizontal distribution of different emission tracers at $360\,\mathrm{K}$ and then their subsequent transport to the top of the anticyclone up to $460\,\mathrm{K}$. Vogel et al. (2015) showed that the emission tracer for India/China is a good proxy for the location and shape of the Asian monsoon anticyclone using pattern correlations with potential vorticity (PV), and MLS $O_3$ and CO satellite measurements between $360\,\mathrm{K}$ and $400\,\mathrm{K}$. Therefore here we use the India/China tracer as a proxy for the location of the anticyclone.

8. *p8 l10 'simulated horizontal gradients': is there an objective metric or visual inspection? How is the top of the asian monsoon precisely defined?*

   We removed this statement about tracer gradients. The objective metric is the tracer distribution of air masses released by the time pulse for Summer 08 (India/China tracer and tropical adjacent regions). A precise definition of the top of the Asian monsoon does not exist in the literature. In our simulations we use the contribution of the emission tracers for India/China and the tropical adjacent regions to infer the top of the Asian monsoon.

9. *p8 l17 Fig. 2 (3rd row), to be consistent with previous paragraph.*

   done

10. *p10 l15 When are the trajectories started? How many? What release pattern? It would help for understanding what was done to state this clearly in the text. 40 days could be analysed statistically, bur for individual trajectories is a little bit too long.*

We agree that the description of the initialisation procedure of the trajectories is a bit short. The 40-day backward trajectories are presented to illustrate the main transport pathways. A more statistical analysis is presented in Sect. 3.2.2 for global 20-day backward trajectories. We revised this paragraph as follows.

To analyse the transport pathways to the top of the anticyclone in more detail, 40-day backward trajectories are calculated starting in the western (20–50°N,0–70°E) and eastern (20–50°N,70–140°E) modes of the anticyclone. The trajectories are started at the position of the air parcels from the 3-dimensional CLaMS simulation at different levels of potential temperature ($\Theta = 380, 400, 420, 440$ K $\pm 0.25$ K) on 18 August 2018. Note that the air parcels in the 3-dimensional CLaMS simulation are distributed on an irregular grid. To take into account the distribution of the boundary emission tracer at the top of the Asian monsoon anticyclone, only air parcels are selected with contributions of young air masses (age < 6 months, Summer 08) larger than 70% (380 K), 50% (400 K), 20% (420 K), and 5% (440 K) (not all levels of potential temperature are presented here). The percentages are chosen in a way to obtain a number of trajectories (less than 30) that can be reasonably visualised. The results of the 40-day backward trajectories are similar at different levels of potential temperature; therefore we show a selection of trajectories to demonstrate the main transport pathway to the top of the Asian monsoon. A larger set of 20-day backward trajectories analysed statistically will be discussed below in Section 3.2.2.

11. *p10 l20 describe the panels of fig 5 in the text.*

We revised the text as follows. Note that Fig. 5 from the ACPD version of the manuscript is Fig. 6 in the revised version.

Fig. 6 shows trajectories started in the eastern and western part of the Asian monsoon anticyclone around the thermal tropopause at 380 K on 18 August 2018. Air masses are uplifted to approximately 360 K very rapidly by various convective events occurring at different times and locations. Our 40-day backward trajectories show that preferred regions

for fast uplift are continental Asia (mainly the region of the south slope of Himalayas and the Tibetan Plateau) and the western Pacific (not shown here). A lower fraction of trajectories originates in the free troposphere. The trajectories in Fig. 6 demonstrating convection below 380 K are only a snapshot for 18 August 2018. There are several previous studies (e.g., Randel and Park, 2006; Park et al., 2007, 2009; Wright et al., 2011; Chen et al., 2012; Bergman et al., 2013; Fadnavis et al., 2014; Tissier and Legras, 2016) quantifying the contribution of different source regions to the composition of the Asian monsoon anticyclone during the course of the monsoon season (see discussion in Sect. 4).

12. *p10 last paragraph. At this point the reader would feel satisfied with a statistical analysis of a larger number of days to support the case study results. Maybe some additional results such as those later presented in A1 could be mentioned.*

   As mentioned above a statistical analysis of 20-day backward trajectories for the 18 August 2018 is presented in Sect. 3.2.2.

13. *p11 l4 what do you mean with 'single selected trajectories'?*

   We revised the text as follows:

   In the previous section, the transport pathways for a restricted number of trajectories were discussed. Here, for a broader view 20-day backward trajectories for the entire region of the Asian monsoon anticyclone are presented.

14. *p11 l32 'In the previous sections, we could show that the Asian monsoon is an effective circulation pattern in the UTLS that transports very young air masses (< 6 months) from the surface into the lower stratosphere up to ≈ 460 K.' This may have been mentioned before in the literature. You could rephrase this as 'we could show how the effective circulation pattern in the UTLS can be seen with CLaMS and MIPAS data', for example. Also it seems that all your conclusions will be drawn*

*from a single day case study. Additional statistical evidence from more modelling cases could help.*

As proposed by referee #3, we revised the text as follows.

In the previous sections using CLaMS model simulations and MIPAS HCFC-22 measurements, we could show that the circulation of the Asian monsoon is effective in transporting very young air masses ($< 6$ months) from the surface into the lower stratosphere up to $\approx 460\,\mathrm{K}$.

We agree that the most of the analysis is focused on 18 August 2008 as a case study. However, the results of the 3-dimensional CLaMS simulation for 18 August 2008 is a result of the interplay between convection, large-scale upward transport (driven by radiative heating), and the anticyclonic flow in the UTLS during the last weeks of the simulation. The same is true for the 40-day and 20-day backward trajectories as well as for the MIPAS measurements. Thus our results are representative for August 2008. To give a broader view, we already include 20-day backward trajectories showing different days during the monsoon season 2008 within the Appendix.

15. *12 4 is it a CLaMS simulation?*

    Yes, it is a CLaMS simulation as described in Sect. 2.1.

16. *Fig 8. The figure is difficult to read. The label at the color bar is very small and takes time to find. You could replace the title of the subplots '08081812 at 90 E' that is the same for all and put is in the cation. 'horizontal winds (black lines)', do you mean horizontal wind absolute value isolines? 'corresponding levels of pressure' corresponding to what? 'Pressure levels as white lines' would be better. Are the values really zero in W07 (lower left panel) or is the color scale?*

    Following the referee's advice we revised the figure caption as shown in Fig. 1 of this author comment (= Fig. 9 of the revised version of this

[Figure]

Figure 1: Latitude–theta cross sections at 90°E for the fraction of the India/China tracer for the simulation period (1 May 2007 - 18 August 2008 labeled as 'all') (a), for the Summer 08 (S08) pulse (b), for the Winter 07/08 (W07) pulse (c), and for the Summer 07 (S07) pulse (d) on 18 August 2008. The thermal tropopause (primary in black dots and secondary in red dots) and absolute horizontal winds (black lines for 30, 40, 50, and 60 m/s) are shown. The levels of pressure are marked by thin white lines. Note that the maximum value for the Winter 07/08 (W07) pulse (c) is 2.4%.

manuscript).

Further we removed 'corresponding' in all figure captions of the revised manuscript. The 'horizontal winds (black lines)' are calculated by $\sqrt{u^2 + v^2}$ and isolines for 30, 40, 50, and 60 m/s are shown and labelled. The maximum value for W07 is 2.4%.

17. *'demonstrating' may sound a bit strong for this context. 'Suggesting' or 'indicating' could fit better.*

We revised the sentence as follows.

[Figure]

Figure 2: Latitude–theta cross section of dΘ/dt showing the radiative heating above the Asian monsoon anticyclone for the western (30°E) and eastern mode (90°E) of the anticyclone on 18 August 2008. The thermal tropopause (primary in black dots and secondary in red dots) and absolute horizontal winds (black lines for 30, 40, 50, and 60 m/s) are shown. The pressure levels are marked by thin white lines.

Fractions of air from the India/China tracer for the time pulse for Winter 07/08 are below 2.4%, indicating that during winter in the absence of the Asian monsoon anticyclone the transport of boundary layer emissions from India/China into the stratosphere is insignificant weak.

18. *p12 l14 If still describing Fig. 9 better to keep the same paragraph.*

done

19. *Figure 10: again, wind speed contours, 'pressure levels' instead of 'corresponding levels of pressure'*

Following the referee's advice we revised the figure caption as shown in Fig. 1 of this author comment (= Fig. 11 of the revised version of this manuscript).

20. *p13 l11: attract the attention of the reader to the vertical dashed line immediately here.*

We added in the figure caption of Fig. 12 of the ACPD version (=Fig. 13 of the revised manuscript) the following explanation.

Top: The contribution of the three different time pulses S07, W07, and S08 (each set for a time period of 6 months marked the vertical doted lines) for the entire Earth's surface ($\Omega_{S07}$, $\Omega_{W07}$, $\Omega_{S08}$) to the tropical pipe between 30°S and 30°N at 550 K potential temperature from 1 October 2007 until the end of the simulation period (31 October 2008) (top, black lines).

21. *Fig 13 what is the dashed line?*

The dashed line marks the tropical pipe as described in the revised figure caption as follows.

The dashed line marks the tropical pipe which isolates tropical air masses largely from isentropic mixing with mid-latitude air (e.g., Plumb, 1996; Volk et al., 1996).

22. *p15 l12 please refer to the published version.*

The paper Hanumanthu et al., 2018 is not as yet published, therefore we refer here to Vernier et al. (2015, 2018); Brunamonti et al. (2018).

23. *p16 l17 'Thus, air masses in the upward spiral range are uplifted by diabatic heating across the (lapse rate) tropopause, which does not act as a transport barrier against this diabatic vertical transport process.': This assertion is likely to be 'consistent with previous studies'.*

We added the following references in the revised version of the paper.

Thus, air masses in the upward spiralling range are uplifted by diabatic heating across the (lapse rate) tropopause, which does not act

as a transport barrier against this diabatic vertical transport process. This transport across the tropopause is consistent with previous studies (e.g., Bergman et al., 2012; Garny and Randel, 2016; Ploeger et al., 2017).

24. *p16 l 22 occurs where?*

   We revised this paragraph as follows:

   Above 380 K, within the upward spiralling range above the anticyclone, young air masses from along the edge of the anticyclone originating in the tropical adjacent regions are mixed with air masses from inside the anticyclone mainly originating in India/China. Therefore, a significant fraction of air masses from the tropical adjacent regions is found within a widespread area around the anticyclone and above caused by the large-scale anticyclonic flow in this region, acting as a large-scale stirrer. This transport pattern up to 460 K is consistent with previous results focused on lower levels of potential temperature (up to $\approx$400 K (Vogel et al., 2014, 2016; Li et al., 2017).

25. *The last paragraph is a nice summary of the mechanism but it undoubtedly draws from the conclusions of many previous studies. This should somehow be acknowledged.*

   We added within the conclusions in the revised version of the manuscript some additional references.

**References**

[revised manuscript text omitted]

---

## Author Response (AR2)

**Author Comment to Referee #1**

ACP Discussions doi: 10.5194/acp-2018-724
(Editor - Peter Haynes)
**'Lagrangian simulations of the transport of young air masses to the top of the Asian monsoon anticyclone and into the tropical pipe'**
* * *
We thank Referee #1 for further guidance on how to to revise our paper. Following the reviewers advice we deleted the statement about the findings of Bourassa et al. (2012) in the introduction. In particular, we want to thank the reviewer for the elaborate language corrections. This was a very great support. Our reply to the reviewer comments is listed in detail below. Questions and comments of the referee are shown in italics. Passages from the revised version of the manuscript are shown in blue.

*The authors have made quite a number of modifications to the manuscript in response to the comments of the three referees. Most notably, they have substantially rearranged / rewritten the Discussion section (Section 4). There and elsewhere in the paper they have done a much better job of clearly summarizing their major findings and appropriately placing them into the context of previously reported results. For the most part I feel that the reviewers' concerns have been adequately addressed. I have only a few remaining suggestions (mostly minor wording changes where edits or additions were made during revision) that I feel should be considered before publication.*

**Specific comments:**

1. *P2, L18: are breaking −− > break*

    done

2. *P2, L22: ATAL −− > the ATAL*
   done

3. *P2, L32 - P3, L1: The conclusions of Bourassa et al. [2012] are accurately characterized here. However, their contention that the plume from the Nabro eruption only attained stratospheric altitudes through transport processes associated with deep convection in the Asian summer monsoon has been disproved by several subsequent publications, including Vernier et al. [Science, 2013], Fromm et al. [Science, 2013; JGR-A, 2014], and Fairlie et al. [ACP, 2014]. I feel that it is important to balance the assertions of Bourassa et al. reported in these sentences by noting that later studies showed that Nabro did in fact result in direct lower stratospheric (i.e., above 380 K) injection of volcanic aerosol, calling into question the critical role of monsoon convection in transporting volcanic aerosol from Nabro into the stratosphere. I have no objection to the more general statement in Section 4.1 (P16, L22-24; also present in the original submitted version) that the monsoon anticyclone affects the transport of volcanic sulphate aerosol, but these two sentences in the Introduction specifically about Nabro and the conclusions of Bourassa et al. need to either be amended to present the full picture or deleted.*

   Thank you for this feedback. We agree that for the full picture, a more detailed discussion is needed such as stated in the Discussion section. Following the reviewer's advice we deleted these two sentences about the conclusions of Bourassa et al. in the Introduction.

4. *P3, L16: both '2008' and 'normal' are repeated twice in this sentence. I think it would read better as: 'Further, there was a normal monsoon season in terms of rainfall over India in summer 2008.'*
   done

5. *P4, L5: a second −− > and a second*
   done

6. *P4, L32: greater than 380 K $--$ $>$ above 380 K; add a comma after the parenthetical in this line (before 'which')*
   done

7. *P5, L15 and L21: The authors might consider slightly rewording here to avoid starting two paragraphs in a row with the same phrase ('In the two-monsoon-season simulation')*
   done

8. *P7, L15: intercomparions $--$ $>$ intercomparisons*
   done

9. *P7, L26-34: I appreciate that information about the MIPAS HCFC-22 data quality has been added. However, three sentences on measurement precision and total uncertainty have been stuck in between sentences on vertical resolution. Similarly, a sentence on horizontal resolution has been inserted between statements about vertical resolution. This paragraph would flow much better if first measurement uncertainty were covered, then horizontal resolution, and then all statements pertaining to vertical resolution were grouped together. Also, 'consistent to' should be 'consistent with' (L29).*

   We revise the paragraph as follows:

   The precision of an individual data point of the MIPAS HCFC-22 measurement in the altitude region of the Asian monsoon tropopause is 7 to 8 pptv in terms of measurement noise. Parameter errors contribute to a total uncertainty of about 15 pptv in this region for each data point. Thus, the scatter of the HCFC-22 data points (e.g. as shown in Fig. 4) is consistent with the total error. Further it has to be noted that tropical HCFC-22 profiles from MIPAS seem to have a high bias below 30 km, that, however, is constant with altitude (Chirkov et al., 2016); thus, it does not affect the comparisons made here. The horizontal resolution (in terms of the full width at half maximum of the horizontal averaging kernel) increases from 300 km at 15 km altitude to 600 km

at 20 km altitude. In general, the limited vertical resolution of satellite remote sensing instruments like MIPAS needs to be taken into account in comparisons to model results. According to Chirkov et al. (2016), the vertical resolution (in terms of the full width at half maximum of the vertical averaging kernel) increases from about 3.3 km at 12 km to 5.5 km at 20 km altitude (see Fig. 2 in Chirkov et al., 2016). Given the rather smooth profiles expected in this study, however, the limited altitude resolution has a minor effect only; in contrast, it turns out to be crucial when highly structured profiles, such as typically occur at the edge of the polar vortex, are analysed.

10. *P8, L7-8: 'weaker' might be better than 'less strong'; consequences on −− > consequences for*
done

11. *P8, L15-16: 1 May 2008 until end of October −− > 1 May 2008, running through the end of October*
done

12. *P8, L16: a strong −− > strong*
done

13. *P9, L8: CO has already been used on p5 (L1), so this definition should be moved there (or deleted)*

The definition is moved to p5.

14. *P9, L13: high contribution −− > large contribution; fractions −− > fractions of*
done

15. *P9, L31: features at −− > features in; add a comma after 'prominent'*
done

16. *P10, L6-7: shows −− > show; Sect. 2.4) and longitude −− > Sect. 2.4); also shown are the longitude*
    done

17. *P10, L11: distribution −− > distributions*
    done

18. *P11, L8: The horizontal distribution of the tropical adjacent regions compared to the −− > The comparison between the horizontal distribution of the tropical adjacent regions and the*
    done

19. *P11, L11: the anticyclone and at 400K within −− > the anticyclone, while at 400K they are found within*
    done

20. *P11, L12: both tracers India/China and tropical adjacent regions show −− > both the India/China and the tropical adjacent region tracers show*
    done

21. *P12, L20: is ranging −− > ranges*
    done

22. *P12, L23: move 'therefore' to after 'anticyclone,'*
    done

23. *P12, L33: pattern −− > patterns*
    done

24. *P13, L3 and L16: 20 day −− > 20 days*
    done

25. *P13, L9: 'is inhomogeneous and not homogeneously distributed' seems redundant to me – isn't just saying 'is not homogeneously distributed' sufficient?*
done

26. *P14, L3: insignificant weak −− > insignificantly weak*
done

27. *P14, L5 and L9: 'tropical AR': either fully write out tropical adjacent regions (as done in most other places in the manuscript) or use the acronym TAR here*
done

28. *P14, L7-8: similar as for −− > similar to that for the*
done

29. *P14, L10: larger than from −− > larger than that from*
done

30. *P14, L11: details −− > for details*
done

31. *P14, L16: reanalysis −− > reanalyses*
done

32. *P15, L1: transport −− > transport of*
done

33. *P15, L23: Fig. 13 (top) −− > (Fig. 13 (top)) – or label the panels as is done in other figures so that this could be (Fig. 13a)*
done

34. *P15, L28-29: Lower fractions are from tropical the Indian −− > Smaller fractions are estimated for the tropical Indian*
done

35. *P15, L30: are lower −− > are each smaller*
done

36. *P16, L10: a debate −− > debate*
done

37. *P16, L12: depending on the used reanalysis data −− > depending on the reanalysis data used*
done

38. *P16, L13: there is a strong intraseasonal variability of boundary source regions −− > there is strong intraseasonal variability in the contributions of different boundary layer source regions*
done

39. *P16, L18: I found the wording 'Further, in addition to previous studies, we' awkward and confusing, and I don't think that it is exactly what the authors want to say. Rather, I think they mean something along the lines of 'Further, going a step beyond previous studies, we'*
done

40. *P17, L8: This sentence ('... into the outer anticyclonic flow of the anticyclone follow the flow around the anticyclone') is hard to read because there are too many instances of 'anticyclone' or 'anticyclonic' – are they all strictly necessary?*
done

41. *P17, L22-23: found a −− > found; fractions around −− > fractions of around*

done

42. *P17, L24: consistent to − − > consistent with; about 6% contributions − − > about a 6% contribution*
   done

43. *P17, L26: Similar to the comment above about P16, L18, I dont think that 'in addition to' is quite right here – I think it would be better to say something like 'However, going beyond the results of Randel and Ploeger, we'*
   done

44. *P18, L9: is calculated, resulting in up to 44% at 360K (up to 35% at 380 K) end of October − − > was calculated to result in up to 44% young air at 360K (up to 35% at 380 K) at the end of October*
   done

45. *P18, L12-13: end of October − − > at the end of October; also delete 'compared to 2012' and change 'This difference is' to 'The differences between 2008 and 2012 are'*
   done

46. *P19, L1: consistent to − − > consistent with*
   done

47. *P19, L5: a anticyclonic − − > an anticyclonic*
   done

48. *P20, L7: add a semicolon after 'months)'*
   done

49. *P20, L18: regions −− > region*
   done

50. *P38, Fig. 10 caption: Southeast Asia (SEA) −− > tropical adjacent regions (TAR)*
   done

51. *P40, Fig. 13 caption: marked the vertical doted lines −− > marked by the vertical dotted lines*
   done

52. *P41, Fig. 14 caption: add a semicolon after 'months)'; pipe which isolates tropical air masses largely −− > pipe, which largely isolates tropical air masses*
   done

**(Editor - Peter Haynes)**
**'Lagrangian simulations of the transport of young air masses to the top of the Asian monsoon anticyclone and into the tropical pipe'**

─────────────────────────────────

We thank Referee #2 for further guidance on how to to revise our paper, however sometimes it was hard to follow. Our reply to the reviewer comments is listed in detail below. Questions and comments of the referee are shown in italics. (R1) is used to refer to the revised version of the manuscript and (R2) to the current revision (shown in blue). R1 is also used to refer to the reviewer comment to the ACPD version of the paper.

*Dear Authors, thank you for considering my previous comments. I am pleased with the revised version of the manuscript. In particular, with the better linking to previous results in the published literature especially within the updated discussion section and the conclusions, which was my major concern with the publication. However, I still would like to see some minor (and in part technical) corrections before the manuscript can be published from my point of view. The corresponding comments are given below. Here, the page, line and figure numbers are those of the ACPD version of the manuscript. If (R1) is appended the numbers refer to the revised version.*

**Additional minor comments:**

1. *Figure A1 (R1): Revise the figure caption. Remove that the trajectories are started on '18 August 2008'.*
   done

2. *P1 L17: Please clarify which conditions you mean with: '...under these conditions.'*

We revises the sentence as follows:

R1: In the upward spiralling range, air masses are uplifted by diabatic heating across the (lapse rate) tropopause, which does not act as a transport barrier under these conditions. $--->$
R2: In the upward spiralling range, air masses are uplifted by diabatic heating across the (lapse rate) tropopause, which does not act as a transport barrier in contrast to the extratropical tropopause.

*These comments refer to my previous comments on the ACPD version of the manuscript and are listed in the respective section.*

**Regarding previous "General comments":**

1. *P19 L24 (R1): Upward transport in the edge region is clear at 440K from Fig.8 (R1). If only 440K is meant then write at 440K.*

   We revised the sentence in Sect. 5 as follows:

   Further between 420 K and 440 K, highest contributions from young air masses are found around the edge of the anticyclone ...

2. *Thank you for your comment. Please, consider to add parts of your clarification regarding the transport on the eastern and western side of the anticyclone above and below the tropopause to the revised manuscript.*

   Following the reviewer's advice, we added the following paragraph to Sect. 3.3.1:

   The focus of our study is to demonstrate that in the upward spiralling range (above 360 K) a slow upward transport is found over the region of entire anticyclone (west and east mode) with diabatic heating rates of up to 1–1.5 K inferred from ERA-interim. Our 40-day backward trajectory calculations (see Fig. 6 and 7) demonstrate that a diabatic heating above 360 K is found at both the western and eastern side of

We decided to not include parts of our clarification regarding the transport on the eastern and western side of the anticyclone 'below' the tropopause. Transport 'below' the tropopause is not the main focus within our paper therefore we think that too much additional information would make the paper more difficult to read.

**Regarding previous "Specific comments":**

1. *P3 L7-10: Thank you for the revision...However, I am sorry, but I cannot find this relation in the reference you have given! I see anomalous positive rainfall over India associated with La Nina (during summer) and also with additional La Nina conditions in the winter before (blue bars) in Fig. 1 c)-h) of Chakraborty, (2018). As this is not key to your analysis consider to cut this part regarding rainfall and ENSO...*

   The relation is found in Fig. 2c in Chakraborty (2018). We added this information in the paper (R2) in Sect. 1 as follows:

   There is evidence that a strong La Niña in winter (e.g. 2007/08 (DJF) according to the Oceanic Niño Index[1]) in combination with La Niña conditions during the subsequent summer (as in 2008) is correlated with normal rainfall over India with a certain variability in precipitation between different Indian regions (e.g., see Fig. 2c in Chakraborty, 2018).

2. *P5 L3-5: Thank you for the clarification. I agree that the impact could be small and that other uncertainties might be more important. However, I still think that this relation (the equality) only holds during mixing time steps, i.e. time steps of tracer initialisation.*

   We changed the sentence as follows:
* * *
[1]see e.g. `http://ggweather.com/enso/oni.htm`

R1: Within the model boundary layer, the sum of all the different emission tracers ($\Omega_i$) including the emission tracer for the background (remaining surface) is equal to 1 ($\Omega = \sum_{i=1}^{n} \Omega_i = 1$, see Table 1).
$-- >$
R2: Within the model boundary layer, the sum of all the different emission tracers ($\Omega_i$) including the emission tracer for the background (remaining surface) is equal to 1 ($\Omega = \sum_{i=1}^{n} \Omega_i = 1$, see Table 1) at the mixing time step.

3. *P9 L26-27: I think it would be beneficial to also relate to the findings of Pan et al. (2016) here!*

Following the reviewer's advice, we added in version (R2) of the paper a reference in Sect. 3-1-2 to Sect. 3.1.1 to strengthen the link between these sections and to Pan et al. (2016) as follows:

Thus below 350–360 K smaller mixing ratios of HCFC-22 are measured than above, indicating that below the western mode of the anticyclone there exists no upward transport from boundary sources for HCFC-22. The enhanced values of HCFC-22 within the western mode and below the thermal tropopause along the longitude–theta cross sections at 25°N (Fig. 4b) confirm the horizontal westward transport within the Asian monsoon anticyclone as found for the India/China tracer in the CLaMS simulations (see Sect. 3.1.1 and Fig. 2c), which is consistent with CO simulations by Pan et al. (2016).

4. *P11 L5: You are releasing trajectories in a broad box not only within the anticyclone. Please consider my previous comment!*

In the current version (R2) of the paper, we revised this 1st paragraph in Sect.3.2.2 as follows:

In the previous section, the transport pathways for a restricted number of trajectories in the region of the Asian monsoon anticyclone were discussed. Here, for a broader view 20-day backward trajectories between

0–160°E and 10–60°N are presented including the entire region of the Asian monsoon anticyclone. In this longitude-latitude region (0–160°E and 10–60°N), backward trajectories are calculated on a $1.0° \times 0.5°$ longitude-latitude-grid at 360 K, 380 K, 400 K, 420 K, and 440 K starting on 18 August 2008.

5. *P11 L14: I do not think, that the studies by Randel et al. (2010) and Ploeger et al. (2017) make any claim to the homogeneity of the uplift. Please check if this is meant and stated in these references and revise accordingly.*

We agree that Randel et al. (2010) and Ploeger et al. (2017) do not make any claim to the homogeneity of the uplift. For clarification, we revised the sentence as follows.

Thus the upward transport in the region of the anticyclone is not homogeneously distributed over the entire anticyclone at a certain day. In previous studies (e.g., Randel et al., 2010; Ploeger et al., 2017) climatological mean values (over the monsoon season of several years) are presented which can not be used to analyse the inhomogeneity of the upward transport in the region of the anticyclone at a certain point in time.

6. *P14 L6-8: Thank you for the clarification. Please consider to also add the reference to Vogel et al. (2015) in the revised manuscript.*

We added the reference in Sect. 3.3.2 (R2) as follows:

Further, our findings show that air masses from India/China, thus mainly from the Asian monsoon anticyclone (see Vogel et al., 2015), contribute to a smaller fraction of the composition of air within the tropical pipe at 550 K; the major part is from Southeast Asia and the tropical Pacific.

**Regarding previous "Minor suggestions/corrections":**

1. *To 1 and 2: I am sorry, I can not see any changes here. Please consider to include my previous suggestions, especially the second one.*
done

We are sorry, but we didn't catch your point.

To 1 (R1): *P1 L11: Either change to "Second, these air masses..." or "Second, air masses are uplifted within the anticyclone..." or something similar.*

In the revised version (R1/R2) is stated: Second, air masses are uplifted from about $360\,\mathrm{K}$ up to $460\,\mathrm{K}$ within 'an upward spiralling range' within a few months.

To 2 (R1):*P1 L14: As before, maybe clarify by changing your sentence to something like: "Third, transport of air masses affected by the Asian monsoon (anticyclone)..." or something similar.*

In the revised version (R1/R2)is stated: Third, transport of air masses occurs within the tropical pipe up to $550\,\mathrm{K}$ associated with the large-scale Brewer-Dobson circulation within $\sim$ one year.

2. *To 3: consider to cut "the most" (cf. Conclusion of Mason and Anderson, 1963)*

We are not sure what exactly is your point, but we have revised the sentence in the current version (R2) of Sect. 1 as follows

R1: The Asian summer monsoon is associated with deep convection over the Indian subcontinent and is the most pronounced circulation pattern in boreal summer with an anticyclonic flow that extends from the upper troposphere into the lower stratosphere (UTLS) region (e.g., Li et al., 2005; Randel and Park, 2006; Park et al., 2007). $-->$

R2: The Asian summer monsoon is associated with deep convection over the Indian subcontinent and with an anticyclonic flow that extends from the upper troposphere into the lower stratosphere (UTLS) region which is the most pronounced circulation pattern in these altitudes during boreal summer (e.g., Mason and Anderson, 1963; Li et al., 2005; Randel and Park, 2006; Park et al., 2007).

...according to the first sentence in the Conclusion of Mason and Anderson (1963): 'With the exception of the polar vortex itself, the Asian 100-mb. anticyclone is the most intense and persistent circulation found at this pressure surface over the Northern Hemisphere.'

3. *To 10: P13 L1 (R1): this should read "...transport of...." ("of" is missing)*

We think that in the following sentence there is missing no 'of':

The patterns of $\Delta\Theta$ at $360\,\mathrm{K}$ within the anticyclone and in the tropics are very patchy, reflecting that the strong upward transport in this region is caused by single convective events.

4. *To 12: switch "the" and "tropical"*

The sentence 'To 12' is as follows. We are sorry, but we didn't catch your point.

[revised manuscript text omitted]